# A nanoparticle-based sonodynamic therapy reduces *Helicobacter pylori* infection in mouse without disrupting gut microbiota

Tao Liu [1,2,3], Shuang Chai [1,2,3], Mingyang Li [1,2,3], Xu Chen[1,2,3], Yutao Xie[1,2,3], Zehui Zhao[1,2,3], Jingjing Xie[1,2,3], Yunpeng Yu [1,2,3], Feng Gao [1], Feng Zhu[4] & Lihua Yang [1,2,3] ✉

Infection by *Helicobacter pylori*, a prevalent global pathogen, currently requires antibiotic-based treatments, which often lead to antimicrobial resistance and gut microbiota dysbiosis. Here, we develop a non-antibiotic approach using sonodynamic therapy mediated by a lecithin bilayer-coated poly(lactic-co-glycolic) nanoparticle preloaded with verteporfin, Ver-PLGA@-Lecithin, in conjunction with localized ultrasound exposure of a dosage permissible for ultrasound medical devices. This study reveals dual functionality of Ver-PLGA@Lecithin. It effectively neutralizes vacuolating cytotoxin A, a key virulence factor secreted by *H. pylori*, even in the absence of ultrasound. When coupled with ultrasound exposure, it inactivates *H. pylori* by generating reactive oxygen species, offering a potential solution to overcome antimicrobial resistance. In female mouse models bearing *H. pylori* infection, this sonodynamic therapy performs comparably to the standard triple therapy in reducing gastric infection. Significantly, unlike the antibiotic treatments, the sonodynamic therapy does not negatively disrupt gut microbiota, with the only major impact being upregulation of *Lactobacillus*, which is a bacterium widely used in yogurt products and probiotics. This study presents a promising alternative to the current antibiotic-based therapies for *H. pylori* infection, offering a reduced risk of antimicrobial resistance and minimal disturbance to the gut microbiota.

H*elicobacter pylori* (*H. pylori*) is a human-transmittable pathogen that infects over 50% of the world's population[1]. For example, in China, the *H. pylori* infection rate for adults is 40–50%[2]. *H. pylori* infection can lead to various complications including chronic gastritis, gastric ulcers[3] and even gastric cancer[4]. Notably, chronic *H. pylori* infection is classified as a Class I human carcinogen[5] and is the most significant known risk factor for gastric cancer[6], which stands as the third leading cause of cancer-related deaths worldwide.

Currently, standard clinical treatments for gastric *H. pylori* infection are triple therapy (a regimen comprising two antibiotics and a proton pump inhibitor (PPI)) and quadruple therapy (a regimen comprising two antibiotics plus a PPI and a bismuth salt)[7], both of which eradicate *H. pylori* with antibiotics administered orally over a 7- or 14-day period[7]. Such a reliance on oral antibiotics, however, has resulted in antimicrobial resistance in *H. pylori*[8–10] and consequently increased both the failure rate[11] and the recurrence rate[7] after standard

[1]Hefei National Research Center for Physical Sciences at the Microscale, University of Science and Technology of China, Hefei, Anhui 230026, China. [2]CAS Key Laboratory of Soft Matter Chemistry, University of Science and Technology of China, Hefei, Anhui 230026, China. [3]School of Chemistry and Materials Science, University of Science and Technology of China, Hefei, Anhui 230026, China. [4]Division of Life Science and Medicine, University of Science and Technology of China, Hefei, Anhui 230026, China. ✉e-mail: lhyang@ustc.edu.cn

clinical treatments. Indeed, clarithromycin-resistant *H. pylori* is one of the 12 antibiotic-resistant bacteria that pose the greatest threat to human health according to the World Health Organization[12]. Moreover, antibiotic-based *H. pylori* eradication therapies cause patients to readily exhibit gut microbiota dysbiosis[13] (Fig. 1, Supplementary Data 1), which in some patients remains significant even at ≥12 months post treatment (Fig. 1, Supplementary Data 1) and should raise concern

because the gut microbiota is crucial for human health[14] and its dysbiosis is closely related to the occurrence of diverse diseases[15]. In addition, antibiotic-based standard clinical therapies have neglected vacuolating cytotoxin A (VacA), a major virulence factor in the colonization and infection of *H. pylori* in the stomach[16]. Therefore, gastric *H. pylori* infection needs treatments better than antibiotic-based *H. pylori* eradication therapies.

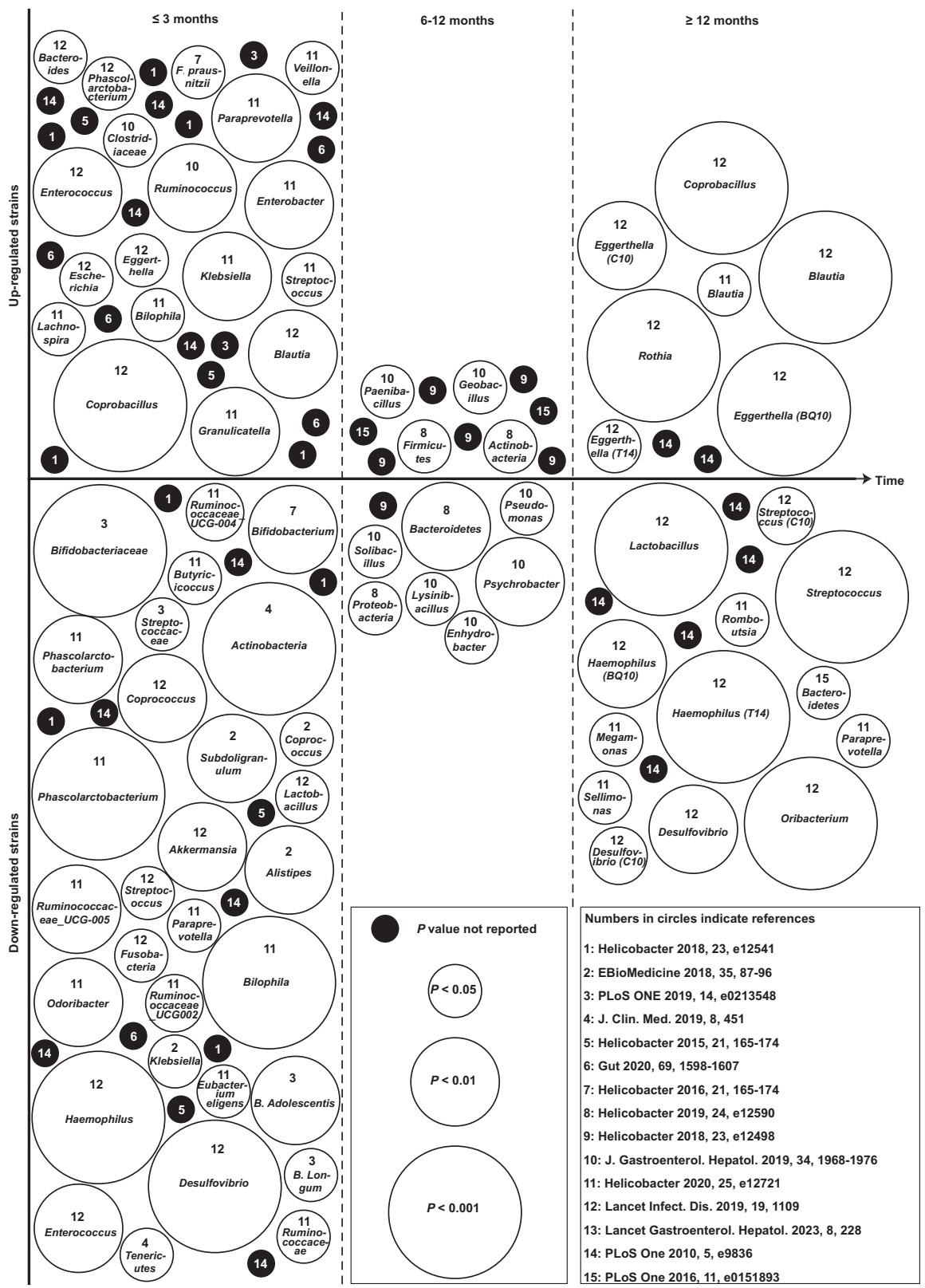

**Fig. 1 | Up- and down-regulated human gut commensal bacterial strains after antibiotic-based *H. pylori* eradication therapies.** Each circle, whether solid black or hollow white, represents a gut commensal bacterial strain that has been reported to experience change in abundance after antibiotic-based *H. pylori* eradication treatment, and the number enclosed within the circle corresponds to the relevant reference, which can be located in the reference list provided at the bottom right. A solid black circle represents a commensal bacterial strain for which the relevant reference did not report the *P* value pertaining to its change of abundance in host gut after antibiotic-based *H. pylori* eradication treatment. The hollow white circles represent commensal bacterial strains, for which the relevant references have provided *P* values pertaining to their changes of abundance in host gut after antibiotic-based treatment for *H. pylori* eradication, and the sizes of these white circles progressing from smallest through medium to largest correspond to *P* values of <0.05, <0.01, and <0.001, respectively. More detailed information is provided in Supplementary Data 1. Source data are provided as a Source Data file.

To this end, researchers have reported many nonantibiotic substances active against *H. pylori*[17–22]; unfortunately, the as-reported organic substances[18–21] in general have wide-spectrum activity − linolenic acid-incorporated liposomes, for example, kill both *H. pylori*[19] (gram-negative) and *Propionibacterium acnes*[23] (gram-positive) − and consequently may cause gut microbiota dysbiosis after oral administration, while the as-reported inorganic substances[17,22] unanimously lack biocompatibility. Recently, probiotics (i.e., gut commensal bacteria from healthy donors) have been proposed to outcompete gastric *H. pylori*[24]; nevertheless, in the gastric microbiota of *H. pylori*-infected patients, *H. pylori* is the single most abundant bacterium (accounting for 72–97%) despite the diverse nature of gastric microbiota for their *H. pylori*-negative counterparts[25], which therefore dims the chance for probiotics to out-compete *H. pylori* in an environment that naturally favors the latter. In recent years, dynamic therapies that eradicate target cells with reactive oxygen species (ROS) generated responsively in situ have attracted significant research attention because ROS can simultaneously oxidize diverse cellular substances (e.g., nucleic acids, proteins, and lipids) crucial for proper cell function[26] and consequently eliminate drug-resistant bacteria[27] while delaying the onset of bacterial resistance emergence[28,29]. For example, intragastric photodynamic therapy triggered by ultraviolet light has been proposed to treat gastric *H. pylori* infection[30]; unfortunately, that ultraviolet light is a well-recognized carcinogen limits its translational potential. In 2022, a study reported the first use of sonodynamic therapy in treating *H. pylori* infection[31]; nevertheless, to achieve >90% *H. pylori* killing in vitro, that study needed an ultrasound (US) output (1.5 W/cm²), which is >2-fold of the maximum safety limit of 0.72 W/cm² established for ultrasound medical devices[32,33]. Additionally, none of the previous efforts have specifically addressed the adverse roles of VacA in gastric *H. pylori* infection. Ideally, an effective therapy for gastric *H. pylori* infection should not only exhibit potent antibacterial activity against *H. pylori* but also maintain the integrity of a patient's gut microbiota, ensure biosafety, facilitate the elimination of VacA, and circumvent antimicrobial resistance mechanisms. However, to date, no such comprehensive therapy has been reported in the literature to the best of our knowledge.

To close this gap, we herein develop a non-antibiotic approach using sonodynamic therapy mediated by a nanoparticle composed completely of components approved for clinical use by the Food and Drug Administration of the United States of America (U.S. FDA) in conjunction with localized ultrasound (US) exposure of a dosage (at 0.5 W/cm² for 10 min) within the range allowable for ultrasound medical devices (i.e., output of ≤0.72 W/cm²[32,33] and exposure time of ≤15 min[32]). As a proof-of-concept, the model nanoparticle we used is a lecithin bilayer-coated poly (lactic-co-glycolic acid) (PLGA) nanoparticle preloaded with verteporfin (Ver), Ver-PLGA@Lecithin, in which all three components (i.e., lecithin, PLGA, and verteporfin) have been approved for clinical use by the U.S. FDA[34–36]. In the absence of US exposure, Ver-PLGA@Lecithin efficiently removed VacA via protein corona formation. In response to US exposure at only 0.5 W/cm² for 10 min, Ver-PLGA@Lecithin efficiently generated ROS and killed *H. pylori* even in simulated gastric fluid. In female mouse models bearing gastric *H. pylori* infection, the sonodynamic therapy mediated by Ver-PLGA@Lecithin in conjunction with US exposure (at 0.5 W/cm²

for 10 min) on the skin over stomach reduced *H. pylori* infection as effectively as the standard triple therapy. Significantly, unlike antibiotic-based *H. pylori* eradication therapies, the sonodynamic therapy does not negatively disrupt gut microbiota, with the only major impact being upregulation of *Lactobacillus*, a bacterium widely used in yogurt products and probiotics, as the only significant perturbation to gut microbiota composition. This work may have implications in offering an alternative therapy for *H. pylori*-infected patients.

## Results and discussion

### PLGA@Lecithin nanoparticles efficiently remove VacA

In sonodynamic therapy, an ultrasound sensitizer (USS) (i.e., sonosensitizer) is indispensable. Note that VacA is an 88-kDa soluble protein secreted by *H. pylori* into the extracellular space[16] (Fig. 2a) and that nanoparticles rapidly (within 0.5 min[37]) adsorb proteins from environmental fluids to form protein coronas[37–39]. We hence propose a sonosentizing nanoparticle as the sonosensitizer for the sonodynamic therapy to achieve VacA removal even in the absence of US exposure. Considering that poly(lactic-co-glycolic) acid (PLGA) is a polymer approved by the U.S. FDA for clinical uses[34] and that PLGA nanoparticles are widely used drug carriers, we used PLGA nanoparticles as the platform to construct the model nanosonosensitizer. As the surface chemistry of a nanoparticle plays crucial roles in protein corona formation[38,40–42], we used two PLGA nanoparticles that are similar in size and surface charge but differ in surface coating materials (Fig. 2b), a lecithin bilayer-coated PLGA nanoparticle (PLGA@Lecithin) (d ~ 167.8 nm, ζ ~ −30.3 mV) (Supplementary Fig. 1) and a PEGylated PLGA nanoparticle (PLGA@PEG) (d ~ 196.3 nm, ζ ~ −35.8 mV) (Supplementary Fig. 2), in an effort to identify the one more efficient in removing VacA. Of note, both lecithin and polyethylene glycol (PEG) have been approved for clinical use by the U.S. FDA[43] and PEGylation cannot completely eliminate protein adsorption but rather enrich the adsorption of certain proteins[41,44,45], although PEG is commonly viewed as the current gold-standard stealth material for repelling protein adsorption.

We first examined whether proteins naturally available in *H. pylori* culture supernatant (HCS) adsorb onto the model nanoparticles by quantifying the total amounts of adsorbed proteins on nanoparticles recollected after their incubation in and then separation from HCS-supplemented water. For both PLGA@Lecithin and PLGA@PEG, protein adsorption occurred rapidly within 100 min after their incubation initiation but reached a saturated steady phase afterwards (Fig. 2c). Notably, at any examined time point, the total amount of adsorbed proteins on PLGA@Lecithin was ~2-fold that on PLGA@PEG (Fig. 2c). For example, at 12 h after incubation in HCS-supplemented water, the total amount of adsorbed proteins on PLGA@Lecithin was ~0.154 mg/mL, while that on PLGA@PEG was ~0.075 mg/mL (Fig. 2c and Supplementary Fig. 3b). Obviously, both PLGA@Lecithin and PLGA@PEG can adsorb proteins from HCS (Supplementary Fig. 4), and in doing so, PLGA@Lecithin is more efficient than PLGA@PEG.

The adsorption of proteins from HCS onto the model nanoparticles was further verified with cryo-electron microscopy (Cryo-EM) and sodium dodecyl sulfate–polyacrylamide gel electrophoresis (SDS–PAGE). Under cryo-EM, both PLGA@Lecithin and PLGA@PEG

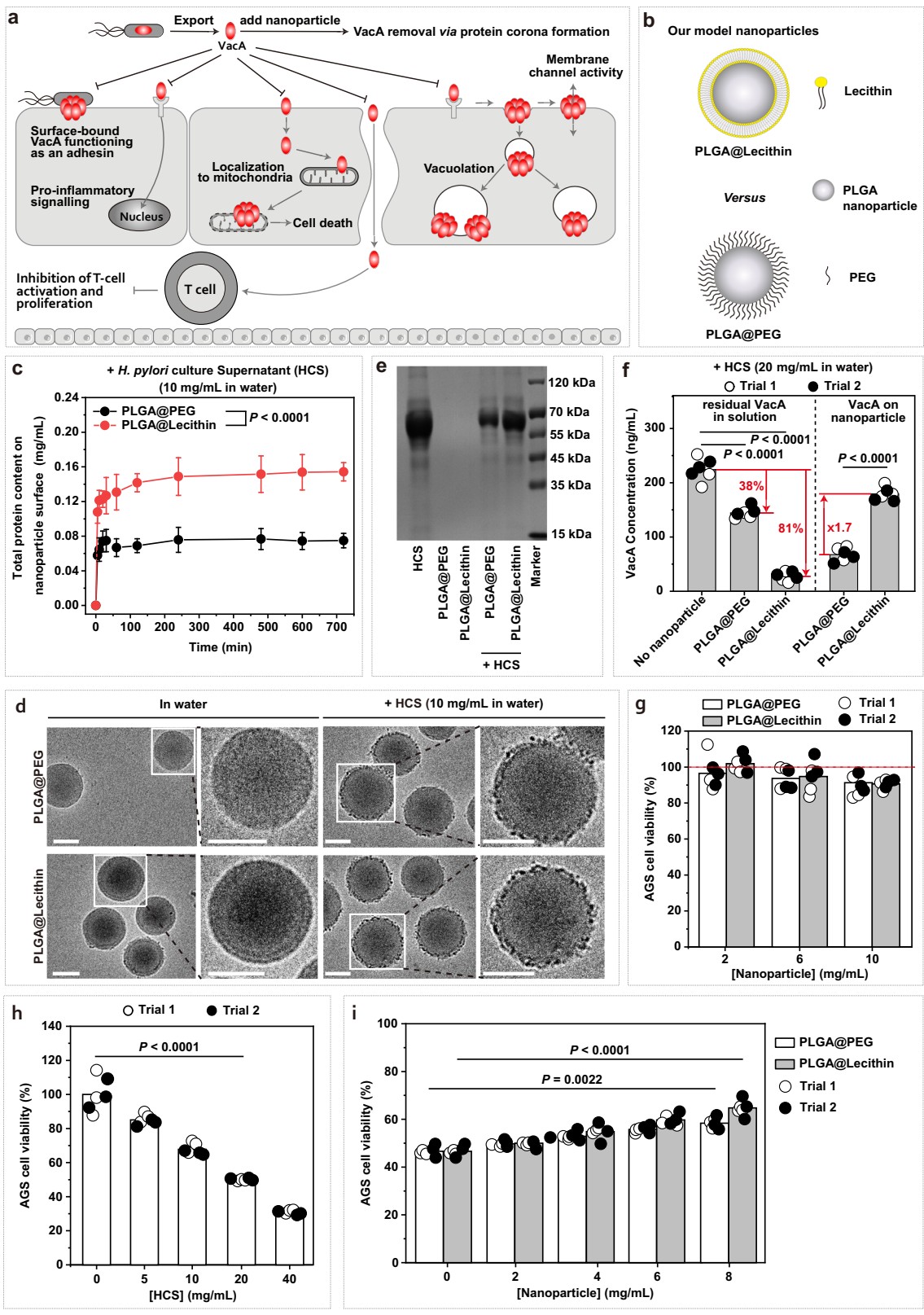

incubated (for 12 h) in water appeared spherical with smooth surfaces that were nearly uniform in contrast, whereas their counterparts incubated in HCS-supplemented water, though still spherical, acquired rough and wrinkled surfaces that were nonuniform in contrast and exhibited many dark dots (Fig. 2d). In SDS–PAGE gels, both PLGA@-Lecithin and PLGA@PEG after a 12-h incubation in HCS-supplemented water (10 mg/mL) acquired similar protein bands as those exhibited by

HCS-supplemented water, whereas their counterparts incubated in water exhibited none. Combined, these results confirmed the adsorption of proteins (and possibly other biomolecules) naturally present in HCSs onto the model nanoparticles (Fig. 2e). Moreover, after 12-h incubation in HCS-supplemented water, PLGA@Lecithin exhibited more dark dots under Cryo-EM and darker protein bands in SDS–PAGE gels compared to PLGA@PEG, indicating higher efficiency

**Fig. 2 | Selecting the nanoparticle efficient in removing vacuolating cytotoxin A (VacA). a** Schematic illustration of the pathways by which VacA secreted by *H. pylori* attacks gastric epithelial cells and their blockage due to VacA removal via protein adsorption onto externally added nanoparticles. **b** Schematic illustration of the model nanoparticles, PLGA@Lecithin and PLGA@PEG. **c** Contents of total adsorbed proteins on PLGA@PEG and PLGA@Lecithin recollected at different time-points after the initiation of their incubation in water supplemented with *H. pylori* culture supernatant (HCS). Data points are reported as the average ± standard deviation (*n* = 3 independent trials). Statistical analysis was carried out with a two-way analysis of variance (ANOVA) with Tukey's multiple-comparison test. **d** Cryo-EM images of (top) PLGA@PEG and (bottom) PLGA@Lecithin recollected after 12-h preincubation (left) in water and (right) in HCS-supplemented water. Scale bar = 50 nm. Two times, each experiment was repeated independently with similar results. **e** Photographs of SDS–PAGE gels with absorbed proteins on PLGA@PEG and PLGA@Lecithin (80 mg/mL) with and without 12-h preincubation

in HCS-supplemented water (10 mg/mL), with that of HCS-supplemented water (5 mg/mL) included as a reference. **f** Contents of adsorbed VacA on PLGA@PEG and PLGA@Lecithin (8 mg/mL) recollected after 24 h of preincubation in HCS-supplemented water (20 mg/mL) and those of residual VacA in the as-treated solution. Bar heights are reported as averages of two independent trials (*n* = 3 in each independent trial). Statistical analysis was carried out with a one-way ANOVA with Tukey's multiple-comparison test. **g–i** Viability ratios of human gastric adenocarcinoma cells (AGS cells) (**g**) after 24 h of treatment with PLGA@PEG and PLGA@Lecithin at differing doses in PBS. **h** 24-h incubation in HCS-supplemented PBS at differing concentrations (**i**) after 24-h incubation in HCS-supplemented PBS (20 mg/mL) that was pretreated (for 12 h) with a nanoparticle (PLGA@PEG or PLGA@Lecithin) at differing doses. Bar height represents the average of two independent trials (*n* = 3 in each independent trial). Statistical analysis was carried out with a (**h**) one-way and (**i**) two-way ANOVA with Tukey's multiple-comparison test. Source data are provided as a Source Data file.

in adsorbing proteins for the former than the latter, consistent with the quantitative protein adsorption assays above (Fig. 2c and Supplementary Fig. 3b).

Note that in SDS–PAGE gels (Fig. 2e), both PLGA@Lecithin and PLGA@PEG exhibited protein bands of 70–120 kDa, which cover the size of VacA (88 kDa). We hence monitored both the content of adsorbed VacA on nanoparticles after 12-h incubation in HCS-supplemented water and that of residual VacA in the solution (Supplementary Fig. 5a). Our results (Fig. 2f and Supplementary Fig. 5b) revealed that although water supplemented with HCS from the *H. pylori* strain we used naturally contained VacA to appreciable extent, its VacA content was reduced significantly after incubation with the model nanoparticles, which resulted in the accumulation of VacA on the model nanoparticles. Notably, PLGA@Lecithin was significantly more efficient than PLGA@PEG both in reducing the VacA content in HCS-supplemented water (>80% *versus* 38%) and in adsorbing VacA (~2-fold higher) (Fig. 2f and Supplementary Fig. 5b).

VacA contributes to the colonization and infection of *H. pylori* in the stomach through multiple effects on gastric epithelial cells, including targeting their plasma membrane permeability[16] (Fig. 2a). Therefore, we next examined whether VacA removal by the model nanoparticles helps detoxify HCS (Supplementary Fig. 6a) using human gastric adenocarcinoma (AGS) cells as the representative for gastric epithelial cells, as in prior studies on how *H. pylori* infection and/or VacA influence gastric epithelial cells[46–48]. For AGS cells, both PLGA@PEG and PLGA@Lecithin lacked intrinsic cytotoxicity even up to 10 mg/mL (in PLGA dose) (Fig. 2g), while phosphate-buffered saline (PBS) supplemented with HCS from the *H. pylori* strain used in this work was intrinsically cytotoxic (Fig. 2h). Nevertheless, preincubating (for 12 h) the HCS-supplemented PBS with either PLGA@PEG or PLGA@Lecithin mitigated its cytotoxicity (Fig. 2i and Supplementary Fig. 6b). For example, HCS-supplemented PBS (20 mg/mL) reduced the average AGS cell viability ratio to ~46.6%, but its preincubation with PLGA@PEG and PLGA@Lecithin (8 mg/mL in PLGA dose) enhanced the average AGS cell viability ratio to 53.9% and 64.7%, respectively. Clearly, both PLGA@PEG and PLGA@Lecithin can detoxify HCS, and in doing so, PLGA@Lecithin is more efficient than PLGA@PEG. Taken together, the above results suggest that, both in removing VacA and in detoxicifying HCS, PLGA@Lecithin is superior to PLGA@PEG. Therefore, we used PLGA@Lecithin as the carrier for constructing the model nanosonosensitizer.

**Ver-PLGA@Lecithin as a nanosonosensitizer for *H. pylori* killing**
We next determined what sonosensitizer (USS) to preload into PLGA@Lecithin, to obtain the model nanosonosensitizer. Although scarce, reported sonosensitizers can be classified into two classes: small molecular organic sonosensitizers and inorganic sonosensitizers[49]. As inorganic sonosensitizers generally lack biocompatibility, we herein focused on small molecular organic sonosensitizers. In particular, we

selected the to-be-preloaded USS among verteporfin (Ver)[50], indocyanine green (ICG)[51], and chlorin e6 (Ce6)[52], three recently demonstrated small molecular sonosensitizers[50–52]. By preloading a USS into the PLGA core of PLGA@Lecithin (Supplementary Fig. 7), we obtained three USS-PLGA@Lecithin nanoparticles that are similar both in size and in surface charge, Ver-PLGA@Lecithin (d ~ 196.2 nm, ζ ~ −36.8 mV), ICG-PLGA@Lecithin (d ~ 197.9 nm, ζ ~ −38.1 mV), and Ce6-PLGA@Lecithin (d ~ 191.7 nm, ζ ~ −37.7 mV) (Supplementary Information, Supplementary Figs. 8–10). To determine the nanosonosensitizer for the sonodynamic therapy, we compared these USS-PLGA@Lecithin nanoparticles in ROS generation and *H. pylori* killing.

Using DCFH as the ROS probe[53] (Fig. 3a), we found that upon US exposure (at an output power density of 0.5 W/cm$^2$ for 10 min), all three USS-PLGA@Lecithin nanoparticles rendered DCFH solution green fluorescent, whereas their empty counterpart PLGA@Lecithin did not (Fig. 3b), indicative of ROS generation due to their preloaded USS. Moreover, the observed ROS production is dependent on the US output power density (Fig. 3c), the US exposure period (Fig. 3d), and the nanoparticle dose (in dose of preloaded USS) (Fig. 3e), indicative of US-triggered ROS production by their impregnated USS. In addition, using probes selective for specific ROS types, such as singlet oxygen sensor green (SOSG) for singlet oxygen ($^1O_2$)[54], we further demonstrated singlet oxygen ($^1O_2$) − but neither $O_2$·$^-$ nor •OH − as the ROS generated by the USS-PLGA@Lecithin nanoparticles upon US exposure (Fig. 3f, Supplementary Figs. 11–13, Supplementary Information). Notably, in the US-triggered generation of ROS and in that of $^1O_2$, Ce6-PLGA@Lethicin and Ver-PLGA@Lecithin were comparably efficient, whereas ICG-PLGA@Lecithin was the least efficient.

Next, we compared these USS-PLGA@Lecithin nanoparticles in *H. pylori* killing (Fig. 3g). As US exposure alone at 0.5 W/cm$^2$ for 10 min killed only <4% of inoculated *H. pylori* cells (Fig. 3h and Supplementary Fig. 14a, b), we used this US exposure dosage (at 0.5 W/cm$^2$ for 10 min) for all following antibacterial assays and animal studies throughout this work unless specified otherwise; notably, this US exposure dosage is within the safe range for ultrasound medical devices (at ≤0.72 W/cm$^2$[32,33] and for ≤15 min[32]). Without US exposure (i.e., No US), all three USS-PLGA@Lecithin particles killed only <3% of inoculated *H. pylori* cells even at 100 μg/mL (in USS dose); nevertheless, upon US exposure (i.e., US), they all killed *H. pylori* in a USS dose-dependent manner (Fig. 3h and Supplementary Fig. 14c–e), with Ver-PLGA@Lethicin and Ce6-PLGA@Lethicin both offering 99.9% *H. pylori* killing at 100 μg/mL (in USS dose). Moreover, the observed *H. pylori* killing depended on the US output power density and on the US exposure time (Fig. 3i). Clearly, all three USS-PLGA@Lecithin nanoparticles exhibited US-triggered killing of *H. pylori* due to their preloaded USS, and in doing so, Ce6-PLGA@Lethicin and Ver-PLGA@Lecithin were similarly efficient, whereas ICG-PLGA@Lecithin was the least efficient (Fig. 3h). Based on the performance of the USS-PLGA@Lecithin nanoparticles in ROS generation and *H. pylori* killing and considering that Ver has been

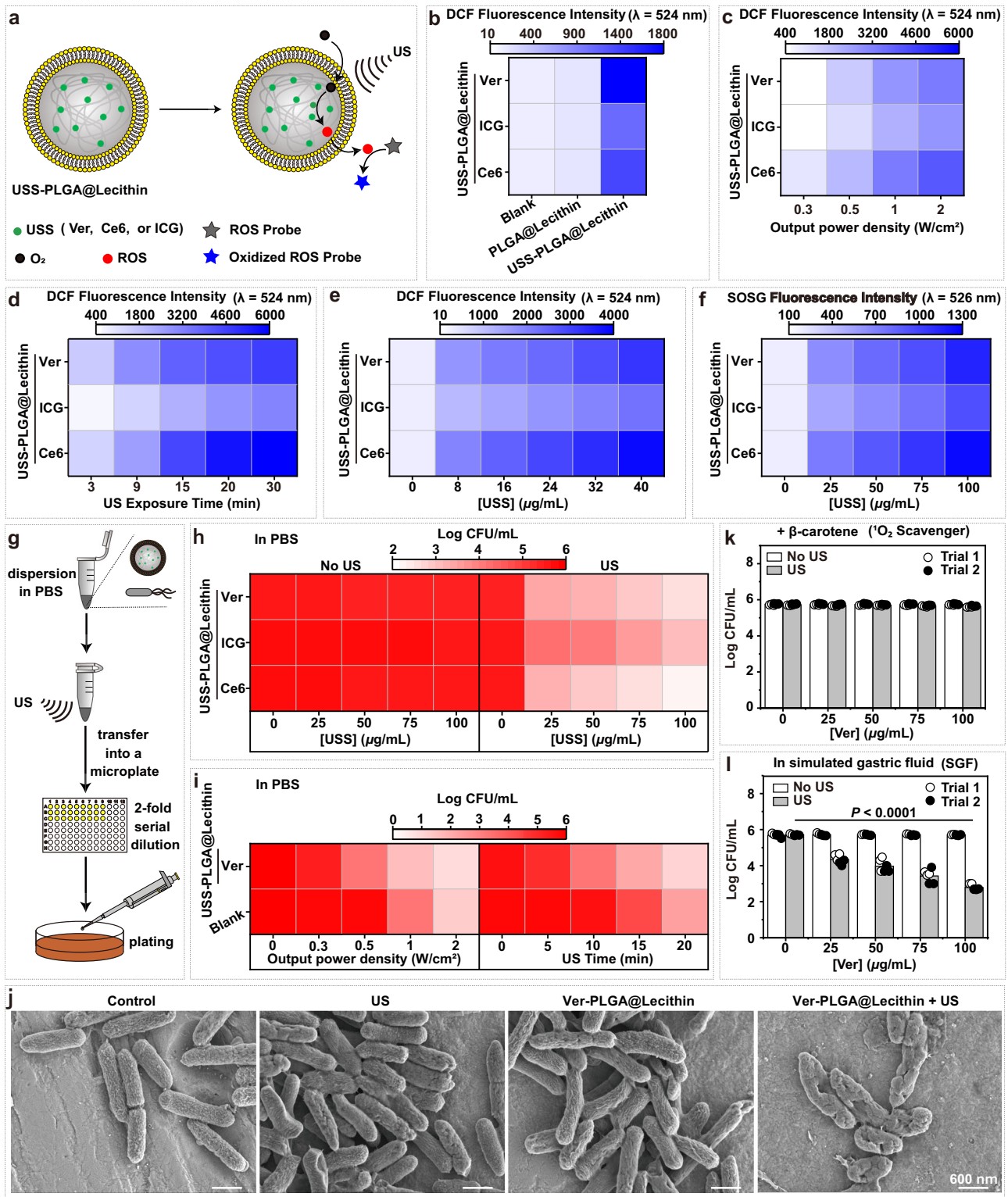

approved for clinical use by the U.S. FDA[35], whereas Ce6 has not, we selected Ver-PLGA@Lecithin as the nanosonosensitizer for the sonodynamic therapy.

The US-triggered *H. pylori* killing by Ver-PLGA@Lecithin in the above quantitative assays was further verified under scanning electron microscopy (SEM) (Fig. 3j), in which *H. pylori* cells treated with both Ver-PLGA@Lecithin and US exposure (i.e., Ver-PLGA@Lecithin + US) appeared collapsed with holes in their cell walls, while those treated with either Ver-PLGA@Lecithin or US alone remained intact, as did their untreated counterparts. To confirm that US-exposed Ver-

PLGA@Lecithin kills *H. pylori* with ${}^1O_2$ generated in situ upon US exposure, we performed similar antibacterial assays but in PBS supplemented with β-carotene, a singlet oxygen scavenger[55], and observed complete inactivation even in the presence of US exposure (Fig. 3k).

The gastric environment is more complex than PBS, the medium used in the above antibacterial assays. To simulate the gastric environment, we performed similar antibacterial assays with Ver-PLGA@Lecithin but in simulated gastric fluid (SGF) and observed similar US-triggered dose-dependent killing of *H. pylori* (Fig. 3l).

**Fig. 3 | In vitro performances of the model nanosonosensitizers. a** Schematic illustration of the detection of reactive oxygen species (ROS) generated by USS-PLGA@Lecithin in response to ultrasound (US) exposure. **b** Fluorescence intensity of DCFH solution with USS-PLGA@Lecithin or its empty counterpart PLGA@Lecithin upon US exposure (at 0.5 W/cm² for 10 min). DCFH solution without any nanoparticles was included as a blank. **c–e** Fluorescence intensity of DCFH solution containing USS-PLGA@Lecithin (**c, d**) at a fixed nanoparticle concentration upon US exposure (**c**) at differing output power densities (for 10 min) and (**d**) for differing exposure times (at 0.5 W/cm²) and (**e**) at differing nanoparticle concentrations (in USS dose) upon US exposure (at 0.5 W/cm² for 10 min). **f.** Fluorescence intensity of SOSG solution containing USS-PLGA@Lecithin at differing concentrations (in USS dose) upon US exposure (at 0.5 W/cm² for 10 min). **g** Schematic illustration of the in vitro antibacterial assays against *H. pylori* using USS-PLGA@Lecithin. **h** Survival *H. pylori* cell counts (in Log CFU/mL, where CFU means colony

forming unit) after treatment with a USS-PLGA@Lecithin at differing concentrations (in USS dose) in PBS (right) with and (left) without US exposure (at 0.5 W/cm² for 10 min). **i** Survival *H. pylori* cell counts (in Log CFU/mL) after treatment with Ver-PLGA@Lecithin in PBS upon US exposure (left) at differing output power density (for 10 min) and (right) for differing exposure time (at 0.5 W/cm²). **j** Scanning electron microscopy (SEM) images of *H. pylori* cells after different treatments. Two times, each experiment was repeated independently with similar results. **k, l** Survival *H. pylori* cell counts (in Log CFU/mL) after treatment with Ver-PLGA@Lecithin at differing concentrations (in Ver dose) (**k**) in β-carotene supplemented PBS and (**l**) in simulated gastric fluid (SGF) in the presence and absence of US exposure (at 0.5 W/cm² for 10 min). Bar height represents the average of two independent trials (*n* = 3 in each independent trial). Statistical analysis was carried out with a two-way ANOVA with Tukey's multiple-comparison test. Source data are provided as a Source Data file.

## Protein corona formation negligibly affects the sonosensitizer property of Ver-PLGA@Lecithin

A protein corona forms rapidly (within 0.5 min[37]) after a nanoparticle is introduced into a protein-rich fluid[37–39]. Indeed, Ver-PLGA@Lecithin (100 µg/mL in Ver dose) preincubated (for 12 h) in HCS-supplemented water (10 mg/mL) exhibited 0.068 mg/mL adsorbed proteins, whereas its counterpart preincubated in water did not (Fig. 4a). Even more adsorbed protein (0.077 mg/mL) was observed for Ver-PLGA@Lecithin preincubated similarly but in HCS-supplemented SGF, a better simulation of the gastric fluids of *H. pylori*-infected patients. Clearly, proteins naturally available in HCS and in SGF readily adsorb onto Ver-PLGA@Lecithin.

To preliminarily test whether protein adsorption on Ver-PLGA@Lecithin affects the nanoparticle's performance as a nanosonosensitizer in its targeted application environment, we examined whether the adsorption of HCS proteins affects the nanoparticle's efficiency in ROS generation and in *H. pylori* killing in response to ultrasound stimulation. Using DCFH as the ROS probe, we found that in response to US exposure, Ver-PLGA@Lecithin recollected (after 12-h preincubation) from HCS-supplemented PBS (10 mg/mL) generated slightly less ROS than its counterpart recollected from PBS (Fig. 4b), despite the significant adsorption of proteins observed on the former but not on the latter. Similar results were observed when HCS-supplemented PBS was replaced with HCS-supplemented SGF (Fig. 4b), a better simulation for the gastric fluids of *H. pylori*-infected patients, although Ver-PLGA@Lecithin recollected from HCS-supplemented SGF (10 mg/mL) exhibited even more adsorbed proteins than its counterpart from HCS-supplemented PBS. Obviously, protein adsorption on Ver-PLGA@Lecithin has a negligible impact on the particle's efficiency in ROS generation.

Moreover, *H. pylori* killing assays in vitro showed that Ver-PLGA@Lecithin recollected (after 12-h preincubation) from HCS-supplemented PBS (10 mg/mL) exhibited US-triggered killing of *H. pylori*, as did its counterpart recollected from PBS (Fig. 4c); under comparable US exposure conditions, Ver-PLGA@Lecithin recollected from HCS-supplemented PBS reduced the viability of inoculated *H. pylori* cells to a similar extent as its counterpart from PBS (2.86 *versus* 3.01, in Log(CFU/mL)) (Fig. 4c and Supplementary Fig. 15d–f), despite the significant protein adsorption observed on the former but not on the latter. Similar results were observed when comparing Ver-PLGA@Lecithin recollected from HCS-supplemented SGF (10 mg/mL) with that recollected from PBS (2.81 *versus* 3.01 in Log(CFU/mL)), although even more adsorbed proteins were observed on Ver-PLGA@Lecithin recollected from HCS-supplemented SGF than that from HCS-supplemented PBS (Fig. 4a). Obviously, protein adsorption on Ver-PLGA@Lecithin has a negligible impact on the US-triggered killing of *H. pylori* by the particles. Taken together, these results suggest a negligible impact of protein adsorption on the performance of Ver-PLGA@Lecithin as a nanosonosensitizer (Fig. 4d) and support the particle's use as a sononanosensitizer in vivo (Fig. 5).

## Efficacy of the sonodynamic therapy mediated by Ver-PLGA@Lecithin for gastric *H. pylori* infection

When administered orally alone, Ver-PLGA@Lecithin exhibited good biocompatibility in healthy female mouse models (Supporting Information, Supplementary Fig. 16) and gut microbiota friendliness in gastric *H. pylori*-infected female mouse models (Supporting Information, Supplementary Figs. 17, 18), which combined with the high efficiency of the particle as a sonosensitizer (Fig. 3), even in a protein-rich environment (Fig. 4), encouraged us to examine the efficacy of sonodynamic therapy mediated by Ver-PLGA@Lecithin in gastric *H. pylori*-infected female mouse models (Fig. 6). As a control for comparison, triple therapy of a commonly used regimen comprising clarithromycin and amoxicillin (two antibiotics) and omeprazole (a PPI) was included (Fig. 6a). Briefly, forty-two C57BL/6 J mice were randomly divided into six groups (*n* = 7 per group), among which one was randomly selected to stay naive throughout the whole study (i.e., the healthy group), while the remaining five received gastric *H. pylori* infection and then treatment with PBS alone (i.e., the control group), triple therapy (i.e., the triple therapy group), US exposure alone (i.e., the US group), Ver-PLGA@Lecithin alone (i.e., the Ver-PLGA@Lecithin group), or Ver-PLGA@Lecithin followed by US exposure (i.e., Ver-PLGA@Lecithin + US, also called the sonodynamic therapy group); all drugs and nanoparticles were dispersed into PBS and administered orally, and US exposure was applied locally on the skin over the stomach. At 48 h after treatment completion, these mice were sacrificed, and their stomachs were collected for homogenization and subsequent quantification of *H. pylori* burden therein (Fig. 6b).

Notably, the average gastric *H. pylori* burden of the sonodynamic therapy group (i.e., Ver-PLGA@Lecithin + US) was comparable to that of the triple therapy group but became approximately 50-fold lower than that of the control group (Fig. 6c). In contrast, the average gastric *H. pylori* burden of the US group and that of the Ver-PLGA@Lecithin group were similar, both of which were comparable to that of the control group, indicating that neither US exposure nor Ver-PLGA@Lecithin alone is effective in reducing the gastric *H. pylori* burden. Clearly, sonodynamic therapy mediated by Ver-PLGA@Lecithin reduced gastric *H. pylori* burden as efficiently as triple therapy, and its observed efficacy is attributable to the US-triggered ROS generation of Ver-PLGA@Lecithin.

Gastric *H. pylori* infection results in higher-than-normal serum levels of certain proinflammatory cytokines, such as interleukin-1 beta (IL-1β), interleukin-6 (IL-6) and tumor necrosis factor-α (TNF-α)[56,57]. Indeed, the *H. pylori*-infected yet untreated mice (i.e., the control group) consistently exhibited >2-fold higher levels of IL-6, TNF-α, and IL-1β than their healthy counterparts (Fig. 6d–f). To examine whether sonodynamic therapy mediated by Ver-PLGA@Lecithin effectively suppresses the inflammatory responses induced by gastric *H. pylori* infection, we compared the serum levels of IL-6, TNF-α, and IL-1β in different treatment groups (Fig. 6d–f). We found that the sonodynamic therapy significantly suppressed the *H. pylori* infection-induced

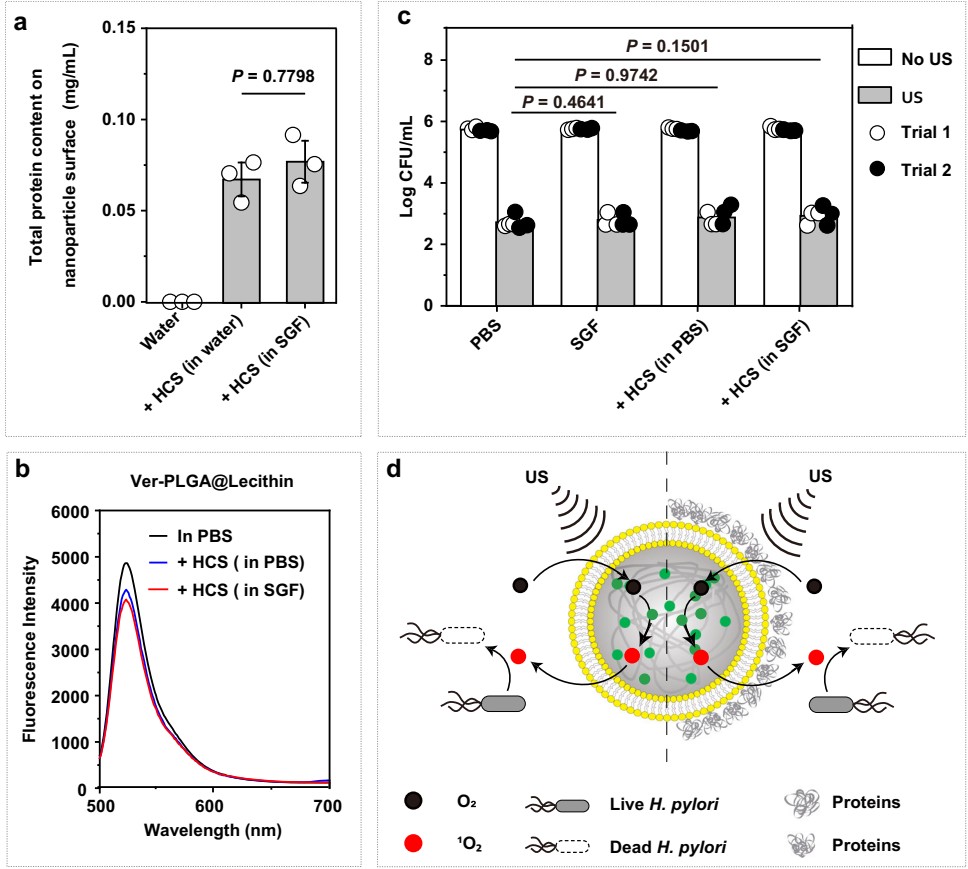

**Fig. 4 | Effects of the protein corona on Ver-PLGA@Lecithin performance in vitro. a** Contents of total adsorbed proteins on Ver-PLGA@Lecithin (100 μg/mL, in Ver dose) recollected via centrifuge after 12-h incubation in HCS-supplemented water or in HCS-supplemented SGF (both at 10 mg/mL), with those after 12-h incubation in water included for comparison. Bar heights are reported as the average ± standard deviation (*n* = 3 in each independent trial). Statistical analysis was carried out with a one-way ANOVA with Tukey's multiple-comparison test. **b** Fluorescence emission spectra of DCFH (40 μM) upon US exposure (0.5 W/cm², 10 min) in the presence of Ver-PLGA@Lecithin (100 μg/mL in Ver dose) recollected via centrifugation after a 12-h incubation in HCS-supplemented PBS or in HCS-supplemented SGF (both at 10 mg/mL), with that after a 12-h incubation in PBS included for comparison. **c** Counts of viable *H. pylori* cells (in Log CFU/mL) upon US exposure (0.5 W/cm², 10 min) after treatment with Ver-PLGA@Lecithin (100 μg/mL in Ver dose) and recollected via centrifugation after a 12-h incubation in HCS-supplemented PBS or in HCS-supplemented SGF (both at 10 mg/mL), with that after a 12-h incubation in PBS included for comparison. Controls are those treated similarly but without US exposure. Bar height represents the average of two independent trials (*n* = 3 in each independent trial). Statistical analysis was carried out with a two-way ANOVA with Tukey's multiple-comparison test. **d** Schematic illustration of the negligible effects of protein corona formation over Ver-PLGA@Lecithin on the efficiency of ROS generation and consequent *H. pylori* killing upon US exposure. Source data are provided as a Source Data file.

upregulation of IL-6, TNF-α, and IL-1β (*P* < 0.0001) to an extent comparable to that provided by triple therapy (*P* > 0.3 for all three cytokines). In contrast, US exposure alone failed to do so (*P* > 0.3 for IL-6 and TNF-α, *P* = 0.0063 for IL-1β). Of note, Ver-PLGA@Lecithin alone significantly suppressed the *H. pylori* infection-induced upregulation of IL-6, TNF-α, and IL-1β (*P* < 0.05) (Fig. 6d–f), possibly owing to its ability to remove VacA (Fig. 2f), which supports our use of a nanoplatform in sonosensitizer construction and underlines the importance of VacA removal in the treatment of gastric *H. pylori* infection. Nevertheless, in doing so, Ver-PLGA@Lecithin alone was apparently less efficient than the sonodynamic therapy, possibly due to its inability to kill *H. pylori* in the absence of US exposure (Fig. 3h, i, l), which underlines the importance of US-triggered ROS generation by Ver-PLGA@Lecithin. Clearly, sonodynamic therapy mediated by Ver-PLGA@Lecithin significantly suppressed the *H. pylori* infection-induced inflammatory responses as effectively as triple therapy, owing to the particle's ability to remove VacA even in the absence of US exposure and to generate ROS upon US exposure.

Pepsinogen I (PG-1), pepsinogen II (PG-2), and gastrin (G-17) are three biomarkers for the state and function of the gastric mucosa[58,59]. To examine whether the sonodynamic therapy mediated by Ver-PLGA@Lecithin helps renormalize gastric function, we monitored the serum levels of PG-1, PG-2, and G-17 at 48 h after treatment completion (Fig. 6g–i). Indeed, all three biomarkers exhibited significantly lower serum levels in the control group than in the healthy group, indicative of significantly damaged gastric function due to *H. pylori* infection. Nevertheless, in the sonodynamic therapy group, serum levels of these three biomarkers were unanimously close to those in the healthy group and consistently comparable to those in the triple therapy group (*P* > 0.9). In contrast, for all three biomarkers, their serum levels in the US group and those in the Ver-PLGA@Lecithin group were comparable to those in the control group (*P* > 0.6). Collectively, these results suggest that the sonodynamic therapy mediated by Ver-PLGA@Lecithin offered similar effective renormalization of gastric function as did triple therapy, owing to the US-triggered ROS generation of Ver-PLGA@Lecithin.

To examine how the sonodynamic therapy mediated by Ver-PLGA@Lecithin affects stomach tissue, we collected female mouse stomachs at 48 h after treatment completion and stained slices of the as-collected stomach tissues with hematoxylin-eosin (H&E) or TUNEL (TdT-mediated dUTP nick-end labelling) for examination under microscopy. Images of H&E-stained tissues revealed that the stomach tissues from mice of the sonodynamic therapy group appeared similar to those from their healthy counterparts (i.e., the healthy group)

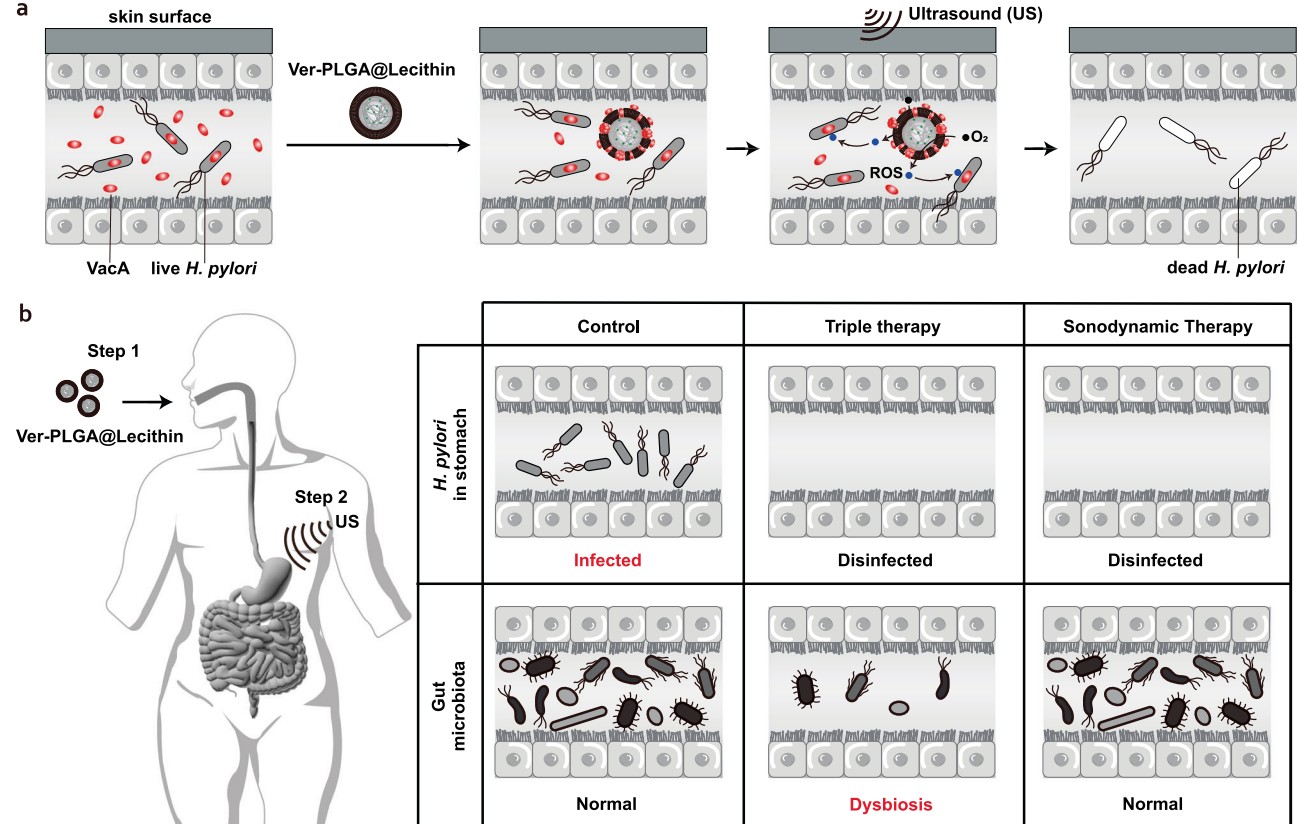

**Fig. 5 | Sonodynamic therapy for gastric *H. pylori* infection. a** Schematic illustration of the sonodynamic therapy in which a lipid bilayer-coated nanosonosensitizer removes VacA even in the absence of ultrasound (US) exposure and generates reactive oxygen species (ROS) for *H. pylori* killing upon US exposure. **b** Schematic illustration of the superiority of the sonodynamic therapy to the current standard clinical treatments for gastric *H. pylori* infection, such as triple therapy.

(Fig. 6j). Meanwhile, images of TUNEL-stained tissues (Supplementary Fig. 19, Supplementary Information) demonstrated that although gastric *H. pylori* infection significantly increased the relative ratio of TUNEL-positive (i.e., apoptotic) cells ($P < 0.0001$), the sonodynamic therapy mediated by Ver-PLGA@Lecithin effectively reduced this ratio back to normal ($P > 0.3$); in stark contrast, triple therapy further increased, rather than decreased, cell apoptosis in the *H. pylori*-infected stomach. Taken together, these results indicate that the sonodynamic therapy mediated by Ver-PLGA@Lecithin, though effective in treating gastric *H. pylori* infection, failed to cause detectable damage to the gastric mucosa or cells.

### Gut microbiota after the sonodynamic therapy mediated by Ver-PLGA@Lecithin

To evaluate whether the sonodynamic therapy mediated by Ver-PLGA@Lecithin is non-disruptive to host gut microbiota, we collected female mouse feces at 48 h after treatment completion for 16 S rRNA gene sequencing (Fig. 7a), with the standard triple therapy included as a representative of antibiotic-based *H. pylori* eradication therapies for comparison and the healthy mice and the *H. pylori*-infected yet untreated counterparts included as references (Fig. 7 and Supplementary Fig. 20). Though *H. pylori* infection alone exerts negligible effects on mouse gut microbiota (Supplementary Information, Supplementary Data 2), the sonodynamic therapy mediated by Ver-PLGA@Lecithin and the standard triple therapy diverged significantly on their effects on mouse gut microbiota. Analysis of the alpha (α) diversity of the gut microbiota revealed that the sonodynamic therapy resulted in apparently higher α diversity of the gut microbiota than triple therapy, which exhibited apparently lower α diversity than both the control group and the healthy group (Fig. 7b and Supplementary Fig. 20b).

Principal component analysis (PCA) on beta (β) diversity of gut microbiota shows that the sonodynamic therapy group was located in the close vicinity of both the healthy group and the control group, whereas the triple therapy group was located far away from them (Fig. 7c and Supplementary Fig. 20c). Collectively, these results suggest significantly lower disturbance to the diversity of the gut microbiota after sonodynamic therapy than after triple therapy.

To unveil how the sonodynamic therapy mediated by Ver-PLGA@Lecithin affects gut microbiota composition, we further quantified the relative abundances of gut commensal bacteria. At both the phylum and genus levels, the control group exhibited similar patterns of relative abundance for gut commensal bacteria as the healthy group. To the naked eye, the sonodynamic therapy group exhibited similar patterns as the healthy group except that *Lactobacillus*, a bacterium widely used in yogurt products and probiotics, was obviously upregulated (Fig. 7d, e and Supplementary Fig. 20d, e); in contrast, the triple therapy group exhibited significant upregulation of *Proteobacteria* at the phylum level (Fig. 7d and Supplementary Fig. 20d) and of *Enterococcus* and *Bacteroides* at the genus level (Fig. 7e and Supplementary Fig. 20e).

To identify the bacterial species with significantly changed relative abundances in mouse gut microbiota after the sonodynamic therapy, we analysed the fold of change, a widely used index in such analysis[60,61], in bacterial relative abundance after the sonodynamic therapy relative to the healthy group (i.e., log$_2$(Sonodynamic Therapy / Healthy)) (Fig. 7f) and that relative to the control group (i.e., log$_2$(Sonodynamic Therapy / Control)) (Fig. 7h). For a bacterial species to marked as a significantly up- or downregulated one, it needs to simultaneously have the *P* value (Student's t test) associated with its difference in relative abundance of <0.05 and a fold of

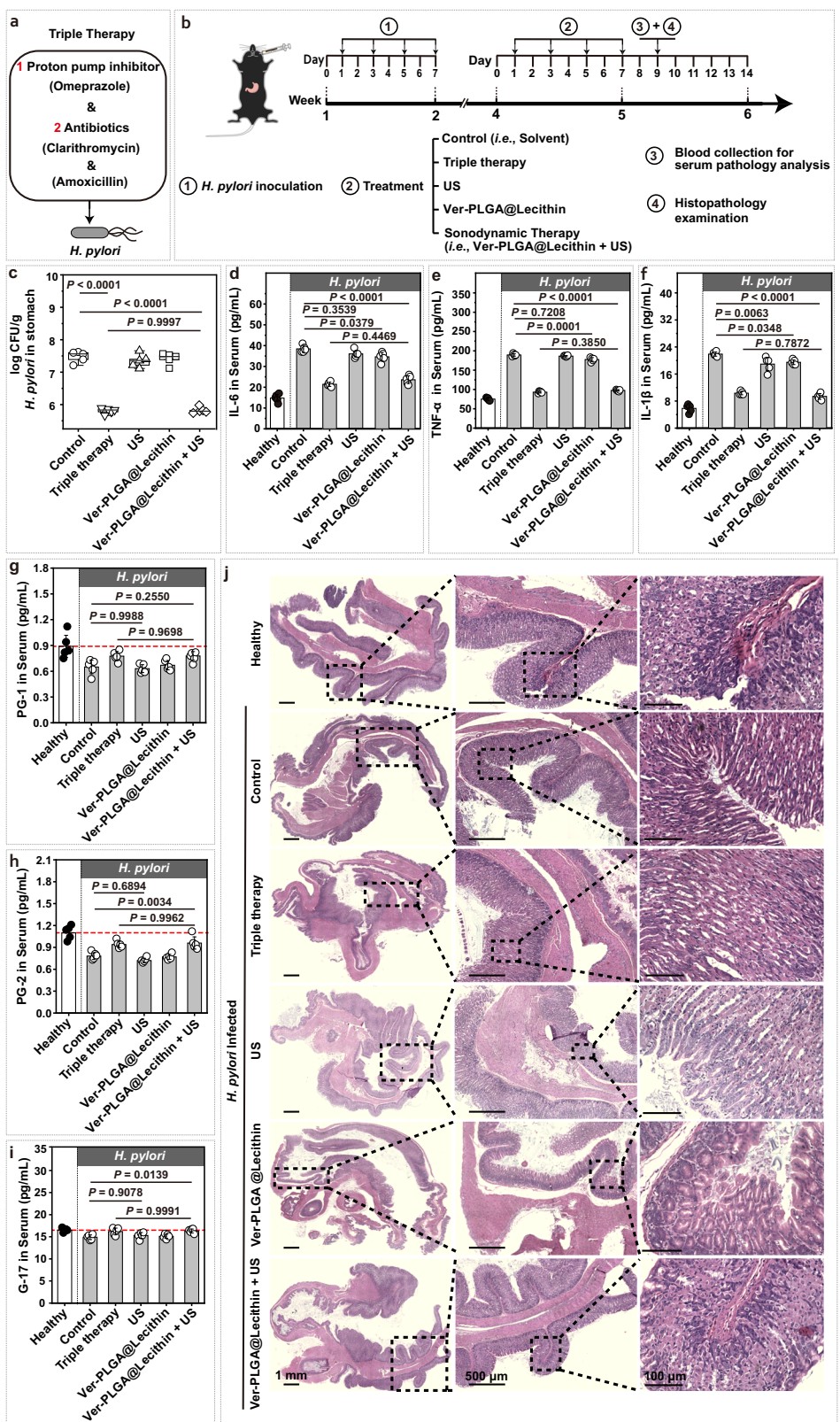

change in bacterial relative abundance of ≥2 or ≤1/2. Compared to the healthy group, the sonodynamic therapy resulted in only one significant change to mouse gut microbiota composition (Fig. 7f), which is the upregulation of *Lactobacillus* at the genus level. Compared to the control group, the sonodynamic therapy significantly changed the relative abundances of two commensal bacterial species at the genus level but none at the phylum level (Fig. 7h); specifically, it significantly upregulated *Lactobacillus* while significantly downregulated *Turicibacter*. Similar analysis was performed for triple therapy, using the fold of change in bacterial relative abundance after triple therapy relative to the healthy group (i.e., log₂(Triple Therapy / Healthy)) (Fig. 7g) and that relative to the control group (i.e., log₂(Triple Therapy / Control)) (Fig. 7i). Compared to the healthy group, triple therapy (Fig. 7g) significantly

**Fig. 6 | Efficacy of sonodynamic therapy mediated by Ver-PLGA@Lecithin.**
**a** Schematic illustrations on the triple therapy regimen used for treating gastric *H. pylori* infection. **b** Schematic illustration on the schedule of different treatments including sonodynamic therapy (i.e., "Ver-PLGA@Lecithin + US") to mouse models bearing gastric *H. pylori* infection and follow-up blood and tissue collection. **c** *H. pylori* burden per gram of stomach tissues (in Log CFU/g) from mice treated with "Ver-PLGA@Lecithin + US" group, with those from mice treated with PBS (i.e., control), triple therapy, US exposure alone, and Ver-PLGA@Lecithin alone included for comparison. The boxes denote the lower 25% quantile, upper 75% quantile, and centerline the median, with whiskers extending to a limit of ±1.5 interquartile ranges (IQRs) (*n* = 5 biologically independent mice in one trial). Statistical analysis was carried out with a one-way ANOVA with Tukey's multiple-comparison test.
**d–f** Serum levels of (**d**) interleukin-6 (IL-6), (**e**) tumor necrosis factor-α (TNF-α), and (**f**) interleukin-1 beta (IL-1β) in gastric *H. pylori* infection-bearing mouse models after differing treatments. Bar heights are reported as the average ± standard deviation (*n* = 5 biologically independent mice in one trial). Statistical analysis was carried out with a one-way ANOVA with Tukey's multiple-comparison test. **g–i** Serum levels of (**g**) pepsinogen I (PG-1), (**h**) pepsinogen II (PG-2) and (**i**) gastrin-17 (G-17) in gastric *H. pylori* infection-bearing mouse models at 48 h after differing treatments. Bar heights are reported as the average ± standard deviation (*n* = 5 biologically independent mice in one trial). The red dashed line indicates the average serum level in healthy mice. Statistical analysis was carried out with a one-way ANOVA with Tukey's multiple-comparison test. **j** Microscopy images of hematoxylin and eosin (H&E)-stained stomach tissues collected from gastric *H. pylori* infection-bearing mouse models at 48 h after differing treatment, with those of H&E-stained stomach tissues collected from healthy counterparts included as references. Source data are provided as a Source Data file.

downregulated one bacterial species at the phylum level (specifically, it is *Actinobacteria*) and nine bacterial species at the genus level (specifically, they are *Lactobacillus*, *Turicibacter*, *Clostridium_sensu_stricto*, *Parabacteroides*, *Romboutsia*, *Bifidobacterium*, unclassified_*Porphyromonadaceae*, unclassified_*clostridiales*, unclassified_*Erysipelotrichaceae*) while significantly upregulated one bacterial species at the phylum level (specifically, it is *Proteobacteria*) and five bacterial species at the genus level (specifically, they are *Enterococcus*, *Clostridium_XlVa*, *Bacteroides* and *Escherichia_Shigella*, unclassified_*Lachnospiraceae*). Same results were observed after triple therapy in reference to the control group (Fig. 7i), except for the significant upregulation of one additional bacterial specis (specifically, it is *Parasutterella* at the genus level). Obviously, unlike triple therapy, the sonodynamic therapy caused negligible perturbation to mouse gut microbiota composition.

Note that, after triple therapy, some gut commensal bacterial species, though having fold of change in bacterial relative abundance ranging from 1/2 to 2, exhibited an absolute difference in relative abundance of ≥0.1% from their counterparts either in the healthy group or in the control group. Hence, to exclude possible underestimation on the sonodynamic therapy's perturbation to mouse gut microbiota composition, we performed similar analysis but with the absolute change in bacterial relative abundance (i.e., Relative Abundance (Sonodynamic Therapy – Healthy, %), and Relative Abundance (Sonodynamic Therapy – Control, %)) (Supplementary Fig. 21a, b), with triple therapy included as a representative for antibiotic-based *H. pylori* eradication therapies (Supplementary Fig. 21c, d). In the resulting plots of -$\mathrm{Log}_{10}$ (*P* value) *versus* the absolute change in bacterial relative abundance (Supplementary Fig. 21), a bacterial species was marked as a significantly up- or downregulated one as long as the *P* value for its difference in relative abundance was <0.05, which may overestimate the perturbation to gut microbiota composition. Still, after the sonodynamic therapy, the sole significant impact to mouse gut microbiota composition observed consistently in both trials 1 and 2 is the upregulation of *Lactobacillus* and this is the case no matter whether the reference is the healthy group or the control group (Supplementary Information), consistent with the observations above based on the fold of change in bacterial relative abundance (Fig. 7f–i).

Inspiringly, no matter whether the change in bacterial relative abundance is gauged with the fold of change or with the absolute change, *Lactobacillus*, a bacterium widely used in yogurt products and probiotics, is consistently observed to be significantly downregulated after triple therapy but significantly upregulated after the sonodynamic therapy, suggesting that the sonodynamic therapy, rather than triple therapy, may help promote the mouse gut microbiota health. Of note, the downregulation of *Actinobacteria* consistently observed after triple therapy here is consistent with that observed at 6 months after triple therapy in human objects[62]. Collectively, these results indicate that the sonodynamic therapy mediated by Ver-PLGA@Lecithin, but not triple therapy, is non-disruptive to host gut microbiota.

## The sonodynamic therapy mediated by Ver-PLGA@Lecithin lacks toxicity to mouse models

Toxicity to the host is a major concern when developing a therapeutic agent or modality. To preliminarily examine whether the sonodynamic therapy mediated by Ver-PLGA@Lecithin is biosafe, we carried out liver and renal function tests and complete blood count (CBC) tests at 48 h after treatment completion in gastric *H. pylori* infection-bearing female mouse models (Fig. 8a). Lower-than-normal serum levels of albumin and higher-than-normal serum levels of alanine aminotransferase (ALT) and aminotransferase (AST) indicate liver damage or dysfunction. In the sonodynamic therapy group, neither the serum level of albumin was downregulated (Supplementary Fig. 22a) nor those of ALT and AST were upregulated compared with the healthy group (Fig. 8b, c), indicating a lack of liver damage after sonodynamic therapy. Higher-than-normal serum levels of blood urea nitrogen (BUN), creatinine (CREA), and uric acid (UA) may indicate renal damage or dysfunction. In the sonodynamic therapy group, the serum levels of BUN, CREA, and UA were unanimously lower than those in the healthy group (Fig. 8d, e and Supplementary Fig. 22b), indicating a lack of kidney damage or dysfunction after the sonodynamic therapy.

A CBC test preliminarily examines the overall health of a host (Fig. 8f–m and Supplementary Fig. 22c–l). In the sonodynamic therapy group, all eight indicators for the red blood cell (RBC) state (Fig. 8f, g and Supplementary Fig. 22c–h) and 3 out of the 5 examined white blood cell (WBC) indicators (i.e., WBC count and blood levels of eosinophils and lymphocytes) (Fig. 8h–j) were comparable to their counterparts in the healthy group, indicating that RBCs, WBCs overall, eosinophils, and lymphocytes are healthy after the sonodynamic therapy. Meanwhile, in the sonodynamic therapy group, blood levels of neutrophils and monocytes (2 other WBC indicators) (Fig. 8k, l) and all five platelet indicators (Fig. 8m and Supplementary Fig. 22i–l), though lower than those in the healthy group, were consistently comparable to their counterparts in the triple therapy group, indicating that monocytes, neutrophils and platelets after the sonodynamic therapy are similarly healthy as those after triple therapy. Collectively, these results suggest excellent overall health for female mouse models after the sonodynamic therapy mediated by Ver-PLGA@Lecithin.

Endogenous interleukin-1 receptor antagonist (IL-1RA) is a natural anti-inflammatory protein important in diseases including arthritis, colitis, and chronic renal failure[63]. For example, elevated plasma levels of IL-1RA have been found in acutely ill patients after surgery or with sepsis and in patients with chronic renal failure[63]. Therefore, to compare the relatively long-term effects on host health after the sonodynamic therapy mediated by Ver-PLGA@Lecithin with those after triple therapy, we monitored the serum levels of IL-1RA for groups after different treatments over an observation window of 12 weeks (Fig. 9a). We found that throughout the 12-week observation window, the control group consistently exhibited similar serum levels of IL-1RA as the

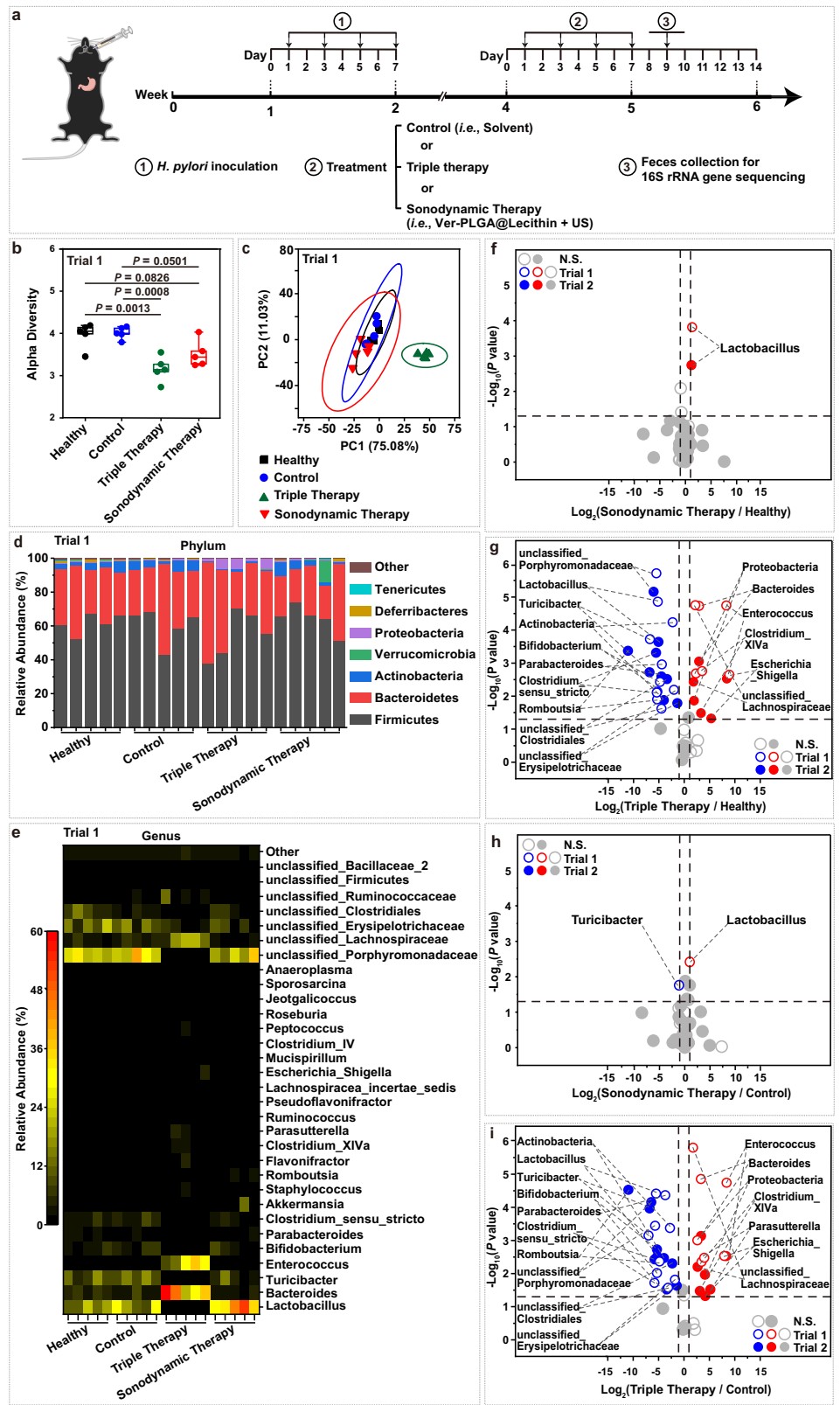

healthy group (*P* > 0.9) (Fig. 9b–e), indicative of negligible effects of gastric *H. pylori* infection alone on the host serum level of IL-1RA. Interestingly, the sonodynamic therapy group consistently exhibited similar serum levels of IL-1RA as the healthy group (*P* > 0.05) (Fig. 9b–e), whereas the triple therapy group consistently exhibited significantly higher serum levels of IL-1RA than the healthy group (*P* < 0.001) (Fig. 9b–e), suggesting that certain relatively long-term

inflammatory responses may have been induced after triple therapy but not after sonodynamic therapy. Unfortunately, at the current stage, we do not know what specific disease or dysfunction the observed elevation in IL-1RA serum level may indicate. Nevertheless, its presence in the triple therapy group but not in the sonodynamic therapy group indicates superiority in the biosafety of the latter to the former.

**Fig. 7 | Effects of sonodynamic therapy on gut microbiota. a** Schematic illustration on the schedule of differing treatment to mouse models bearing gastric *H. pylori* infection and follow-up feces collection for (b-e) gut microbiota analysis. **b**, **c** The (**b**) α diversity and (**c**) β diversity of gut microbiota by principal component analysis (PCA) in mouse models after differing treatment, with those of the healthy group included for comparison. The boxes denote the lower 25% quantile, upper 75% quantile, and centerline the median, with whiskers extending to a limit of ±1.5 interquartile ranges (IQRs) (*n* = 5 biologically independent mice in one trial). Statistical analysis was carried out with a one-way ANOVA with Tukey's multiple-comparison test. **d**, **e** The relative abundances of gut bacterial species at the (**d**) phylum and (**e**) genus levels in mouse models after differing treatments, with those of the healthy group included for comparison. The data were clustered according to the number of samples per group on the abscissa. **f**–**i** Statistical analysis of how (**f**, **h**) sonodynamic therapy and (**g**, **i**) triple therapy affect the composition of gut

commensal bacteria, using (**f**, **g**) the healthy group and (**h**, **i**) the control group as references. The change in the relative abundance of gut bacterial species after sonodynamic therapy is indicated both as (**f**) the fold change compared to the healthy group (i.e., $\log_2$(Sonodynamic therapy/Healthy)) and (**h**) that compared to the control group (i.e., $\log_2$(Sonodynamic therapy/Control)). Similarly, the change in the relative abundance of gut bacterial species after triple therapy is indicated both as (**g**) the fold change compared to the healthy group (i.e., $\log_2$(Triple therapy/Healthy)) and (**i**) that compared to the control group (i.e., $\log_2$(Triple therapy/Control)). For a bacterium to be marked as a significantly perturbed bacterium, its fold change in bacterial relative abundance needs to be ≥2 or ≤1/2, and its *P* value needs to be <0.05. Bacterial species whose relative abundances were significantly up- and downregulated after sonodynamic therapy are marked in red and blue, respectively. Statistical analysis was carried out with a two-sided Student's *t* test. Source data are provided as a Source Data file.

To summarize, we have demonstrated the superiority of sonodynamic therapy mediated by Ver-PLGA@Lecithin in conjunction with localized US exposure (0.5 W/cm², 10 min) compared to the conventional antibiotic-based therapies for *H. pylori* infection. Notably, the US dosage used in this sonodynamic therapy falls within the safety limits specified for ultrasound medical devices (at ≤0.72 W/cm² [32,33] and for ≤15 min[32]) and all components of Ver-PLGA@Lecithin have been approved for clinical uses by the U.S. FDA. In reducing *H. pylori* infection, Ver-PLGA@Lecithin exhibits dual functionality: it removes VacA, a key virulence factor secreted by *H. pylori*, even in the absence of US exposure and efficiently generates ROS to kill *H. pylori* upon US exposure. Importantly, the adsorption of proteins naturally present in gastric fluids or secreted by *H. pylori* onto Ver-PLGA@Lecithin does not compromise the particle's efficacy in ROS production or *H. pylori* elimination. In female mouse models bearing gastric *H. pylori* infection, the sonodynamic therapy using Ver-PLGA@Lecithin and US exposure (0.5 W/cm², 10 min) applied to the skin overlying the stomach eradicated gastric *H. pylori* as effectively as the standard triple therapy. Remarkably, unlike triple therapy, this sonodynamic therapy had minimal impact on the α or β diversities of mouse gut microbiota and left the upregulation of *Lactobacillus*, a bacterium widely used in yogurt products and probiotics, as the only significant perturbation to mouse gut microbiota composition. Additionally, in female mouse models, this sonodynamic therapy mediated by Ver-PLGA@Lecithin did not adversely affect their liver or kidney functions or their overall health at 48 h after its implication, aligning with the safety profile of standard triple therapy. Surprisingly, a significant difference emerged in IL-1RA levels, a key anti-inflammatory protein in various diseases, over a 12-week post-treatment observation: while triple therapy upregulated IL-1RA levels, sonodynamic therapy did not. Our findings suggest the sonodynamic therapy using Ver-PLGA@Lecithin as an effective alternative to antibiotic-based treatments for *H. pylori* infection, with added benefits such as VacA removal, preservation of gut microbiota diversity, and potential avoidance of antimicrobial resistance.

## Methods
### Materials
Poly(lactic-co-glycolic) acid (PLGA) (molar ratio of lactic acid to glycolic acid = 75: 25, Mn ~40,000 Da) was purchased from Polymtek Biomaterial (Shenzhen, China). Poly(lactic-co-glycolic) (75:25)-b-poly(ethylene glycol) ($PLGA_{40k}$-$PEG_{2k}$) (PLGA, average Mn ~40,000 Da; PEG, average Mn ~2000 Da) was purchased from Xi'an Ruixi Biological Technology (Shanxi, China). Verteporfin (Ver) was purchased from Shanghai Aladdin Bio-Chem Technology Co., Ltd, China. Chlorin e6 (Ce6) and indocyanine green (ICG) were obtained from Frontier Scientific, Inc. (USA). Lecithin (a mixture of phospholipids from soybean) was purchased from Shanghai Macklin Biochemical Co., Ltd. (Shanghai, China). 2′,7′-Dichlorodihydrofluorescein diacetate (DCFH-DA) and pure phthalic acid (PTA) were purchased from Shanghai Yuanye Bio-

Technology (Shanghai, China). 2,2,6,6-Tetramethylpiperidinooxy (TEMP) and β-carotene were purchased from TCI Development Co., Ltd. Dihydroethidium (DHE) was purchased from Sigma–Aldrich, Shanghai. Phosphate-buffered saline (PBS) (1.35 M NaCl, 47 mM KCl, 100 mM $Na_2HPO_4$, 20 mM $NaH_2PO_4$, pH = 7.4), Bradford Protein Assay Kit and cell counting kit-8 (CCK-8) were purchased from Beyotime Biotechnology (Shanghai, China). Singlet oxygen sensor green (SOSG) was purchased from Thermo Fisher (Shanghai, China). Columbia agar broth powder and tryptic soy broth (TSB) were purchased from Qingdao Hope Bio-Technology (Qingdao, China). Fetal bovine serum (FBS) was purchased from Shanghai ExCell Biology. Inc. Sterile defibrinated sheep blood was obtained from Bianjian Biotechnology Co., Ltd. (Nanjing, China). Simulated gastric fluid (SGF) was purchased from Shanghai Yuanye Bio-Technology (Shanghai, China). For interleukin-1 beta (IL-1β), interleukin 6 (IL-6), tumor necrosis factor-α (TNF-α), creatinine (CREA), uric acid (UA) and blood urea nitrogen (BUN), enzyme-linked immunosorbent assay (ELISA) kits were purchased from Beyotime Biotechnology (Shanghai, China). For interleukin 1-receptor antagonist (IL1-RA), the ELISA kit was purchased from Cloud-Clone Corp. Wuhan Co., Ltd. (Wuhan, China). For pepsinogen I (PG-1), pepsinogen II (PG-2) and gastrin-17 (G-17), ELISA kits were purchased from Shanghai Enzyme-linked Biotechnology Co., Ltd. (Shanghai, China). For albumin (ALB), alanine aminotransferase (ALT) and aspartate aminotransferase (AST), ELISA kits were purchased from Cloud-Clone Corp. Wuhan Co., Ltd. (Wuhan, China). Ham's F-12K medium (PM150910) was purchased from Procell Life Science & Technology Co., Ltd. (Wuhan, China). The *H. pylori* strain (ATCC no. 43504) used in this work was purchased from the American Type Culture Collection (ATCC) (Virginia, USA). Human gastric adenocarcinoma cells (AGS cells) were kindly provided by Cell Bank, Chinese Academy of Sciences. All other reagents were purchased from Sinopharm Chemical Reagent Company (Shanghai, China). All reagents were used as received without further purification unless specified otherwise.

### Quantification of VacA content with an ELISA kit
The HCS was treated with a nanoparticle (PLGA@Lecithin or PLGA@PEG) to remove VacA from the supernatant. Briefly, a nanoparticle dispersion (2 mg/mL in PBS) was added to the acid-activated HCS (10 mg/mL in PBS, pH = 4), and the resulting mixture was subsequently incubated at 4 °C for 24 h and then centrifuged (10,000 g, at 4 °C, for 10 min) (5417 R, Eppendorf) to separate the nanoparticle and the supernatant; a similar procedure was used when the HCS at 20 mg/mL (in PBS, pH = 4) was treated with a nanoparticle dispersion at 8 mg/mL (in PBS). The resulting pellet was then dispersed into solubilization buffer (4 mL) that was provided in a Protein Extraction Kit (Thermo Science Mem-PER Plus Membrane Protein Extraction Kit, China) and cooled to 4 °C prior to use, and the resultant mixture was left alone for 30 min and then centrifuged (10,000 g, at 4 °C, for 15 min), which yielded the extracted VacA in the resulting supernatant.

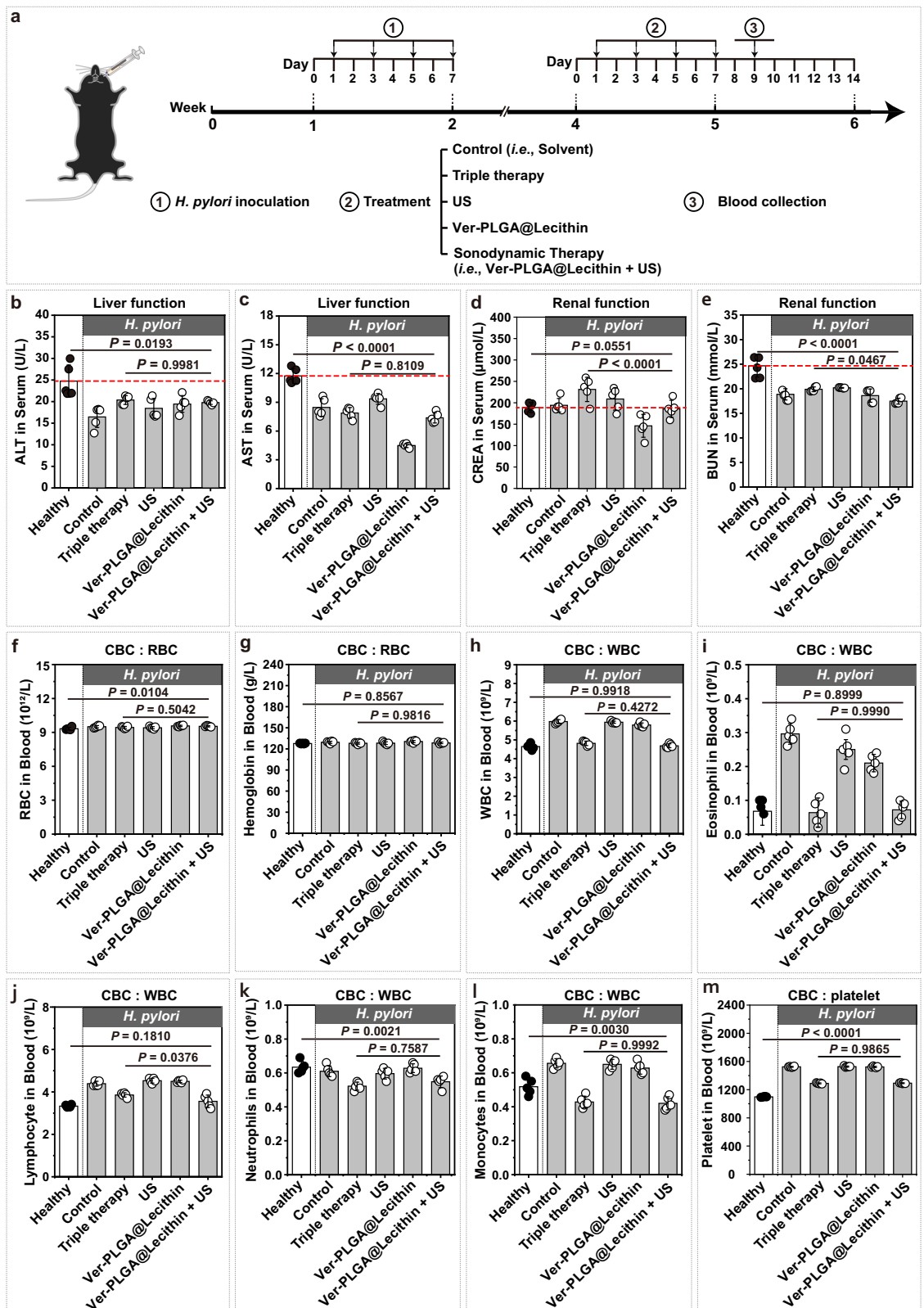

**Fig. 8 | Biosafety of the sonodynamic therapy mediated by Ver-PLGA@Lecithin. a** Schematic illustration on the schedule of differing treatment to mouse models bearing gastric *H. pylori* infection and follow-up blood collection for liver and renal function tests and complete blood count (CBC) test. **b**–**e** Serum levels of (**b**) alanine aminotransferase (ALT), (**c**) aspartate aminotransferase (AST), (**d**) creatinine (CREA), and (**e**) blood urea nitrogen (BUN). **f**–**m**. Blood levels of (**f**) red blood cell count (RBC), (**g**) hemoglobin, (**h**) white blood cell count (WBC), (**i**) eosinophil, (**j**) lymphocyte, (**k**) neutrophils, (**l**) monocyte, and (**m**) platelet. The red dashed line indicates the average serum level in healthy mice. Bar heights are reported as the average ± standard deviation (*n* = 5 biologically independent mice in one trial). Statistical analysis was carried out with a one-way ANOVA with Tukey's multiple-comparison test. Source data are provided as a Source Data file.

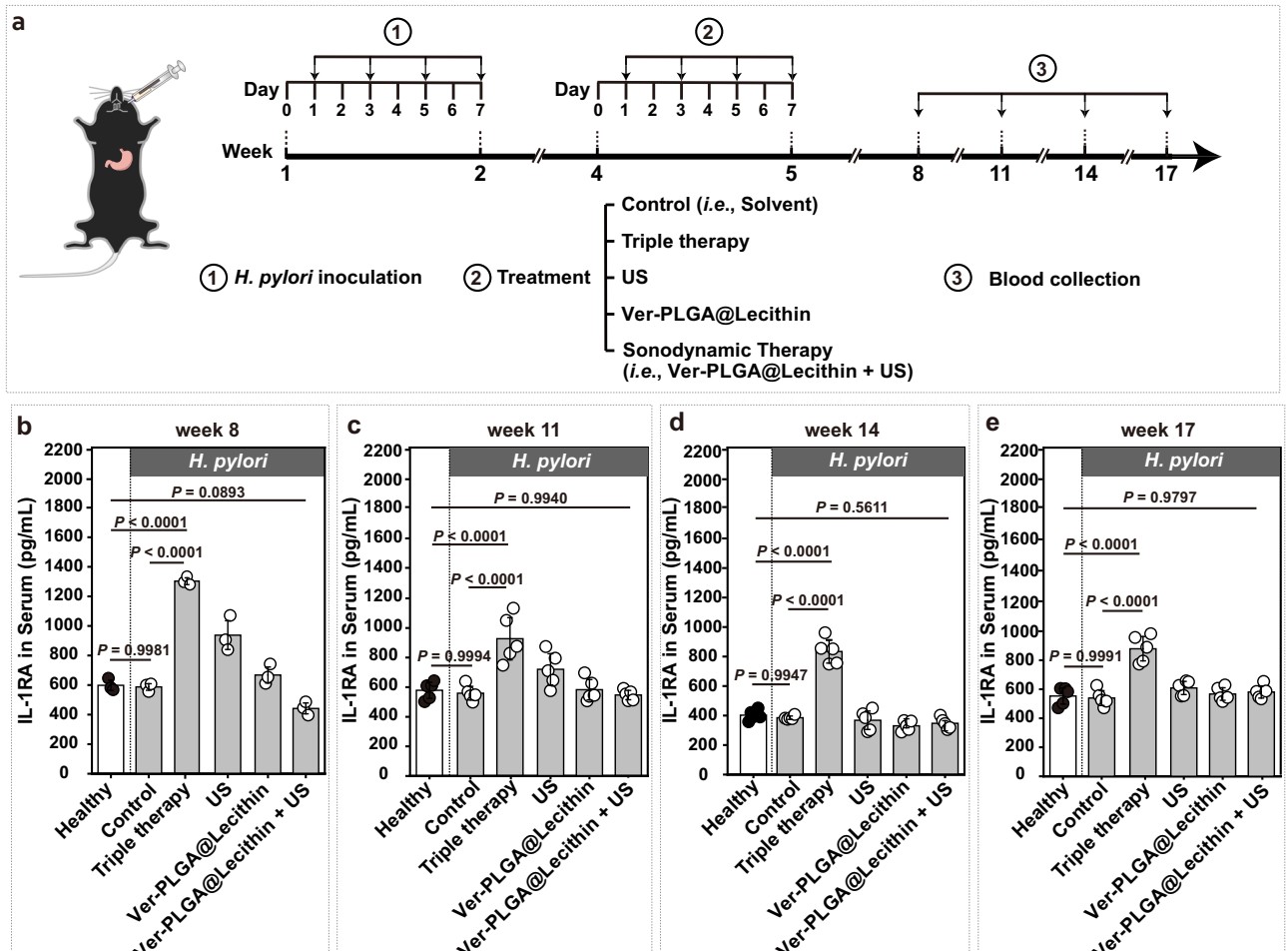

**Fig. 9 | Effects of different treatments on the serum level of IL-1RA. a** Schematic illustration of the schedule of different treatments, including triple therapy and sonodynamic therapy (i.e., "Ver-PLGA@Lecithin + US"), to mouse models bearing gastric *H. pylori* infection and follow-up blood sample collection for monitoring (**b**–**e**) serum levels of IL-1RA. **b**–**e** Serum levels of IL-1RA in mouse models at different time points after treatment completion. Bar heights are reported as the average ± standard deviation ($n = 5$ biologically independent mice in one trial). Statistical analysis was carried out with a one-way ANOVA with Tukey's multiple-comparison test. Source data are provided as a Source Data file.

An ELISA kit (catalog no. ml025361, Shanghai Enzyme-linked Biotechnology Co., Ltd., China) was used to determine the content of residual VacA in the supernatant and that on the nanoparticle surface after coincubation of nanoparticles with HCSs. Briefly, a VacA-containing sample (the nanoparticle-treated HCS, or the supernatant containing the extracted VacA) (100 μL, in PBS) was added to a well of a 96-well microplate precoated with VacA antibody (provided in the kit), followed by the addition of horseradish peroxidase-labelled detection antibody (provided in the kit) (100 μL), incubation at 37 °C for 1 h, washing with wash buffer (provided in the kit) 5 times, addition of the substrate solution (provided in the kit) (90 μL), incubation at 37 °C for 15 min, and then addition of the stop solution (provided in the kit) to terminate the reaction. The resulting microplate was monitored with a microplate reader (Varioskan, Thermo Scientific) to obtain the optical density reading at 540 nm ($OD_{540}$). The calibration curve of $OD_{540}$ *versus* VacA concentration was determined using the VacA standard solutions provided in the kit by the vendor. Each trial was performed in triplicate, and the reported results are averages of two independent trials.

### Treating gastric *H. pylori* infection in female mouse models with sonodynamic therapy

To evaluate the in vivo efficacy of the Ver-PLGA@Lecithin-mediated sonodynamic therapy, we established female mouse models bearing gastric infection with *H. pylori* and monitored the efficacy of the sonodynamic therapy, with Ver-PLGA@Lecithin administered alone and US exposure alone included for comparison. The mouse models used in this work were of one sex, which is female. All animal experiments were conducted in compliance with the guidelines for the care and use of research animals established by the Animal Care and Use Committee at the University of Science and Technology of China (USTCACUC1501010). Mice were housed at a temperature of 22–25 °C and a 12 h/12 h dark/light cycle. The animal studies involved in this work did not take into account the effect of sex, as sex-based analysis is commonly not involved in laboratory animal studies. Briefly, forty-two C57BL/6 J female mice (8 weeks old, Shanghai Slack Laboratory Animal Technology Co., Ltd.) were randomly divided into 6 groups ($n = 7$ per group), with one group to stay uninfected throughout the whole assay (i.e., the healthy group) while the remaining five groups were infected via gavage administration of *H. pylori* containing FBS-supplemented TSB (0.3 mL of ~1.0 × 10⁸ CFU/mL for each female mouse) every 48 h 4 times (on days 1, 3, 5, and 7, respectively) and then left alone for 2 weeks to allow *H. pylori* infection to be established. Subsequently, the healthy group was treated with PBS (i.e., healthy), and the five infected groups were treated with PBS (i.e., control), triple therapy, Ver-PLGA@Lecithin alone, US exposure alone, and Ver-PLGA@Lecithin plus US. For both the healthy group and the control group, each female mouse was treated with PBS (0.3 mL each time) via oral administration

every 48 h and 4 times (gavage on days 1, 3, 5, and 7). For the triple therapy group, each female mouse was treated via oral administration with a triple therapy regimen every 48 h 4 times (gavage on days 1, 3, 5, and 7, respectively), and at each time, the regimen was administered omeprazole (a proton pump inhibitor, Macklin Biochemical Co., Ltd.) (400 µmol per kilogram of female mouse weight) and, 30 min later, with amoxicillin (28.5 mg/kg, Sigma–Aldrich) and clarithromycin (14.3 mg/kg, Sigma–Aldrich). For the Ver-PLGA@Lecithin group, each female mouse was treated via oral administration of Ver-PLGA@Lecithin (0.3 mL of 10 mg/mL in PBS each time) every 48 h for 4 times (gavage on days 1, 3, 5, and 7). For the US group, each female mouse was treated with a combination of PBS and US exposure every 48 h 4 times (gavage on days 1, 3, 5, and 7, respectively), and at each time, the treatment was performed with PBS via oral administration (0.3 mL each time) and, 30 min later, with US exposure (0.5 W/cm$^2$ for 10 min) with a medical US instrument (WED-100, WELLD) on the skin over the stomach. For the sonodynamic therapy group, each female mouse was treated with a combination of Ver-PLGA@Lecithin and US exposure every 48 h 4 times (gavage on days 1, 3, 5, and 7) and, at each time, the treatment was performed with Ver-PLGA@Lecithin (0.3 mL of 10 mg/mL in PBS) via oral administration and, 30 min later, with US exposure (0.5 W/cm$^2$, 10 min) with a medical US instrument (WED-100, WELLD) on the skin over stomach. At 48 h after treatment completion, blood samples were collected from 5 randomly selected mice from each group above, all mice in the above groups were sacrificed, and their stomachs were collected from the abdominal cavity.

Blood was collected in the orbits of mice into a 1.5-mL centrifuge tube containing 10 µL of 100 U heparin sodium solution (Changzhou Qianhong Biopharma Co., Ltd. China) and then temporarily stored on ice. Each as-collected blood sample was subsequently subjected to centrifugation (at 1500× g for 10 min at 4 °C) (5417 R, Eppendorf), and the resulting supernatant, which was female mouse serum, was collected, stored in an aliquot (200 µL each aliquot) at −80 °C, and used within one week. To quantify the levels of interleukin-1 beta (IL-1β), interleukin-6 (IL-6) and tumor necrosis factor-α (TNF-α), three proinflammatory factors, in the resulting female mouse serum samples, three ELISA kits (catalog no. PI301, PI326 and PT512, respectively, Beyotime Biotechnology) for IL-1β, IL-6 and TNF-α were used. Briefly, an aliquot of as-collected serum (200 µL) was removed from the −80 °C freezer and warmed naturally to room temperature and placed into a well in a 96-well microplate of an ELISA kit. Ten microlitres of the resulting serum and 90 µL of PBS were successively added, followed by incubation at 37 °C for 2 h, washing with wash buffer of the ELISA kit 3 times, and then the addition of horseradish peroxidase-labelled detection antibody and incubation at 37 °C for 1 h. The resulting sample was subsequently washed again with wash buffer of the ELISA kit 3 times, followed by the addition of the substrate 3,3',5,5'-tetramethylbenzidine (TMB), incubation at 37 °C in the dark for 20 min, and then the addition of the stop solution of the ELISA kit to terminate the reaction. The microplate was then subjected to optical density reading at 540 nm (OD$_{540}$) with a microplate reader (Varioskan, Thermo Scientific). To measure the levels of PG-1, PG-2 and G-17, three gastric function biomarkers, in the resulting female mouse serum samples, three ELISA kits for PG-1, PG-2 and G-17 (catalog no. ml037658, ml037635 and ml037633, respectively, Shanghai Enzyme-linked Biotechnology Co., Ltd. China) were used. Briefly, an aliquot of the as-collected serum sample (200 µL) was removed from the -80 °C freezer and warmed naturally to room temperature, and 10 µL of the resulting serum and 90 µL PBS were successively added to a well in a 96-well microplate of an ELISA kit, followed by incubation at 37 °C for 1 h, washing with wash buffer of the ELISA kit 3 times, and then the addition of horseradish peroxidase-labelled detection antibody provided in the ELISA kit and incubation at 37 °C for 30 min (PG-1 and PG-2) or 1 h (G-17). The resulting sample was subsequently washed again with wash

buffer from the ELISA kit 3 times (PG-1 and PG-2) or 5 times (G-17), followed by the addition of the substrate 3,3',5,5'-tetramethylbenzidine (TMB), incubation at 37 °C in the dark for 10 min (PG-1 and PG-2) or 15 min (G-17), and then the addition of the stop solution from the ELISA kit to terminate the reaction. The microplate was then subjected to optical density reading at 540 nm (OD$_{540}$) with a microplate reader (Varioskan, Thermo Scientific). For each proinflammatory factor (i.e., IL-1β, IL-6, or TNF-α) or each gastric function biomarker (i.e., PG-1, PG-2 and G-17), the calibration curve of OD$_{540}$ versus its concentration was determined using the standard solution provided in the ELISA kit by the vendor. Each trial was performed in triplicate, and the reported results are the averages of two independent trials.

The female mouse stomachs collected at 48 h after treatment completion were then cut open and rinsed with PBS to remove the contents therein. For the 7 mice in each group, 5 were used for quantifying the gastric H. pylori burden, while the remaining 2 were used for histology analysis. To quantify the gastric H. pylori burden, the as-rinsed stomachs were weighed, and each of them was subsequently immersed in a homogenization test tube containing PBS (1 mL) for subsequent tissue homogenization (for 2 min) with a homogenizer (Precellys Evolution, Bertin Instruments, OMNI-BEAD RUPTOR 12), followed by dilution of the resulting homogenate (1 mL) into sterile PBS (9 mL). The resulting mixture (1 mL) was subsequently subjected to serial 10-fold dilution 5 times, and for each resultant dilution, 20 µL was plated onto Columbia blood agar plates (Columbia agar medium supplemented with sterile defibrinated sheep blood (v./v. 5%)) and incubated at 37 °C under microaerobic conditions (10% CO$_2$, 85% N$_2$ and 5% O$_2$) for 4 days to form visible colonies. For histology analysis, female mouse gastric tissues (n = 2) from each group were fixed in 10% neutral buffered formalin, processed routinely into paraffin, sectioned, and stained with hematoxylin and eosin (H&E) for analysis under a microscope (MuVi-SPIM, LUXENDO).

## 16 S rRNA gene sequencing for gut microbiota

To analyse the effects of different treatments on the gut microbiota, we collected female mouse feces and sent the as-collected feces samples to Sangon Biotech (Shanghai) Co., Ltd. (Shanghai, China) for 16 S rRNA gene sequencing. Briefly, at 48 h after treatment completion, feces from mice in each group were collected and placed into a 1.5-mL centrifuge tube, and the centrifuge tube was then sealed and sent to Sangon Biotech Co., Ltd. (Shanghai) for 16 S rRNA gene sequencing analysis. The V3-V4 hypervariable regions of the 16 S rRNA gene were amplified with the primers 341-F (CCTACGGGNGGCWGCAG) and 805-R (GACTACHVGGGTATCTAATCC). The purified amplicons from a total of 30 samples from 6 groups (5 samples/group). The α diversity estimates were calculated using Shannon indices. Principal component analysis (PCA) was performed with the package "ade4" implemented in the statistical software R (version 3.6.0) to display the differences in different samples and to measure β diversity.

## Statistics and reproducibility

Statistical analysis was carried out with the two-sided Student's t test method and analysis of variance (ANOVA) with Tukey's multiple-comparison test. A P value of < 0.05 was considered statistically significant. A minimum of two independent experiments were conducted for all the studies, yielding consistent results. No statistical method was used to predetermine the sample size. No data was excluded. The experiments were not randomized. The investigators were not blinded to allocation during experiments and outcome assessment.

## Reporting summary

Further information on research design is available in the Nature Portfolio Reporting Summary linked to this article.

## Data availability

The authors declare that all data supporting the findings of this study are available within the paper and its Supplementary Information files. Source data is provided. The 16 S rRNA sequencing data generated in this study have been deposited in the Genome Sequence Archive (GSA) (https://ngdc.cncb.ac.cn/gsa/) at accession number CRA013610. Source data are provided with this paper.

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

## Acknowledgements

This work was supported in part by the National Natural Science Foundation of China (12174366, L.Y.), the Ministry of Education of China (via the Fundamental Research Funds for the Central Universities) (WK3450000005, L.Y.), and the Anhui Provincial Natural Science Foundation (2108085MC93, L.Y.).

## Author contributions

L.Y. conceived the idea; T.L., S.C., M.L., Y.X., and Y.Y. performed the experiments; T.L., X.C., Z. Z. and J.X. analysed the results; L.Y., T.L. and S.C. wrote the paper. F.G. and F. Z provided technical support for the experiments.

## Competing interests

A patent application was submitted to the China National Intellectual Property Administration (patent applicant: University of Science and Technology of China; application number: 202311440399.0); Status of application: submitted; authors of this manuscript who are also listed as inventors of this patent: Lihua Yang, Shuang Chai, Tao Liu; specific contents of the manuscript covered in the patent application: the preparation of two sonosensitizing nanoparticles coated with lecithin membranes (namely, they are Ver-PLGA@Lecithin, and Ce6-PLGA@Lecithin) and the characterizations of their hydrodynamic diameters using dynamic light scattering (DLS), in vitro plate killing assays of Ver-PLGA@Lecithin and Ce6-PLGA@Lecithin nanoparticles against *H. pylori*, determination of the effect of US exposure-triggered killing of *H. pylori* by Ver-PLGA@Lecithin nanoparticles in gastric *H. pylori* infection-bearing mouse models with clinical triple therapy included for comparison, and determination of VacA removal by Ver-PLGA@Lecithin. The other authors of this manuscript (Mingyang Li, Xu Chen, Yutao Xie, Zehui Zhao, Jingjing Xie, Yunpeng Yu, Feng Gao and Feng Zhu) do not have competing interests.
