## [Peer Review File · Nature Communications]

REVIEWER COMMENTS

Reviewer #1 (Remarks to the Author):

Sonodynamic therapy is innovative antibacterial approach that has previously been used in studies to treat infection with drug-resistant *E. coli*, *S. aureus* and *S. epidermidis*, and one previous study on *H. pylori* infection. Here, Liu et al have developed a novel sonodynamic *H. pylori* therapy that relies on ultrasound-induced release of reactive oxygen-species from Ver-PLGA@Lecithin nanoparticles administered to the stomach. Using a mouse model of *H. pylori* infection, the authors demonstrate that the nanoparticle sonosensitizer lead to effective clearance of *H. pylori* infection with minimal impact on gut microbiome. Importantly, this new therapy killed *H. pylori* as effectively as a standard triple therapy. The Ver-PLGA@Lecithin nanoparticles were composed of FDA-approved materials, making this work translationally relevant. The nanoparticles also sequestered VacA, a key *H. pylori* toxin, although this was only demonstrated in vitro and may have been unspecific. Overall, this is an interesting, novel and comprehensive study that may inform new clinical studies in humans. Major weaknesses include missing or inappropriate statistical analyses and the lack of a strong premise for some of the analyses and interpretations, e.g., the link between *H. pylori* therapy and NFALD, or major problems associated with intestinal dysbiosis following triple therapy. Some suggestions for improvement are outlined below:

- Overall, the manuscript is fairly lengthy at 50 pages, and almost 100 pages of additional material is provided in the supplemental data. I would suggest making the manuscript text more concise by eliminating overly detailed descriptions, interpretations, generic background information (e.g., line 378 “red blood cells carry oxygen”), and some of the overview cartoons from the results and limiting the supplemental data to essential material.
- Several previous studies performed in human patients have demonstrated that *H. pylori* infection alone is sufficient to drive significant changes in fecal microbiota. However, the data shown in Figure 1 suggest that *H. pylori* did not alter the gut microbiome. Do the authors have an explanation why their data differed from that obtained in a number of independent studies published by other groups (Dash, Khoder et al. 2019, Frost, Kacprowski et al. 2019, Wang, Li et al. 2019, Iino, Shimoyama et al. 2020)?
- How is damage to the gastric epithelium by the ROS released following ultrasound exposure prevented? The authors state that no damage was observed (Fig. 6f), but the magnification of the images is too low to assess this.
- Many of the figures, especially in the supplemental data, lack information about statistical significance (e.g., Fig. 3, 4, 6b, 8) and statistical tests performed. Some tests are suboptimal (e.g., use of multiple T test to compare five different treatments). It is recommended that graphs with individual data points, mean and SD, or confidence intervals are shown rather than heatmaps, and that P values are included in the figures throughout the manuscript.

- Figure 1: What was the rationale for considering a change of >10% in relative abundance significant? Given the general variability of the gut microbiome, are there any previous studies / guidelines that suggest that a 10% change is biologically relevant?
- Figure 2 (and supplementary Figure 24): While the increase in IL-1-RA observed in *H. pylori*-infected mice after triple therapy is interesting, it is an overinterpretation to state that these changes indicate a risk for NAFLD in the mice in the absence of other changes.
- Figure 3: The authors imply that the nanoparticles readily bind proteins, including VacA. Therefore, the observed sequestration of VacA from the *H. pylori* culture supernatants in vitro may be unspecific and thus irrelevant in a situation where other proteins are present including the normal stomach. It also is unclear why macrophages rather than gastric epithelial cells were tested in the experiments. Please comment.
- Figure 6b: Why was the triple therapy unable to clear *H. pylori* infection in the model system?
- Figure 8. Cell counts should be in blood not serum (which is cell free).
- Supplementary Fig. 21: Images in the last column appear to have different magnifications (in site of a size bar that is the same).
- Supplementary Fig. 25: Legend should say extended data for Fig. 8 (not 6). Cell counts should be labeled as "lymphocytes in blood", not "lymphocytes in serum".
- Please include section headings (Introduction, Results, etc.).
- Methods: Include information on the strain of *H. pylori* used for infecting the mice.
- Methods, line 700: How did the researchers determine where the area over the stomach was in the mice? It might be useful to include a diagram or photograph.

Relevant papers:

Dash, N. R., G. Khoder, A. M. Nada and M. T. Al Bataineh (2019). "Exploring the impact of *Helicobacter pylori* on gut microbiome composition." *PLoS One* 14(6): e0218274.

Frost, F., T. Kacprowski, M. Ruhlemann, C. Bang, A. Franke, K. Zimmermann, M. Nauck, U. Volker, H. Volzke, R. Biffar, C. Schulz, J. Mayerle, F. U. Weiss, G. Homuth and M. M. Lerch (2019). "*Helicobacter pylori* infection associates with fecal microbiota composition and diversity." *Sci Rep* 9(1): 20100.

Iino, C., T. Shimoyama, D. Chinda, H. Sakuraba, S. Fukuda and S. Nakaji (2020). "Influence of *Helicobacter pylori* Infection and Atrophic Gastritis on the Gut Microbiota in a Japanese Population." *Digestion* 101(4): 422-432.

Wang, D., Y. Li, H. Zhong, Q. Ding, Y. Lin, S. Tang, Y. Zong, Q. Wang, X. Zhang, H. Yang, R. Wang and X. Liu (2019). "Alterations in the human gut microbiome associated with *Helicobacter pylori* infection." *FEBS Open Bio* 9(9): 1552-1560.

Reviewer #2 (Remarks to the Author):

In this work, Liu et al. used a lecithin bilayer-coated and bio-compatible nanoparticle, Ver-PLGA@Lecithin, as the model sonosensitizer to enhance the sonodynamic elimination of gastric H. pylori in mouse models without inducing gut microbiota dysbiosis. The author collected a lot of evidence to demonstrate the efficacy and safety of this sonodynamic therapy against gastric H. pylori infection. Although a lot of results are comprised, some mechanisms are not well demonstrated. Meanwhile, to this reviewer's knowledge, the nanoparticle in this manuscript is also not well designed and characterized, and some results are conflicted with the previous reports. Major revision is needed.

1.The author claimed that a lecithin bilayer-coated nanoparticle, whose surface is full of lecithin, can help remove VacA by adsorbing proteins from the environmental fluids to form protein coronas. However, this lecithin molecule consists of a phosphorylcholine group, which is a zwitterionic material. Such phosphorylcholine-abundant nanoparticle surface is already demonstrated with the protein-adsorption-resistance capacity, widely used as the antifouling coatings for nanomedicines or implant devices (Biomaterials 2019, 219, 119357; Biomaterials 2020, 231, 119680; Adv. Mater. 2021, 33, 2008493; Adv. Mater. 2022, 34, 2109189). Even many studies also reported the prior anti-protein-adsorption capacity of zwitterionic material than PEG (Adv. Mater. 2022, 34, 2109189). Therefore, the anti-protein adsorption results in Fig. 3b and 3c are not convincing.

2.To support the result in Fig. 3c, the author needs to confirm the higher binding affinity between lecithin bilayer-coated nanoparticles with VacA than that of PLGA-PEG nanoparticles by isothermal titration microcalorimetry and quartz crystal microbalance.

3.The formation of protein corona on lecithin bilayer-coated nanoparticle surface should be characterized by TEM, AFM, or SEM. Protein adsorption curve is also required.

4.The lecithin bilayer-coated nanoparticle was prepared by embedding the PLGA nanoparticle into the Lecithin vesicles under US (Supplementary Fig. 4, 7, 8, 9). How is the encapsulating efficiency? This coating efficiency will eventually determine the protein-adsorption results in Fig 3c.

5.All TEM images should present multiple nanoparticles to confirm the successful construction of a uniform core-shell structure.

6. TEM figures in Supplementary Fig. 8b and 9b, Fig. 7b and 5b, are the same particles. However, the author claimed they are different.

7. Is the formation of protein corona affecting the ROS-generating efficacy?

8. Although the manuscript has many figures (1-8), a lot of schemes are presented in the figures.

9. The lecithin chemical structure in Fig 4a lacks an O atom near the fatty acid chains.

10. The English expression should be briefer and to the point. The numbers of experiments are not very many, but the presentation words have exceeded the SI.

Reviewer #3 (Remarks to the Author):

The manuscript submitted by Liu et al entitled "A safe and gut microbiota-friendly therapy for gastric *Helicobacter pylori* infection" reports a novel strategy for *Helicobacter pylori* eradication in mice.

The so called sonodynamic therapy reflects the increasing number of primary antibiotic resistances against frequently used antibiotics as metronidazole, clarithromycin and levofloxacin and also the temporary disturbance of gut microbiota after eradication treatment.

Helicobacter pylori infection affects more than the half world's population and is associated with more than 90% of gastric cancer cases. In 2015, *H. pylori* infection was defined as infectious disease with the need to eradicate irrespective of symptoms and complications.

In daily routine, virulence factors of the pathogen associated to distinct phenotypes and risks for complication are not used to stratify eradication treatment.

Unfortunately, there are some points in the manuscript submitted that have to be discussed here:

1. The references used are not up to date and some relevant publications are missed. E.g. Systematic review: gastric microbiota in health and disease. Rajilic-Stojanovic M, Figueiredo C, Smet A, Hansen R, Kupcinskas J, Rokkas T, Andersen L, Machado JC, Ianiro G, Gasbarrini A, Leja M, Gisbert JP, Hold GL. *Aliment Pharmacol Ther.* 2020 Mar;51(6):582-602. doi: 10.1111/apt.15650. Epub 2020 Feb 13. PMID: 32056247. The interplay between *Helicobacter pylori* and gastrointestinal microbiota. Chen CC, Liou JM, Lee YC, Hong TC, El-Omar EM, Wu MS. *Gut Microbes.* 2021 Jan-Dec;13(1):1-22. doi: 10.1080/19490976.2021.1909459. PMID: 33938378. Long-term changes of gut microbiota, antibiotic resistance, and metabolic parameters after *Helicobacter pylori* eradication: a multicentre, open-label, randomised trial. Liou JM, Chen CC, Chang CM, Fang YJ, Bair MJ, Chen PY, Chang CY, Hsu YC, Chen MJ, Chen CC, Lee JY, Yang TH, Luo JC, Chen CY, Hsu WF, Chen YN, Wu JY, Lin JT, Lu TP, Chuang EY, El-Omar EM, Wu MS; Taiwan Gastrointestinal Disease and *Helicobacter* Consortium. *Lancet Infect Dis.* 2019 Oct;19(10):1109-1120. doi: 10.1016/S1473-3099(19)30272-5. PMID: 31559966. Second-line levofloxacin-based quadruple therapy versus bismuth-based quadruple therapy for *Helicobacter pylori* eradication and long-term changes to the gut microbiota and antibiotic resistance: a multicentre, open-label, randomised controlled trial. Liou JM, Jiang XT, Chen CC, Luo JC, Bair MJ, Chen PY, Chou CK, Fang YJ, Chen MJ, Chen CC, Lee JY, Yang TH, Yu CC, Kuo CC, Chiu MC, Chen CY, Shun CT, Hu WH, Tsai MH, Hsu YC, Tseng CH, Chang CY, Lin JT, El-Omar EM, Wu MS; Taiwan Gastrointestinal Disease and *Helicobacter* Consortium. *Lancet Gastroenterol Hepatol.* 2023 Mar;8(3):228-241. doi: 10.1016/S2468-1253(22)00384-3. Epub 2022 Dec 19.

PMID: 36549320.

..and many others in the field. As the authors discussed, there are ongoing discussions regarding the effect of antibiotic treatment of *H. pylori* infection and (temporary) disturbances of gut microbial pattern. But, as shown in the publications of Liou and his group, the effect is only temporary and initial changes of alpha and beta diversity are recovered after one year. Please discuss.

2. The structure of the manuscript needs to be revised from my point of view. Reporting a novel and innovative antibacterial strategy the main focus have to be linked to this concept. The data presented about the effect a triple therapy on microbial dysbiosis might be an argument for novel strategies, but, on the one hand, are overcome in human studies as shown above and, on the other hand, the argument is explained in the introduction. The novelty has to be rated low.

The increased risk for the development of NAFLD as shown in the experiment can be presented in another manuscript, but the long term effect of these changes has to be studied before this conclusion can be made.

3. Please explain more detailed the infection challenge of the model used, especially I couldn't find the strain used in your experiments.

To summarize, I recommend to focus on the novel and important antibacterial strategy and the methods used. The arguments for the need of novel strategies after WHO statement are explained in the introduction and do not have been proven again in the experiments (Fig 1 and 2), especially when using only short periods of follow- up.

Responses to Reviewer Comments

A safe and gut microbiota-friendly therapy for gastric *Helicobacter pylori* infection

Tao Liu^{1,2,3}, *Shuang Chai*^{1,2,3}, *Mingyang Li*^{1,2,3}, *Xu Chen*^{1,2,3}, *Yutao Xie*^{1,2,3}, *Zehui Zhao*^{1,2,3}, *Jingjing Xie*^{1,2,3}, *Yunpeng Yu*^{1,2,3}, *Feng Gao*¹, *Feng Zhu*⁴, *Lihua Yang*^{1,2,3*}

~~~~~

### ***REVIEWER COMMENTS***

*Reviewer #1 (Remarks to the Author):*

*Sonodynamic therapy is innovative antibacterial approach that has previously been used in studies to treat infection with drug-resistant E. coli, S. aureus and S. epidermidis, and one previous study on H. pylori infection. Here, Liu et al have developed a novel sonodynamic H. pylori therapy that relies on ultrasound-induced release of reactive oxygen-species from Ver-PLGA@Lecithin nanoparticles administered to the stomach. Using a mouse model of H. pylori infection, the authors demonstrate that the nanoparticle sonosensitizer lead to effective clearance of H. pylori infection with minimal impact on gut microbiome. Importantly, this new therapy killed H. pylori as effectively as a standard triple therapy. The Ver-PLGA@Lecithin nanoparticles were composed of FDA-approved materials, making this work translationally relevant. The nanoparticles also sequestered VacA, a key H. pylori toxin, although this was only demonstrated in vitro and may have been unspecific. Overall, this is an interesting, novel and comprehensive study that may inform new clinical studies in humans.*

Author response: We gratefully thank the reviewer for the positive remarks.

*Major weaknesses include missing or inappropriate statistical analyses and the lack of a strong premise for some of the analyses and interpretations, e.g., the link between H. pylori therapy and*

*NFALD, or major problems associated with intestinal dysbiosis following triple therapy. Some suggestions for improvement are outlined below:*

- *Overall, the manuscript is fairly lengthy at 50 pages, and almost 100 pages of additional material is provided in the supplemental data. I would suggest making the manuscript text more concise by eliminating overly detailed descriptions, interpretations, generic background information (e.g., line 378 “red blood cells carry oxygen”), and some of the overview cartoons from the results and limiting the supplemental data to essential material.*

Author response: We gratefully thank the reviewer for the kind reminder. In revision, to make the manuscript and the Supplementary Information file more concise, we have accordingly deleted the unnecessary statements in them and moved the unnecessary cartoons in the manuscript to the Supplementary Information file. Because of the Results and Discussion sections and the Materials and Methods sections on the experiments we added in revision, the manuscript and the Supplementary Information file (both double-spaced) are now at 49 and 88 pages, respectively.

- *Several previous studies performed in human patients have demonstrated that *H. pylori* infection alone is sufficient to drive significant changes in fecal microbiota. However, the data shown in Figure 1 suggest that *H. pylori* did not alter the gut microbiome. Do the authors have an explanation why their data differed from that obtained in a number of independent studies published by other groups (Dash, Khoder et al. 2019, Frost, Kacprowski et al. 2019, Wang, Li et al. 2019, Iino, Shimoyama et al. 2020)?*

Author response: We gratefully thank the reviewer for the critical comments and for suggesting these references. In revision, we have accordingly carried out literature research on the effects of *H. pylori* infection on host gut microbiota, and the results are summarized in our Supplementary Dataset 2 (Effects of *H. pylori* infection on host gut microbiota). We agree with the reviewer that, in human patients, *H. pylori* infection does cause changes to host gut microbiota, as demonstrated in the references suggested by the reviewer (N. R. Dash, G. Khoder, et al., *PLOS ONE*, 2019, 14, e0218274; F. Frost, T. Kacprowski, et al., *Sci. Rep.*, 2019, 9, 20100; D. Wang, Y. Li, et al., *FEBS Open Bio*, 2019, 9, 1552-1560; C. Iino, T. Shimoyama, et al., *Digestion*, 2019, DOI:

10.1159/000500634) and as summarized in a recently published review paper (*Gut Microbes*, 2021, 13, 1909459) and also in our Supplementary Dataset 2 (Effects of *H. pylori* infection on host gut microbiota). But there are two things we need keep in mind when looking at these results.

First, as to the specific changes which gastric *H. pylori* infection causes to human gut microbiota, controversy exists in the previous studies (*Gut Microbes*, 2021, 13, 1909459). For example, some studies observed non-significant difference in  $\alpha$ -diversity of gut microbiota between *H. pylori*-positive patients and *H. pylori*-negative controls (D. Wang, Y. Li, et al., *FEBS Open Bio*, 2019, 9, 1552-1560; *Helicobacter*, 2020, 25, e12721; *Front. Cell. Infect. Microbiol.*, 2019, 9, 375; *EBioMedicine*, 2018, 35, 87-96; *Front. Cell. Infect. Microbiol.*, 2018, 8, 202), while some others showed significantly higher  $\alpha$ -diversity for *H. pylori*-positive patients than *H. pylori*-negative controls (*Sci. Rep.*, 2019, 9, 20100; *Digestion*, 2019, DOI: 10.1159/000500634; *PLOS ONE*, 2019, 14, e0218274; *Helicobacter*, 2019, 24, e12590). Similarly, some studies show no difference in  $\beta$ -diversity between *H. pylori*-positive patients and *H. pylori*-negative controls while others find “ $\beta$ -diversity differs” (*Gut Microbes*, 2021, 13, 1909459) likely because  $\beta$ -diversity varies significantly even within the group of *H. pylori*-positive human subjects depending on individual *H. pylori* load determined by stool antigen test (*Sci. Rep.*, 2019, 9, 20100). Moreover, the specific bacterial commensals (phyla or genera) that show differences in relative abundance in the gut microbiota between *H. pylori*-positive patients and *H. pylori*-negative controls vary in different studies (*Gut Microbes*, 2021, 13, 1909459). Such controversy arises likely because *H. pylori* infection may not be the only health disorder the human subjects involved in some studies have. For example, children with gastritis but tested *H. pylori*-negative exhibit significantly different gut microbiota composition as compared with their healthy controls and gastric *H. pylori* infection aggravates the gut microbiota dysbiosis in children with gastritis (*Front. Cell. Infect. Microbiol.*, 2019, 9 article 375). For *H. pylori*-infected human subjects, those with severe atrophic gastritis exhibit significantly higher relative abundance of *Lactobacillus* in gut microbiota than counterparts with mild or no atrophic gastritis (*Frontier in Immunology*, 2018, 9, article 712), indicating influence of the severity of atrophic gastritis on the relative abundance of *Lactobacillus* in gut microbiota. In addition, one prior study shows increased *Bacteroidetes*-to-*Firmicutes* (B:F) ratio in gut microbiota for *H. pylori*-positive patients after triple therapy even at 3 months after the therapy as compared to that before the therapy (*BMJ Open Gastro.*, 2017, e000182) while another one reveals

decreased B:F ratio for *H. pylori*-positive young adults after antibiotic-based eradication therapy (*PLoS One*, 2016, 11, e0151893), and this difference arises likely because of the difference in age and race between the human subjects involved in these two studies (*BMJ Open Gastro.*, 2017, e000182).

Second, the current clinical tests for *H. pylori* infection (*Crit. Rev. Anal. Chem.*, 2023 DOI: 10.1080/10408347.2022.2163585) tell whether a human subject is *H. pylori*-positive or -negative but, for *H. pylori*-positive cases, cannot indicate for-how-long a human subject has been infected. Note that the duration of infection between the time-point of initial gastric *H. pylori* infection and that of gut microbiota monitoring is readily controllable in animal studies. We, in revision, have also checked through some references on how gastric *H. pylori* infection affects the gut microbiota of animal hosts (Supplementary Dataset 2: Effects of *H. pylori* infection on host gut microbiota) but realized that the shortest duration we found is 1 month, which is still longer than what we used in our animal study, where fecal samples were collected on day 23 after the *H. pylori* inoculation (Supplementary Figure 26b). More importantly, the duration of infection crucially determines whether the gut microbiotas of *H. pylori*-positive hosts are similar as or different from those of their *H. pylori*-negative counterparts; in murine models, examinations on fecal samples collected at different months after *H. pylori* infection show that, although the microbial community structures between infected subjects and naïve controls are similar within 1 month after *H. pylori* infection, they become more and more divergent as the post-infection observation window extends (*Cell Report*, 2016,14, 1-13, <http://dx.doi.org/10.1016/j.celrep.2016.01.017>).

Third, our animal study used mouse models, rather than human subjects, which may have implications on the effects of *H. pylori* infection on host gut microbiota. Still, although our study used a different host type (mouse, not human subjects) and reflects short-term effects of gastric *H. pylori* infection on gut microbiota (23-day after infection establishment), it revealed two significant changes to gut microbiota due to gastric *H. pylori* infection, which are: (1) The *H. pylori* infected yet not-treated mice (*i.e.*, the control group) exhibited significantly lower  $\alpha$ -diversity ( $P = 0.0026$ ) than the *H. pylori* negative mice (*i.e.*, the healthy group) (Supplementary Figure 26c), and (2) one gut commensal bacterium, *unclassified-Clostridiales*, is significantly down-regulated due to gastric *H. pylori* infection (Supplementary Figure 28).

**Supplementary Figure 26. Gut microbiota after triple therapy. a-b.** Schematic illustrations on (a) the triple therapy regimen used for treating gastric *H. pylori* infection and (b) on the schedule of triple therapy treatment in mouse models bearing gastric *H. pylori* infection and follow-up feces collection for (c-f) gut microbiota analysis. **c-d.** The (c) alpha ( $\alpha$ ) diversity (Shannon) and (d) principal component analysis (PCA) on the beta ( $\beta$ ) diversity of mouse gut microbiota. The control group was *H. pylori* infected but not treated. In the box plot, the upper and lower quartile as outlined by top and bottom boundary were divided by line showing the median value. **e-f.** The relative abundances of gut bacterial species (e) at the phylum level and (f) at the genus level according to 16S rRNA gene sequencing data. (e) Data are clustered according to the number of samples per group. **g-j.** Statistical analysis on (g-h) the difference between the triple therapy *versus* healthy groups and (i-j) on the difference between the triple therapy *versus* control groups. The change in relative abundance of gut bacterial species after triple therapy is indicated both as (g, i) the fold of change (*i.e.*, “ $\log_2(\text{triple therapy/Healthy})$ ” and “ $\log_2(\text{triple therapy/Control})$ ”) and (h, j) as the absolute value of change (*i.e.*, “Relative Abundance (triple therapy-Healthy, %)” and “Relative Abundance (triple therapy-Control, %)”). For bacterium to be marked as a significantly perturbed one in the plots of  $-\log_{10}(P \text{ value})$  *versus* the fold change in bacterial relative abundance, its fold change in bacterial relative abundance needs to be  $\geq 2$  or  $\leq 1/2$  and, meanwhile, its  $P$  value needs to be  $< 0.05$ . For bacterium to be marked as a significantly perturbed one in the plots of  $-\log_{10}(P \text{ value})$  *versus* the absolute change in bacterial relative abundance, its  $P$  value needs to be  $< 0.05$ . Bacterial species whose relative abundances got significantly up- and down-regulated after triple therapy are marked in red and in blue, respectively.

**Supplementary Figure 28. Difference in gut microbiota composition between the control *versus* healthy groups.** The difference in relative abundance of gut bacterial species between the control *versus* healthy groups is indicated both (a, c) as the absolute value of change (*i.e.*, “Relative Abundance (Control-Healthy, %)”) and (b, d) as the fold of change (*i.e.*, “ $\log_2(\text{Control}/\text{Healthy})$ ”) and plotted separately at the (a-b) phylum and (c-d) genus levels. For bacterium to be marked as a significantly perturbed one in the plots of  $-\log_{10}(P \text{ value})$  *versus* the fold change in bacterial relative abundance, its fold change in bacterial relative abundance needs to be  $\geq 2$  or  $\leq 1/2$  and, meanwhile, its  $P$  value needs to be  $< 0.05$ . For bacterium to be marked as a significantly perturbed one in the plots of  $-\log_{10}(P \text{ value})$  *versus* the absolute change in bacterial relative abundance, its  $P$  value needs to be  $< 0.05$ . Bacterial species whose relative abundances got significantly up- and down-regulated after triple therapy are marked in red and in blue, respectively.

• *How is damage to the gastric epithelium by the ROS released following ultrasound exposure prevented? The authors state that no damage was observed (Fig. 6f), but the magnification of the images is too low to assess this.*

Author response: We gratefully thank the reviewer for the critical comments. In revision, we have accordingly re-imaged the gastric tissue slices under microscopy but at higher magnifications, as shown in the figure below (*i.e.*, Figure 5f in the revised manuscript).

**Fig. 5 Efficacy of Ver-PLGA@Lecithin mediated sonodynamic therapy. f.** Microscopy images of hematoxylin and eosin (H&E) stained stomach tissues collected from gastric *H. pylori* infection-bearing mouse models at 48-h after differing treatment.

In revision, we have further carried out a same animal study as shown in Figure 6 of our previous manuscript but replaced the H&E staining with terminal deoxynucleotidyl transferase-mediated dUTP nick end labeling (TUNEL) staining, a widely used technique for detecting apoptotic cell death. Specifically, we stained slices of the gastric tissues collected from different treatment groups with dUTP conjugated with the green fluorescent FITC and DAPI (a widely used stain that, upon binding to AT regions of DNA, emits blue fluorescence). Our results (Supplementary Figure 23 in the revised manuscript) revealed negligible extent of green fluorescence for gastric tissues from all treatment groups including those after our sonodynamic therapy (*i.e.*, Ver-PLGA@Lecithin + US), suggesting negligible extent of apoptotic cell death in gastric tissue after our sonodynamic therapy.

**Supplementary Figure 23. Assessment of gastric mucosal injury.** Fluorescence microscopy images of TUNEL (TdT-mediated dUTP Nick-End Labeling) stained stomach tissues collected from gastric *H. pylori* infection-bearing mouse models at 48-h after differing treatment (*i.e.*, control, triple therapy, US, Ver-PLGA@Lecithin and Ver-PLGA@Lecithin + US). Blue fluorescence (DAPI) indicates DNA and green fluorescence (FITC-dUTP) indicates apoptotic cell death.

• *Many of the figures, especially in the supplemental data, lack information about statistical significance (e.g., Fig. 3, 4, 6b, 8) and statistical tests performed. Some tests are suboptimal (e.g., use of multiple T test to compare five different treatments). It is recommended that graphs with individual data points, mean and SD, or confidence intervals are shown rather than heatmaps, and that P values are included in the figures throughout the manuscript.*

Author response: We gratefully thank the reviewer for the attention to details and for the kind reminder. In revision, we have accordingly added the *P* values and the corresponding statistical tests used for calculating the *P* values to the figures and the corresponding figure captions, respectively.

Now, Figure 1 in the revised manuscript (some of its panels correspond to Figure 3 in our previous manuscript) appears and its figure caption reads:

**Fig. 1** Selecting the nanoparticle efficient in removing vacuolating cytotoxin A (VacA). **a**. Schematic illustration on the pathways by which VacA secreted by *H. pylori* attacks gastric epithelial cells and their block due to VacA removal *via* protein adsorption onto externally added

nanoparticles. **b.** Schematic illustration on our model nanoparticles, PLGA@Lecithin and PLGA@PEG. **c.** Contents of total adsorbed proteins on PLGA@PEG and PLGA@Lecithin (both at 10 mg/mL) re-collected at different time-points after the initiation of their incubation in water supplemented with *H. pylori* culture supernatant (HCS) (10 mg/mL). **d.** Cryo-EM images of (**top**) PLGA@PEG and (**bottom**) PLGA@Lecithin (both at 30 mg/mL) re-collected after 12-h pre-incubation (**left**) in water and (**right**) in HCS-supplemented water (10 mg/mL). Scale bar = 50 nm. **e.** Photographs of SDS-PAGE gels with adsorbed proteins on PLGA@PEG and PLGA@Lecithin (80 mg/mL) with and without 12-h pre-incubation in HCS-supplemented water (10 mg/mL), with that of HCS-supplemented water (5 mg/mL) included as a reference. **f.** Contents of adsorbed VacA on PLGA@PEG and PLGA@Lecithin (8 mg/mL) re-collected after 24-h pre-incubation in HCS-supplemented water (20 mg/mL) and those of residual VacA in the as-treated solution. Bar heights are reported as averages of two independent trials (n = 3 in each independent trial). **g-i.** Viability ratios of human gastric adenocarcinoma cells (AGS cells) (**g**) after 24-h treatment with PLGA@PEG and PLGA@Lecithin at differing doses in PBS. (**h**) 24-h incubation in HCS-supplemented PBS at differing concentrations. (**i**) after 24-h incubation in HCS-supplemented PBS (20 mg/mL) that was pre-treated (for 12 h) with a nanoparticle (PLGA@PEG or PLGA@Lecithin) at differing doses. Bar height represents the average of two independent trials (n = 3 in each independent trial). *P* values are calculated using two-sided Student's t-test.

Now, Figure 2 in the revised manuscript (*i.e.*, Figure 4 in our previous manuscript) appears and its figure caption reads:

**Fig. 2** *In vitro* performances of our nano-sonosensitizers. **a**. Schematic illustration on the detection of reactive oxygen species (ROS) generated by a USS-PLGA@Lecithin in response to ultrasound (US) exposure. **b**. Fluorescence intensity of DCFH solution with a USS-PLGA@Lecithin (100 μg/mL in USS dose) or its empty counterpart PLGA@Lecithin upon US exposure (at 0.5 W/cm2 for 10 min). DCFH solution without any nanoparticle was included as a

blank. **c-e.** Fluorescence intensity of DCFH solution containing a USS-PLGA@Lecithin (c-d) at a fixed nanoparticle concentration (100  $\mu\text{g}/\text{mL}$  in USS dose) upon US exposure (c) at differing output power density (for 10 min) and (d) for differing exposure time (at 0.5  $\text{W}/\text{cm}^2$ ) and (e) at differing nanoparticle concentration (in USS dose) upon US exposure (at 0.5  $\text{W}/\text{cm}^2$  for 10 min). **f.** Fluorescence intensity of SOSG solution containing a USS-PLGA@Lecithin at differing concentration (in USS dose) upon US exposure (at 0.5  $\text{W}/\text{cm}^2$  for 10 min). **g.** Schematic illustration on the *in vitro* antibacterial assays against *H. pylori* using a USS-PLGA@Lecithin. **h.** Survival *H. pylori* cell counts (in Log CFU/mL, where CFU means colony forming unit) after treatment with a USS-PLGA@Lecithin at differing concentration (in USS dose) in PBS (right) with and (left) without US exposure (at 0.5  $\text{W}/\text{cm}^2$  for 10 min). **i.** Survival *H. pylori* cell counts (in Log CFU/mL) after treatment with Ver-PLGA@Lecithin in PBS upon US exposure (left) at differing output power density (for 10 min) and (right) for differing exposure time (at 0.5  $\text{W}/\text{cm}^2$ ). **j.** Scanning electron microscopy (SEM) images of *H. pylori* cells after different treatments. Ver-PLGA@Lecithin dose was fixed at 1  $\text{mg}/\text{mL}$  in PBS, and US exposure was at 0.5  $\text{W}/\text{cm}^2$  for 10 min. **k-l.** Survival *H. pylori* cell counts (in Log CFU/mL) after treatment with Ver-PLGA@Lecithin at differing concentration (in Ver dose) (k) in  $\beta$ -carotene supplemented (1.5  $\text{mg}/\text{mL}$ ) PBS and (l) in simulated gastric fluid (SGF) in the presence and absence of US exposure (at 0.5  $\text{W}/\text{cm}^2$  for 10 min). Bar height represents the average of two independent trials ( $n = 3$  in each independent trial).

Now, Figure 5 in the revised manuscript (*i.e.*, Figure 6 in our previous manuscript) appears and its figure caption reads:

**Fig. 5 Efficacy of Ver-PLGA@Lecithin mediated sonodynamic therapy.** **a.** Schematic illustration on the schedule of different treatments including sonodynamic therapy (*i.e.*, “Ver-PLGA@Lecithin + US”) to mouse models bearing gastric *H. pylori* infection and follow-up blood and tissue collection. **b.** *H. pylori* burden per gram of stomach tissues (in Log CFU/g) from mice treated with “Ver-PLGA@Lecithin + US” group, with those from mice treated with PBS (*i.e.*, control), triple therapy, US exposure alone, and Ver-PLGA@Lecithin alone included for comparison. In the box plot, the upper and lower quartile as outlined by top and bottom boundary were divided by line showing the median value ( $n = 5$  biologically independent mice in once trial). *P* values are calculated using two-sided Student’s t-test. **c-e.** Serum levels of (c) interleukin-6 (IL-6), (d) tumor necrosis factor- $\alpha$  (TNF- $\alpha$ ), and (e) interleukin-1 beta (IL-1 $\beta$ ) in gastric *H. pylori* infection-bearing mouse models after differing treatment. Bar heights are reported as average  $\pm$  standard deviation ( $n = 5$  biologically independent mice in once trial). *P* values are calculated using two-sided Student’s t-test. **f.** Microscopy images of hematoxylin and eosin (H&E) stained stomach tissues collected from gastric *H. pylori* infection-bearing mouse models at 48-h after differing treatment. **g-i.** Serum levels of (g) pepsinogen I (PG-1), (h) pepsinogen II (PG-2) and (i) gastrin-17 (G-17) in gastric *H. pylori* infection-bearing mouse models at 48-h after differing treatment. Bar heights are reported as average  $\pm$  standard deviation ( $n = 5$  biologically independent mice in once trial). Red dash-line indicates the average serum level in healthy mice. *P* values are calculated using two-sided Student’s t-test.

Now, Figure 7 in the revised manuscript (*i.e.*, Figure 8 in our previous manuscript) appears and its figure caption reads:

**Fig. 7 Biosafety of our sonodynamic therapy.** **a.** Schematic illustration on the schedule of differing treatment to mouse models bearing gastric *H. pylori* infection and follow-up blood collection for liver and renal function tests and complete blood count (CBC) test. **b-e.** Serum levels of (b) alanine aminotransferase (ALT), (c) aspartate aminotransferase (AST), (d) creatinine (CREA), and (e) blood urea nitrogen (BUN). **f-m.** Blood levels of (f) red blood cell count (RBC), (g) hemoglobin, (h) white blood cell count (WBC), (i) eosinophil, (j) lymphocyte, (k) neutrophils, (l) monocyte, and (m) platelet. Red dash-line indicates the average serum level in healthy mice. Bar heights are reported as average  $\pm$  standard deviation ( $n = 5$  biologically independent mice in once trial).  $P$  values are calculated using two-sided Student's t-test.

Data in Figure 4 in the previous manuscript are represented in heat-maps and, hence, it is inconvenient to add  $P$  values to those heat-maps. Therefore, in revision, we have accordingly added  $P$  values to plots on the raw data of the heat-maps reporting reactive oxygen species (ROS) detection results in previous Figure 4 (*i.e.*, Supplementary Fig. 12-14 in the revised Supplementary Information) and to plots on the raw data of the *in vitro* antibacterial assays in previous Figure 4 (*i.e.*, Supplementary Fig. 15 in the revised Supplementary Information). Meanwhile, the statistical analysis tests used for calculating these  $P$  values have been added to figure captions of figures above.

• *Figure 1: What was the rationale for considering a change of >10% in relative abundance significant? Given the general variability of the gut microbiome, are there any previous studies / guidelines that suggest that a 10% change is biologically relevant?*

Author response: We gratefully thank the reviewer for the critical comments. In the plots of  $\log_{10}P$  versus absolute change in relative abundance in our previous manuscript, we considered a phylum or genus that simultaneously has a  $P$  value of  $<0.05$  and an absolute value for change in relative abundance of  $>10\%$  as one that got significantly perturbed, for reasons as follows.

When we were analyzing our data, we noticed that some phyla and/or genera exhibited absolute values for change in relative abundance of  $>10\%$  but a fold change in relative abundance within the range of 0.5-2. Such phyla and/or genera could not be marked as significantly up- or down-regulated ones in our plots of  $\log_{10}P$  versus relative change in relative abundance, despite that an absolute value for change in relative abundance of  $>10\%$  is apparently non-negligible. Therefore,

to account for such phyla and/or genera, we further plotted  $\log_{10}P$  versus absolute change in relative abundance and, in the resulting plots, intuitively marked a phylum and/or genus that simultaneously have a  $P$  value of  $<0.05$  and an absolute change in relative abundance of  $>10\%$  or  $<-10\%$  as a significantly up- or down-regulated one.

In revision, we performed literature research and noticed that, among the reports on the effects of *H. pylori* infection and/or antibiotic-based *H. pylori* eradication therapy on gut microbiota in human subjects, most of them rate whether a difference in relative abundance is significant or not simply by checking whether or not its correspondent  $P$  value is  $<0.05$ , without taking into account the relevant change (absolute or fold) in relative abundance in gut microbiota. Even if some have taken into account the relative abundance in gut microbiota, they normally exclude the species with abundances lower than certain minimum thresholds, rather than considering the differences in relative abundance lower than certain minimum thresholds. For example, for gut commensals at the genus level, one report on how *H. pylori* infection affects gut microbiota (*Sci. Rep.*, 2019, 9, 20100) and one report on how *H. pylori* eradication therapy affects gut microbiota (*Helicobacter*, 2020, 25, e12721) compared only the genera with a mean relative abundance of  $>0.1\%$  and considered those with  $P < 0.05$  as significantly perturbed ones, while another report on how *H. pylori* eradication therapy affects gut microbiota (*Front. Cell. Infect. Microbiol.*, 2018, 8, article 202) analyzed only the genera with average relative abundances of  $>0.01\%$  and considered those with  $P < 0.05$  as significantly perturbed ones. For gut commensals at the phylum level, one report on how *H. pylori* eradication therapy affects gut microbiota (*Helicobacter*, 2020, 25, e12721) compared the phyla with mean relative abundance of  $>1\%$  and considered those with  $P < 0.05$  as significantly perturbed ones. Another report on how antibiotic-based *H. pylori* eradication therapy affects gut microbiota (*Lancet Gastroenterol. Hepatol.*, 2023, 8, 228-241) shows that filtering out low-abundance species with a prevalence of  $<1\%$  and average relative abundance  $<0.00001$  yielded consistent results for the diversity of gut microbiota and resistome compared with analysis of all available samples.

Still, there are some exceptions that, when considering whether a commensal bacterium is a significantly perturbed one, have taken into account its difference in relative abundance after *H. pylori* infection and/or eradication therapy in addition to the corresponding  $P$  value. One report on

how *H. pylori* infection affects gut microbiota (*PLOS ONE*, 2019, 14, e0218274) compared only the genera with an absolute difference in relative abundance of >0.1% and considered those with  $P < 0.05$  as significantly perturbed ones. Two reports on the impact of *H. pylori* infection and eradication therapy on gut microbiota (*Helicobacter*, 2019, 24, e12590; *Front. Cell. Infect. Microbiol.*, 2019, 9, article 375) plotted the  $P$  value *versus* difference between/in mean proportions (%) (there proportion means relative abundance in gut microbiota) but did not introduce minimum thresholds for the difference between/in mean proportions when considering whether or not a specie is significantly perturbed.

In our plots of  $\log_{10}P$  *versus* absolute change in relative abundance, the consideration that only phyla and genera that have both  $P$  values of <0.05 and absolute values for the difference in relative abundance of >10% were marked as significantly perturbed ones apparently underestimates, rather than overestimates, the total numbers of significantly perturbed phyla and genera. Despite of this, our plots still revealed quite some species that got significantly perturbed after triple therapy (Supplementary Figure 26 in our revised manuscript).

In revision, to reduce the possible underestimation on the number of gut commensal species that got significantly perturbed after our sonodynamic therapy due to our prior incorporation of 10% as the minimum threshold for the absolute difference in relative abundance, we deleted the requirement for a minimum threshold for the absolute difference in relative abundance when considering whether or not a commensal bacterium is a significantly perturbed one. Instead, for the plots of  $-\log P$  *versus* absolute difference in relative abundance, we considered a gut commensal bacterium as a significantly perturbed one as long as the bacterium has a  $P$  value of <0.05 (Figure 6g,i and Supplementary Figure 26 h,j in the revised manuscript).

**Supplementary Figure 26. Gut microbiota after triple therapy. a-b.** Schematic illustrations on (a) the triple therapy regimen used for treating gastric *H. pylori* infection and (b) on the schedule of triple therapy treatment in mouse models bearing gastric *H. pylori* infection and follow-up feces collection for (c-f) gut microbiota analysis. **c-d.** The (c) alpha ( $\alpha$ ) diversity (Shannon) and (d) principal component analysis (PCA) on the beta ( $\beta$ ) diversity of mouse gut microbiota. The control group was *H. pylori* infected but not treated. In the box plot, the upper and lower quartile as outlined by top and bottom boundary were divided by line showing the median value. **e-f.** The relative abundances of gut bacterial species (e) at the phylum level and (f) at the genus level according to 16S rRNA gene sequencing data. (e) Data are clustered according to the number of samples per group. **g-j.** Statistical analysis on (g-h) the difference between the triple therapy *versus* healthy groups and (i-j) on the difference between the triple therapy *versus* control groups. The change in relative abundance of gut bacterial species after triple therapy is indicated both as (g, i) the fold of change (*i.e.*, “ $\log_2(\text{triple therapy/Healthy})$ ” and “ $\log_2(\text{triple therapy/Control})$ ”) and (h, j) as the absolute value of change (*i.e.*, “Relative Abundance (triple therapy-Healthy, %)” and “Relative Abundance (triple therapy-Control, %)”). For bacterium to be marked as a significantly perturbed one in the plots of  $-\log_{10}(P \text{ value})$  *versus* the fold change in bacterial relative abundance, its fold change in bacterial relative abundance needs to be  $\geq 2$  or  $\leq 1/2$  and, meanwhile, its  $P$  value needs to be  $< 0.05$ . For bacterium to be marked as a significantly perturbed one in the plots of  $-\log_{10}(P \text{ value})$  *versus* the absolute change in bacterial relative abundance, its  $P$  value needs to be  $< 0.05$ . Bacterial species whose relative abundances got significantly up- and down-regulated after triple therapy are marked in red and in blue, respectively.

• *Figure 2 (and supplementary Figure 24): While the increase in IL-1-RA observed in H. pylori-infected mice after triple therapy is interesting, it is an overinterpretation to state that these changes indicate a risk for NAFLD in the mice in the absence of other changes.*

Author response: We gratefully thank the reviewer for the critical comments. We agree with the reviewer that the increase in IL-1RA serum level in *H. pylori*-infected mice after triple therapy is interesting. Note that the applicability of IL-1RA as an independent biomarker for non-alcoholic steatohepatitis (NASH) (*J. Hepatol.*, 2012, 56,663-670), a severe type of NAFLD, is reported only by just one research article, rather than a widely-used practice in research or in clinics. Therefore, in revision, we have accordingly adopted the kind suggestions by both this reviewer and reviewer #3 and removed our previous statement that the observed increase in IL-1RA serum level in *H. pylori*-infected mice after triple therapy indicates increased risk of NAFLD.

Still, it should be noted that endogenous IL-1RA is a natural anti-inflammatory protein and its elevated level in blood is associated with diverse animal models of disease and with human diseases such as rheumatic diseases and chronic renal failure (*Ann. Rev. Immunol.*, 1998, 16, 27-

55). Therefore, our observation that IL-1RA serum levels of the triple therapy group, but not those of the control group (*i.e.*, *H. pylori*-infected but treated with PBS), were significantly higher than those of the healthy group (Figure 3c-f in the previous manuscript) does suggest certain adverse effects after triple therapy, though what specific adverse effects or disease(s) the observed increase in IL-1RA serum level correspond to are unknown at the current stage. And it is noteworthy that, though five mouse models were included in each of these three treatment groups, the five corresponding data points exhibited a narrow distribution without any odd one, and this was the case for all these three treatment groups. So, in revision, we mentioned this observation but in a rather low-key tone, simply hoping to arouse the audiences' attention to potential adverse effects after triple therapy from an extra-gastrointestinal perspective.

In fact, this work is not the first one that correlate gut microbiota dysbiosis after triple therapy with adverse effects from an extra-gastrointestinal perspective. A prior study shows that, for *H. pylori*-positive patients, 1-week triple therapy renders their *Bacteroidetes*-to-*Firmicutes* (B:F) ratios in gut microbiota significantly higher even at 3 months after the therapy than those before the therapy and, meanwhile, makes their plasma levels of active ghrelin significantly lower even at 3 months after the therapy than those before the therapy and that significant correlation is observed in their plasma level of active ghrelin and their change in B:F ratio (*BMJ Open Gastro.*, 2017, e000182). Considering that ghrelin is a multifunctional hormone that facilitates fat storage and regulates body weight (*Nature*, 2000, 407, 908-913) and a high value for the ghrelin blood test (greater than 650 pg/mL in human subjects) may indicate fasting, cachexia, and anorexia, the authors of this prior study think the observed increase in plasma level of active ghrelin suggests a “lean” phenotype for patients after triple therapy (*BMJ Open Gastro.*, 2017, e000182).

• *Figure 3: The authors imply that the nanoparticles readily bind proteins, including VacA. Therefore, the observed sequestration of VacA from the H. pylori culture supernatants in vitro may be unspecific and thus irrelevant in a situation where other proteins are present including the normal stomach. It also is unclear why macrophages rather than gastric epithelial cells were tested in the experiments. Please comment.*

Author response: We gratefully thank the reviewer for the critical comments. In our previous manuscript, to demonstrate the VacA removal by our model nanoparticles (PLGA@PEG, and PLGA@Lecithin) from solution pre-supplemented with *H. pylori* culture supernatant (HCS) (Figure 3c in the previous manuscript), we co-incubated these nanoparticles with HCS-supplemented water and then separated the nanoparticles from the solution for subsequent VacA quantification. According to our VacA quantification assays (Figure 3c in the previous manuscript) and SDS-PAGE assays performed in revision (Figure 1e), the HCS contains diverse proteins including VacA (a 88-kDa protein). Moreover, our SDS-PAGE assays (Figure 1e) revealed that co-incubating our model nanoparticles with HCS-supplemented media resulted in the accumulation of not only VacA but also some other proteins naturally present in the HCS. In addition, we carried out Bradford assays in revision to quantify the total amount of proteins that accumulated onto our model nanoparticles after their pre-incubation with HCS-supplemented media (Supplementary Figure 3), and comparison between these Bradford assays and our previous VacA quantification assays revealed that, on either nanoparticle (PLGA@Lecithin, or PLGA@PEG), the total amount of proteins thereon (Supplementary Figure 3) is significantly higher than that of VacA (Figure 3c in the previous manuscript) (units: mg/mL *versus* ng/mL), indicating existence of other proteins beside VacA. Collectively, these assays demonstrate that our model nanoparticles can remove VacA from complex environmental fluids where other proteins are present and, though performed *in vitro*, still strongly support their ability to do so in the more complex environment within stomach *in vivo*.

**Fig. 1** Selecting the nanoparticle efficient in removing vacuolating cytotoxin A (VacA). **a**. Schematic illustration on the pathways by which VacA secreted by *H. pylori* attacks gastric epithelial cells and their block due to VacA removal *via* protein adsorption onto externally added

nanoparticles. **b.** Schematic illustration on our model nanoparticles, PLGA@Lecithin and PLGA@PEG. **c.** Contents of total adsorbed proteins on PLGA@PEG and PLGA@Lecithin (both at 10 mg/mL) re-collected at different time-points after the initiation of their incubation in water supplemented with *H. pylori* culture supernatant (HCS) (10 mg/mL). **d.** Cryo-EM images of (**top**) PLGA@PEG and (**bottom**) PLGA@Lecithin (both at 30 mg/mL) re-collected after 12-h pre-incubation (**left**) in water and (**right**) in HCS-supplemented water (10 mg/mL). Scale bar = 50 nm. **e.** Photographs of SDS-PAGE gels with absorbed proteins on PLGA@PEG and PLGA@Lecithin (80 mg/mL) with and without 12-h pre-incubation in HCS-supplemented water (10 mg/mL), with that of HCS-supplemented water (5 mg/mL) included as a reference. **f.** Contents of adsorbed VacA on PLGA@PEG and PLGA@Lecithin (8 mg/mL) re-collected after 24-h pre-incubation in HCS-supplemented water (20 mg/mL) and those of residual VacA in the as-treated solution. Bar heights are reported as averages of two independent trials (n = 3 in each independent trial). **g-i.** Viability ratios of human gastric adenocarcinoma cells (AGS cells) (**g**) after 24-h treatment with PLGA@PEG and PLGA@Lecithin at differing doses in PBS. (**h**) 24-h incubation in HCS-supplemented PBS at differing concentrations. (**i**) after 24-h incubation in HCS-supplemented PBS (20 mg/mL) that was pre-treated (for 12 h) with a nanoparticle (PLGA@PEG or PLGA@Lecithin) at differing doses. Bar height represents the average of two independent trials (n = 3 in each independent trial). *P* values are calculated using two-sided Student's *t*-test.

**Supplementary Figure 3. Quantifying adsorbed proteins on nanoparticles.** **a.** Standard curve measured through a Bradford Protein Assay Kit using BSA as a protein standard. **b.** Contents of total adsorbed proteins on nanoparticles (PLGA@PEG or PLGA@Lecithin) (10 mg/mL) re-collected after 12-h incubation in HCS-supplemented water (10 mg/mL), with those treated similarly but in water included for comparison. Bar heights are reported as average  $\pm$  standard deviation (n = 3 independent experiments). *P* values are calculated using two-sided Student's *t*-test.

Indeed, likely owing to the observed ability of lecithin-bilayer coated nanoparticles to remove VacA (Fig. 3b-e in the previous manuscript), mice in the “Ver-PLGA@Lecithin alone” group exhibited detectably reduced serum levels of interleukin-1 beta (IL-1 $\beta$ ), interleukin-6 (IL-6) and tumor necrosis factor- $\alpha$  (TNF- $\alpha$ ) — three pro-inflammatory cytokines that exhibit higher-than-normal serum levels due to gastric *H. pylori* infection — as compared to those in the “control” group and, in particular for TNF- $\alpha$ , the observed reduction was of statistical difference ( $0.01 < P < 0.05$ ) (Fig. 6c-e and Supplementary Fig. 20 in the previous manuscript). Clearly, for the *H. pylori* infection-induced up-regulation of IL-1 $\beta$  and IL-6, the observed suppressive effects of our sonodynamic therapy is attributable to the US-triggered ROS generation of Ver-PLGA@Lecithin, but not Ver-PLGA@Lecithin or US alone; yet, for the *H. pylori* infection-induced up-regulation of TNF- $\alpha$ , the observed suppression is contributed by both Ver-PLGA@Lecithin alone and its US-triggered ROS generation, which combined with lecithin-bilayer coated nanoparticles’ ability to remove VacA (Fig. 3b-e in the previous manuscript) supports the ability of Ver-PLGA@Lecithin to remove VacA (*via* accumulating VacA onto its surface) in the complex protein-rich environment within stomach *in vivo*.

We examined whether nanoparticles’ removal of VacA from HCS helps detoxify HCS (Supplementary Fig. 6a), using human gastric adenocarcinoma (AGS) cells as the representative for gastric epithelial cells— a target of VacA (*Nat. Rev. Microbiol.* 2005, 3, 320-332)— as did in prior studies on how *H. pylori* infection or VacA influence gastric epithelial cells (*Cell Death Dis.* 2017, 8, 3207; *Glycobiology* 2019, 29, 151-162; *Sci. Rep.* 2019, 9, 38). To AGS cells, PLGA@PEG and PLGA@Lecithin both lack intrinsic cytotoxicity even up to 10 mg/mL (in PLGA dose) (Fig. 1g) while HCS of the *H. pylori* strain used in this work was intrinsically cytotoxic (Fig. 1h). Nevertheless, pre-incubating (for 12 h) the HCS with either PLGA@PEG or PLGA@Lecithin mitigated the cytotoxicity of the HCS (Fig. 1i and Supplementary Fig. 6b). For example, PBS containing 20 mg/mL HCS reduced the average AGS cell viability ratio to ~46.6%, but its pre-incubation with PLGA@PEG and PLGA@Lecithin (8 mg/mL in PLGA dose) rendered the average AGS cell viability ratio bounce upwards to 53.9% and 64.7%, respectively. Similarly, pre-incubating PBS containing 10 mg/mL HCS with PLGA@PEG and PLGA@Lecithin (both at 2 mg/mL in PLGA dose) enabled the average AGS cell viability ratio to increase relatively by ~5.4% and ~12.4%, respectively (Supplementary Fig. 6b). Clearly, both PLGA@PEG and

PLGA@Lecithin can detoxify HCS and, in doing so, PLGA@Lecithin is more efficient than PLGA@PEG. Taken together, above results suggest that, both in removing VacA and in detoxifying HCS, PLGA@Lecithin is superior to PLGA@PEG.

**Fig. 1** Selecting the nanoparticle efficient in removing vacuolating cytotoxin A (VacA). **a**. Schematic illustration on the pathways by which VacA secreted by *H. pylori* attacks gastric epithelial cells and their block due to VacA removal *via* protein adsorption onto externally added

nanoparticles. **b.** Schematic illustration on our model nanoparticles, PLGA@Lecithin and PLGA@PEG. **c.** Contents of total adsorbed proteins on PLGA@PEG and PLGA@Lecithin (both at 10 mg/mL) re-collected at different time-points after the initiation of their incubation in water supplemented with *H. pylori* culture supernatant (HCS) (10 mg/mL). **d.** Cryo-EM images of (**top**) PLGA@PEG and (**bottom**) PLGA@Lecithin (both at 30 mg/mL) re-collected after 12-h pre-incubation (**left**) in water and (**right**) in HCS-supplemented water (10 mg/mL). Scale bar = 50 nm. **e.** Photographs of SDS-PAGE gels with adsorbed proteins on PLGA@PEG and PLGA@Lecithin (80 mg/mL) with and without 12-h pre-incubation in HCS-supplemented water (10 mg/mL), with that of HCS-supplemented water (5 mg/mL) included as a reference. **f.** Contents of adsorbed VacA on PLGA@PEG and PLGA@Lecithin (8 mg/mL) re-collected after 24-h pre-incubation in HCS-supplemented water (20 mg/mL) and those of residual VacA in the as-treated solution. Bar heights are reported as averages of two independent trials (n = 3 in each independent trial). **g-i.** Viability ratios of human gastric adenocarcinoma cells (AGS cells) (**g**) after 24-h treatment with PLGA@PEG and PLGA@Lecithin at differing doses in PBS. (**h**) 24-h incubation in HCS-supplemented PBS at differing concentrations. (**i**) after 24-h incubation in HCS-supplemented PBS (20 mg/mL) that was pre-treated (for 12 h) with a nanoparticle (PLGA@PEG or PLGA@Lecithin) at differing doses. Bar height represents the average of two independent trials (n = 3 in each independent trial). *P* values are calculated using two-sided Student's *t*-test.

**Supplementary Figure 6. Cytotoxicity of HCS after pre-incubation with nanoparticles. a.** Schematic illustration on the cytotoxicity assay of HCS after 12-h pre-incubation with or without a nanoparticle (PLGA@Lecithin, or PLGA@PEG). Human gastric adenocarcinoma cells (AGS

cells) were used as the representative for gastric epithelial cells. **b.** Viability ratios of AGS cells after 24-h treatment with HCS-supplemented PBS (10 mg/mL) pre-incubated with a nanoparticle (PLGA@Lecithin, or PLGA@PEG) at differing doses. Bar height represents the average of two independent trials (n = 3 in each independent trial). *P* values are calculated using two-sided Student's t-test.

• *Figure 6b: Why was the triple therapy unable to clear H. pylori infection in the model system?*

Author response: We gratefully thank the reviewer for the critical comments. In revision, we have accordingly performed literature research on animal studies that show *H. pylori* burdens after triple therapy and found that, though efficacy varies depending on specifics of the triple therapy used, none of the studies completely cleared the gastric *H. pylori* infection (as summarized in the Table below), which is consistent with what we observed in this work. Though many possible reasons may account for this incomplete clearance of gastric *H. pylori* infection, one important reason must be the difference in the frequency of drug dosing between animal models and patients. For patients, physicians usually prescribe triple or quadruple therapy 3 times a day for 14 consecutive days; but, in animal studies, the drug dosage is normally once a day for 7 consecutive days (Table below).

| Type of Animals                 | Amount of H. pylori per injection | Number of H. pylori infections | Drug composition of triple therapy            | Treatment Cycle (week) | Treatment Times | Bacterial burden in stomach after triple therapy (Log CFU/g) | Ref.                                                                |
|---------------------------------|------------------------------------------|---------------------------------------|-----------------------------------------------|------------------------|-----------------|--------------------------------------------------------------|---------------------------------------------------------------------|
| C57BL/6 male mice at 6-8 weeks  | 0.3 mL, $1 \times 10^9$ CFU/mL           | 3 (on days 3, 5 and 7)                | omeprazole, amoxicillin, and clarithromycin   | 1                      | 5               | 6.08                                                         | Proc. Natl. Acad. Sci. U.S.A. 2014, 111 , 17600-17605 |
| BALB/c female mice at 3-4 weeks | 0.3 mL, $5 \times 10^7$ CFU/mL           | 4 (on days 4, 5, 6, and 7)            | omeprazole, amoxicillin, and clarithromycin   | 1                      | 4               | 2.93                                                         | Nat. Commun. 2021, 12 , 2002                          |
| C57BL/6 female mice at 6 weeks  | 0.3 mL, $1 \times 10^9$ CFU/mL           | 4 (on days 1, 3, 5 and 7)             | omeprazole, amoxicillin, and clarithromycin   | 1                      | 3               | 3.60                                                         | Antimicrob. Agents Chemother. 2021,                          |
| C57BL/6 male mice               | $2 \times 10^8$ CFU/mouse                | 3 (on days 2, 4 and 6)                | omeprazole, amoxicillin, and clarithromycin   | 1                      | 3               | 5.52                                                         | Biomaterials 2021, 271 , 120745                       |
| C57BL/6 mice at 4-6 weeks       | $1 \times 10^8$ CFU/mouse                | 7 (once every two days for two weeks) | lansoprazole, amoxicillin, and clarithromycin | 1                      | 5               | 1.70                                                         | J. Control. Release 2022, 348 , 370-385               |
| BALB/c mice at 7 weeks          | 0.3 mL, $1 \times 10^8$ CFU/mL           | 4 (on days 3, 5 and 7)                | omeprazole, amoxicillin, and clarithromycin   | 1                      | 3               | 4.33                                                         | ACS Appl. Mater. Interfaces 2022, 14 , 26418-26430    |
| BALB/c female mice at 6-8 weeks | 0.2 mL, $1 \times 10^8$ CFU/mL           | 4 (on days 4, 5, 6 and 7)             | omeprazole, amoxicillin, and clarithromycin   | 1                      | 4               | 4.08                                                         | Adv. Sci. 2023, 10 , 2206957                          |
| C57BL/6 mice                    | 0.15 mL, $1 \times 10^7$ CFU/mL          | 3 (on days 3, 5 and 7)                | CLA+PPI                                       | 1                      | 5               | 4.96                                                         | Small 2021, 17 , 2006871                              |
| C57BL/6 female mice at 8 weeks  | 0.3 mL, $1 \times 10^8$ CFU/mL           | 4 (on days 1, 3, 5 and 7)             | omeprazole, amoxicillin, and clarithromycin   | 1                      | 4               | 5.75                                                         | This Work                                                           |

- *Figure 8. Cell counts should be in blood not serum (which is cell free).*

Author response: We gratefully thank the reviewer for the kind reminder. In revision, we have accordingly revised the labels from “cells in serum” to “cells in blood”. Now, Figure 7 in the revised manuscript (*i.e.*, Fig. 8 in previous manuscript) appears:

**Fig. 7 Biosafety of our sonodynamic therapy.** **a.** Schematic illustration on the schedule of differing treatment to mouse models bearing gastric *H. pylori* infection and follow-up blood

collection for liver and renal function tests and complete blood count (CBC) test. **b-e.** Serum levels of (b) alanine aminotransferase (ALT), (c) aspartate aminotransferase (AST), (d) creatinine (CREA), and (e) blood urea nitrogen (BUN). **f-m.** Blood levels of (f) red blood cell count (RBC), (g) hemoglobin, (h) white blood cell count (WBC), (i) eosinophil, (j) lymphocyte, (k) neutrophils, (l) monocyte, and (m) platelet. Red dash-line indicates the average serum level in healthy mice. Bar heights are reported as average  $\pm$  standard deviation ( $n = 5$  biologically independent mice in once trial). *P* values are calculated using two-sided Student's t-test.

• *Supplementary Fig. 21: Images in the last column appear to have different magnifications (in site of a size bar that is the same).*

Author response: We gratefully thank the reviewer for the critical comments. In revision, we have accordingly re-imaged the gastric tissue slices under microscopy. Now, this figure (*i.e.*, Supplementary Figure 22 in the revised manuscript) appears:

**Supplementary Figure 22. Histology of mouse stomach tissues.** Microscopy images of hematoxylin and eosin (H&E) stained stomach tissues collected from gastric *H. pylori* infection-bearing mouse models at 48-h after differing treatment (*i.e.*, control, triple therapy, US, Ver-PLGA@Lecithin and Ver-PLGA@Lecithin + US).

• *Supplementary Fig. 25: Legend should say extended data for Fig. 8 (not 6). Cell counts should be labeled as “lymphocytes in blood”, not “lymphocytes in serum”.*

Author response: We gratefully thank the reviewer for the kind reminder. In revision, we have accordingly revised the y-axis labels for Supplementary Fig. 29 (Supplementary Fig. 25 in previous manuscript) from “cells in serum” to “cells in blood”, and the legend from “extended data for Fig. 6” to “extended data for Fig. 7” (Fig. 8 in previous manuscript).

Now, Supplementary Fig. 29 (*i.e.*, Supplementary Fig. 25 in previous manuscript) looks and its legend reads:

**Supplementary Figure 29. Biosafety of our sonodynamic therapy.** The extended data for Fig. 7, which show serum levels of (a) albumin and (b) uric acid (UA) in the liver and renal function tests and blood levels of (c) hematocrit, (d) mean corpuscular hemoglobin (MCH), (e) mean corpuscular volume (MCV), (f) mean corpuscular hemoglobin concentration (MCHC), (g) red blood cell distribution width-standard deviation (RDW-SD) and (h) red blood cell volume distribution width-coefficient of variation (RDW-CV), (i) eosinophil %, (j) lymphocyte %, (k) monocytes count % and (l) neutrophils %, (m) mean platelet volume (MPV), (n) plateletcrit (PCT) %, (o) platelet-larger cell ratio (P-LCR). and (p) platelet distribution width (PDW) in the CBC test. All blood samples collected on day 9 from mice treated with different therapies. Bar heights are reported as average  $\pm$  standard deviation (n = 5 biologically independent mice in once trial). *P* values are calculated using two-sided Student's t-test.

- *Please include section headings (Introduction, Results, etc.).*

Author response: We gratefully thank the reviewer for the kind reminder. In revision, we have accordingly added sections headings to the manuscript.

- *Methods: Include information on the strain of H. pylori used for infecting the mice.*

Author response: We gratefully thank the reviewer for the kind reminder. In revision, we have accordingly checked to make sure that the information on the strain of *H. pylori* used in this work, which is ATCC no. 43504 purchased from American Type Culture Collection (ATCC) (Virginia, USA), is clearly shown in the **MATERIALS** section. It reads: “The *H. pylori* strain (ATCC no. 43504) used in this work was purchased from the American Type Culture Collection (ATCC) (Virginia, USA).”

- *Methods, line 700: How did the researchers determine where the area over the stomach was in the mice? It might be useful to include a diagram or photograph.*

Author response: We gratefully thank the reviewer for the kind reminder. In revision, we have accordingly added photographs of mouse models receiving US exposure for sonodynamic therapy (Supplementary Figure 25).

**Supplementary Figure 25. Pictures of mice receiving US exposure in sonodynamic therapy.** The ultrasound probe was placed snugly on the skin over stomach of a C57BL/6J mouse. Prior to the US exposure, the mouse was subjected to a depilatory treatment on the skin over stomach.

*Relevant papers:*

*Dash, N. R., G. Khoder, A. M. Nada and M. T. Al Bataineh (2019). "Exploring the impact of Helicobacter pylori on gut microbiome composition." PLoS One 14(6): e0218274.*

*Frost, F., T. Kacprowski, M. Ruhlemann, C. Bang, A. Franke, K. Zimmermann, M. Nauck, U. Volker, H. Volzke, R. Biffar, C. Schulz, J. Mayerle, F. U. Weiss, G. Homuth and M. M. Lerch (2019). "Helicobacter pylori infection associates with fecal microbiota composition and diversity." Sci Rep 9(1): 20100.*

*Iino, C., T. Shimoyama, D. Chinda, H. Sakuraba, S. Fukuda and S. Nakaji (2020). "Influence of Helicobacter pylori Infection and Atrophic Gastritis on the Gut Microbiota in a Japanese Population." Digestion 101(4): 422-432.*

*Wang, D., Y. Li, H. Zhong, Q. Ding, Y. Lin, S. Tang, Y. Zong, Q. Wang, X. Zhang, H. Yang, R.*

Wang and X. Liu (2019). "Alterations in the human gut microbiome associated with *Helicobacter pylori* infection." *FEBS Open Bio* 9(9): 1552-1560.

Author response: We gratefully thank the reviewer for suggesting these references. In revision, we have accordingly carried out literature research on the effects of *H. pylori* infection on host gut microbiota, and the results are summarized in our Supplementary Dataset 2 (Effects of *H. pylori* infection on host gut microbiota).

*Reviewer #2 (Remarks to the Author):*

*In this work, Liu et al. used a lecithin bilayer-coated and bio-compatible nanoparticle, Ver-PLGA@Lecithin, as the model sonosensitizer to enhance the sonodynamic elimination of gastric *H. pylori* in mouse models without inducing gut microbiota dysbiosis. The author collected a lot of evidence to demonstrate the efficacy and safety of this sonodynamic therapy against gastric *H. pylori* infection. Although a lot of results are comprised, some mechanisms are not well demonstrated. Meanwhile, to this reviewer's knowledge, the nanoparticle in this manuscript is also not well designed and characterized, and some results are conflicted with the previous reports. Major revision is needed.*

*1. The author claimed that a lecithin bilayer-coated nanoparticle, whose surface is full of lecithin, can help remove VacA by adsorbing proteins from the environmental fluids to form protein coronas. However, this lecithin molecule consists of a phosphorylcholine group, which is a zwitterionic material. Such phosphorylcholine-abundant nanoparticle surface is already demonstrated with the protein-adsorption-resistance capacity, widely used as the antifouling coatings for nanomedicines or implant devices (Biomaterials 2019, 219, 119357; Biomaterials 2020, 231, 119680; Adv. Mater. 2021, 33, 2008493; Adv. Mater. 2022, 34, 2109189). Even many studies also reported the prior anti-protein-adsorption capacity of zwitterionic material than PEG (Adv. Mater. 2022, 34, 2109189). Therefore, the anti-protein adsorption results in Fig. 3b and 3c are not convincing.*

Author response: We gratefully thank the reviewer for the critical comments. We agree with the reviewer that zwitterionic materials are commonly believed to have anti-fouling properties as does polyethylene glycol (PEG). But, even PEGylation cannot completely block the adsorption of plasma proteins; instead, it selectively promotes the adsorption of certain plasma proteins (*Nat. Nanotechnol.*, 2016, 11, 372-377; *Adv. Mater.*, 2019, 31, 1803335). Similarly, modifying nanoparticle surfaces with zwitterionic materials cannot necessarily block protein corona formation; instead, it inhibits protein adsorption to varying extent depending on the chemical structures of the zwitterionic materials. For example, one report (*Biomaterials*, 2019, 219, 119357) compared the protein corona formation around nanoparticles coated with polymers with side-chains bearing three classical zwitterionic groups (sulfobetaine, phosphorylcholine, and carboxybetaine) and revealed that, though protein adsorption is virtually absent around the sulfobetaine-coated nanoparticles, significant protein adsorption and formation of a small protein layer are observed for the carboxybetaine-coated and the phosphorylcholine-coated nanoparticles, respectively. Another report (*Biomaterials*, 2020, 231, 119680) compared the protein adsorption around nanoparticles coated with membranes of three zwitterionic polymers — which are poly(2-methacryloyloxyethyl phosphorylcholine), poly(sulfobetaine methacrylate), and poly(carboxybetaine methacrylate) — with that around counterparts coated with brushes of the same zwitterionic polymers and showed that, no matter whether the zwitterionic polymer coatings are in membrane form or in brush form, appreciable protein adsorption is detected around all the as-coated nanoparticles (albeit the zwitterionic polymer membranes are superior to the corresponding zwitterionic brushes in resisting protein adsorption). Clearly, coating a nanoparticle with zwitterionic polymers, no matter whether in membrane form or in brush form, cannot completely resist the protein adsorption around the particle.

As to lecithin bilayer-coated nanoparticles, protein corona does form around them, as demonstrated in a prior work by us (*Nat. Commun.*, 2022, 13, article no. 4137); by comparing the protein coronas around lecithin bilayer-coated nanoparticles differing in elasticity with those around their PEGylated counterparts, we found that, no matter whether a nanoparticle is coated with a bilayer of lecithin (a PEG-free natural lipid) or with that of synthetic lipids (one of which is PEGylated), protein corona does form around the particle and, as nanoparticle elasticity

increases, the trend in which protein corona patterns vary for lecithin bilayer-coated nanoparticles is similar as that for their PEGylated counterparts.

In our previous manuscript, we quantified the amount of VacA accumulated onto our model nanoparticles (PLGA@PEG, and PLGA@Lecithin) (Figure 3c in the previous manuscript) despite that we co-cultured these nanoparticles with *H. pylori* culture supernatant (HCS), which is naturally a complex mixture containing diverse proteins including VacA (an 88-kDa protein) (Figure 1e in the revised manuscript). So, in revision, we further quantified (*via* Bradford assays) the total amounts of proteins that accumulated onto our model nanoparticles after their incubation with HCS and found significant protein adsorption both on PLGA@Lecithin and on PLGA@PEG (Supplementary Figure 3 in the revised manuscript). Note that, on a same model nanoparticle (PLGA@Lecithin or PLGA@PEG), the total amount of accumulated proteins is significantly higher than that of accumulated VacA, suggesting accumulation of both VacA and non-VacA proteins naturally present in HCS onto the nanoparticle surface. We hence performed protein SDS-PAGE electrophoresis assays for proteins accumulated on our nanoparticles and confirmed the accumulation of VacA and other proteins thereon (Figure 1e in the revised manuscript). Still, it is noteworthy that more HCS proteins were observed on PLGA@Lecithin than on its PLGA@PEG counterpart, as revealed consistently in the VacA quantification assays (Figure 3c in our previous manuscript), in the protein SDS-PAGE electrophoresis assays (Figure 1e in the revised manuscript), and in the Bradford assays (Supplementary Figure 3 in the revised manuscript), suggesting superiority of PLGA@Lecithin to PLGA@PEG in accumulating proteins naturally present in HCS.

**Fig. 1** Selecting the nanoparticle efficient in removing vacuolating cytotoxin A (VacA). **a**. Schematic illustration on the pathways by which VacA secreted by *H. pylori* attacks gastric epithelial cells and their block due to VacA removal *via* protein adsorption onto externally added

nanoparticles. **b.** Schematic illustration on our model nanoparticles, PLGA@Lecithin and PLGA@PEG. **c.** Contents of total adsorbed proteins on PLGA@PEG and PLGA@Lecithin (both at 10 mg/mL) re-collected at different time-points after the initiation of their incubation in water supplemented with *H. pylori* culture supernatant (HCS) (10 mg/mL). **d.** Cryo-EM images of (**top**) PLGA@PEG and (**bottom**) PLGA@Lecithin (both at 30 mg/mL) re-collected after 12-h pre-incubation (**left**) in water and (**right**) in HCS-supplemented water (10 mg/mL). Scale bar = 50 nm. **e.** Photographs of SDS-PAGE gels with absorbed proteins on PLGA@PEG and PLGA@Lecithin (80 mg/mL) with and without 12-h pre-incubation in HCS-supplemented water (10 mg/mL), with that of HCS-supplemented water (5 mg/mL) included as a reference. **f.** Contents of adsorbed VacA on PLGA@PEG and PLGA@Lecithin (8 mg/mL) re-collected after 24-h pre-incubation in HCS-supplemented water (20 mg/mL) and those of residual VacA in the as-treated solution. Bar heights are reported as averages of two independent trials (n = 3 in each independent trial). **g-i.** Viability ratios of human gastric adenocarcinoma cells (AGS cells) (**g**) after 24-h treatment with PLGA@PEG and PLGA@Lecithin at differing doses in PBS. (**h**) 24-h incubation in HCS-supplemented PBS at differing concentrations. (**i**) after 24-h incubation in HCS-supplemented PBS (20 mg/mL) that was pre-treated (for 12 h) with a nanoparticle (PLGA@PEG or PLGA@Lecithin) at differing doses. Bar height represents the average of two independent trials (n = 3 in each independent trial). *P* values are calculated using two-sided Student's t-test.

**Supplementary Figure 3. Quantifying adsorbed proteins on nanoparticles.** **a.** Standard curve measured through a Bradford Protein Assay Kit using BSA as a protein standard. **b.** Contents of total adsorbed proteins on nanoparticles (PLGA@PEG or PLGA@Lecithin) (10 mg/mL) re-collected after 12-h incubation in HCS-supplemented water (10 mg/mL), with those treated similarly but in water included for comparison. Bar heights are reported as average  $\pm$  standard deviation (n = 3 independent experiments). *P* values are calculated using two-sided Student's t-test.

2. To support the result in Fig. 3c, the author needs to confirm the higher binding affinity between lecithin bilayer-coated nanoparticles with VacA than that of PLGA-PEG nanoparticles by isothermal titration microcalorimetry and quartz crystal microbalance.

Author response: We gratefully thank the reviewer for the kind reminder. To further confirm this, we evaluated the interactions of HCS with nanoparticles of differing surface chemistry with isothermal titration calorimetry (ITC), a powerful technique commonly used for examining the nonspecific binding of proteins on nanoparticles (*Proc. Natl Acad. Sci. U. S. A.* 104, 2050–2055 (2007); *Nat. Nanotechnol.* 12, 288–290 (2017)). We carried out ITC analysis using 30 mg/mL *H. pylori* culture supernatant solution (in Millipore water) to titrate 20 mg/mL PLGA@PEG or PLGA@Lecithin nanoparticles (in Millipore water). Upon titration of HCS, appreciable thermal effects (specifically, exothermic) were observed in both PLGA@Lecithin and PLGA@PEG (Supplementary Fig. 4 in the revised manuscript). But in view of the HCS solution is a mixed protein system, it is not possible to compare the binding affinity of the proteins in HCS on these two nanoparticles. Clearly, the proteins in HCS can interact with both particle surfaces, that is, the particle surface does form a protein corona.

**Supplementary Figure 4.** Raw data from isothermal titration calorimetry (ITC) assays, in which *H. pylori* culture supernatant solution was titrated into (a) PLGA@Lecithin nanoparticle and (b) PLGA@PEG nanoparticle dispersions.

*3. The formation of protein corona on lecithin bilayer-coated nanoparticle surface should be characterized by TEM, AFM, or SEM. Protein adsorption curve is also required.*

Author response: We gratefully thank the reviewer for the kind reminder. The adsorption of proteins from HCS onto surfaces of our model nanoparticles was further verified with cryo-electron microscopy (Cryo-EM). Under Cryo-EM, both PLGA@Lecithin and PLGA@PEG appeared spherical with smooth surfaces that are nearly uniform in contrast; nevertheless, after 12-h incubation in HCS-supplemented water, both PLGA@Lecithin and PLGA@PEG, though still spherical, acquired rough and wrinkled surfaces that are non-uniform in contrast and have many dark dots which circle over the nanoparticle surfaces in a noncontinuous way (Fig. 1d in the revised manuscript). We attribute the observed dark dots to be adsorbed proteins (and possibly other biomolecules) from HCS and the circles they assembled over our nanoparticles to be protein coronas. Still, it is noteworthy that, after 12-h incubation in HCS-supplemented water, more dark dots were observed over PLGA@Lecithin than over PLGA@PEG, indicative of higher efficiency in adsorbing proteins for the former, consistent with the quantitative protein adsorption assays above (Figure 1c and Supplementary Figure 3b in the revised manuscript).

**Fig. 1** Selecting the nanoparticle efficient in removing vacuolating cytotoxin A (VacA). **a**. Schematic illustration on the pathways by which VacA secreted by *H. pylori* attacks gastric epithelial cells and their block due to VacA removal *via* protein adsorption onto externally added

nanoparticles. **b.** Schematic illustration on our model nanoparticles, PLGA@Lecithin and PLGA@PEG. **c.** Contents of total adsorbed proteins on PLGA@PEG and PLGA@Lecithin (both at 10 mg/mL) re-collected at different time-points after the initiation of their incubation in water supplemented with *H. pylori* culture supernatant (HCS) (10 mg/mL). **d.** Cryo-EM images of **(top)** PLGA@PEG and **(bottom)** PLGA@Lecithin (both at 30 mg/mL) re-collected after 12-h pre-incubation **(left)** in water and **(right)** in HCS-supplemented water (10 mg/mL). Scale bar = 50 nm. **e.** Photographs of SDS-PAGE gels with adsorbed proteins on PLGA@PEG and PLGA@Lecithin (80 mg/mL) with and without 12-h pre-incubation in HCS-supplemented water (10 mg/mL), with that of HCS-supplemented water (5 mg/mL) included as a reference. **f.** Contents of adsorbed VacA on PLGA@PEG and PLGA@Lecithin (8 mg/mL) re-collected after 24-h pre-incubation in HCS-supplemented water (20 mg/mL) and those of residual VacA in the as-treated solution. Bar heights are reported as averages of two independent trials (n = 3 in each independent trial). **g-i.** Viability ratios of human gastric adenocarcinoma cells (AGS cells) **(g)** after 24-h treatment with PLGA@PEG and PLGA@Lecithin at differing doses in PBS. **(h)** 24-h incubation in HCS-supplemented PBS at differing concentrations. **(i)** after 24-h incubation in HCS-supplemented PBS (20 mg/mL) that was pre-treated (for 12 h) with a nanoparticle (PLGA@PEG or PLGA@Lecithin) at differing doses. Bar height represents the average of two independent trials (n = 3 in each independent trial). *P* values are calculated using two-sided Student's *t*-test.

*4. The lecithin bilayer-coated nanoparticle was prepared by embedding the PLGA nanoparticle into the Lecithin vesicles under US (Supplementary Fig. 4, 7, 8, 9). How is the encapsulating efficiency? This coating efficiency will eventually determine the protein-adsorption results in Fig 3c.*

Author response: We gratefully thank the reviewer for the kind reminder. We took multiple sets of TEM images and counted the number of lecithin bilayer-coated nanoparticles as a proportion of the total particle number. As shown in the figure below, the number of nanoparticles with a membrane coating rate of  $\geq 80\%$  accounted for 64.7% of the total number of nanoparticles, of which 100% completely coated nanoparticles accounted for 59.1% of the total statistics, and more than the vast majority of nanoparticles were coated with better lecithin membrane.

**Statistics of lecithin bilayer-coated nanoparticles. a-b.** TEM images of (a) PLGA nanoparticle and (b) PLGA@Lecithin nanoparticle. **c.** Lecithin coating degree distribution of PLGA nanoparticles, which is calculated from TEM images ( $n = 252$  independent PLGA@Lecithin nanoparticles).

*5. All TEM images should present multiple nanoparticles to confirm the successful construction of a uniform core-shell structure.*

Author response: We gratefully thank the reviewer for the kind reminder. In revision, we have accordingly added TEM images showing multiple nanoparticles to confirm the successful construction of a uniform core-shell structure. Now, figures containing these TEM images appear:

**Supplementary Figure 1. PLGA@Lecithin preparation and characterization.** **a.** Schematic illustration on the preparation of PLGA@Lecithin nanoparticle. **b-c.** Transmission electron microscope (TEM) images of (b) PLGA nanoparticle and (c) PLGA@Lecithin nanoparticle. **d.** Counts of PLGA@Lecithin nanoparticles with varying lecithin membrane coating degrees among a total number of 252 individual PLGA@Lecithin nanoparticles in TEM images. The pie-chart (inset) indicates that the percentages of PLGA@Lecithin nanoparticles with membrane coating degrees of  $\geq 80\%$  and of  $< 80\%$  to be 64.7% and 35.3%, respectively. **e.** The hydrodynamic diameters and Zeta-potentials of PLGA@Lecithin in Millipore water, with those of the precursor PLGA nanoparticle and lecithin vesicle included for comparison. Bar heights are reported as average  $\pm$  standard deviation ( $n = 3$  independent experiments).

**Supplementary Figure 8. Ver-PLGA@Lecithin nanoparticle preparation and characterization.** **a.** Schematic illustration on the preparation of Ver-PLGA@Lecithin nanoparticle. **b-c.** TEM images of (b) Ver-PLGA nanoparticle and (c) Ver-PLGA@Lecithin nanoparticle. **d.** The hydrodynamic diameters and Zeta-potentials of Ver-PLGA@Lecithin in Millipore water, with those of the precursor Ver-PLGA nanoparticle and lecithin vesicle included for comparison. Bar heights are reported as average  $\pm$  standard deviation ( $n = 3$  in independent experiments).

**Supplementary Figure 9. Ce6-PLGA@Lecithin nanoparticle preparation and characterization.** **a.** Schematic illustration on the preparation of Ce6-PLGA@Lecithin nanoparticle. **b-c.** TEM images of (b) Ce6-PLGA nanoparticle and (c) Ce6-PLGA@Lecithin nanoparticle. **d.** The hydrodynamic diameters and Zeta-potentials of Ce6-PLGA@Lecithin in Millipore water, with those of the precursor Ce6-PLGA nanoparticle and lecithin vesicle included for comparison. Bar heights are reported as average  $\pm$  standard deviation ( $n = 3$  in independent experiments).

**Supplementary Figure 10. ICG-PLGA@Lecithin nanoparticle preparation and characterization.** **a.** Schematic illustration on the preparation of ICG-PLGA@Lecithin nanoparticle. **b-c.** TEM images of (b) ICG-PLGA nanoparticle and (c) ICG-PLGA@Lecithin nanoparticle. **d.** The hydrodynamic diameters and Zeta-potentials of ICG-PLGA@Lecithin in Millipore water, with those of the precursor ICG-PLGA nanoparticle and lecithin vesicle included for comparison. Bar heights are reported as average  $\pm$  standard deviation ( $n = 3$  in independent experiments).

6. TEM figures in Supplementary Fig. 8b and 9b, Fig. 7b and 5b, are the same particles. However, the author claimed they are different.

Author response: We gratefully thank the reviewer for the attention to details and for the kind reminder. In revision, we have accordingly checked the figures carefully to make sure that all the TEM images are in the right places they are supposed to be.

Now, TEM image in Supplementary Fig. 5 of the previous Supplementary Information (*i.e.*, Supplementary Figure 2 in the revised Supplementary Information) appears:

**Supplementary Figure 2. PLGA@PEG preparation and characterization.** a. Schematic illustration on the preparation of PLGA@PEG nanoparticle. b. TEM images of PLGA@PEG nanoparticle. c-d. The hydrodynamic diameter and Zeta-potential of PLGA@PEG nanoparticle in Millipore water. Bar heights are reported as average  $\pm$  standard deviation ( $n = 3$  independent experiments).

Now, TEM images in Supplementary Fig. 7-9 of the previous Supplementary Information (*i.e.*, Supplementary Figure 8-10 in the revised Supplementary Information) appear:

**Supplementary Figure 8. Ver-PLGA@Lecithin nanoparticle preparation and characterization.** **a.** Schematic illustration on the preparation of Ver-PLGA@Lecithin nanoparticle. **b-c.** TEM images of (b) Ver-PLGA nanoparticle and (c) Ver-PLGA@Lecithin nanoparticle. **d.** The hydrodynamic diameters and Zeta-potentials of Ver-PLGA@Lecithin in Millipore water, with those of the precursor Ver-PLGA nanoparticle and lecithin vesicle included for comparison. Bar heights are reported as average  $\pm$  standard deviation ( $n = 3$  in independent experiments).

**Supplementary Figure 9. Ce6-PLGA@Lecithin nanoparticle preparation and characterization.** **a.** Schematic illustration on the preparation of Ce6-PLGA@Lecithin nanoparticle. **b-c.** TEM images of (b) Ce6-PLGA nanoparticle and (c) Ce6-PLGA@Lecithin nanoparticle. **d.** The hydrodynamic diameters and Zeta-potentials of Ce6-PLGA@Lecithin in Millipore water, with those of the precursor Ce6-PLGA nanoparticle and lecithin vesicle included for comparison. Bar heights are reported as average  $\pm$  standard deviation ( $n = 3$  in independent experiments).

**Supplementary Figure 10. ICG-PLGA@Lecithin nanoparticle preparation and characterization.** **a.** Schematic illustration on the preparation of ICG-PLGA@Lecithin nanoparticle. **b-c.** TEM images of (b) ICG-PLGA nanoparticle and (c) ICG-PLGA@Lecithin nanoparticle. **d.** The hydrodynamic diameters and Zeta-potentials of ICG-PLGA@Lecithin in Millipore water, with those of the precursor ICG-PLGA nanoparticle and lecithin vesicle included for comparison. Bar heights are reported as average  $\pm$  standard deviation ( $n = 3$  in independent experiments).

7. Is the formation of protein corona affecting the ROS-generating efficacy?

Author response: We gratefully thank the reviewer for the critical comments. In revision, we have accordingly carried out additional experiments to check whether protein corona formation affects the ROS-generating efficacy of our model nano-sonosensitizer, Ver-PLGA@Lecithin, and consequently its *H. pylori* killing efficacy.

Firstly, by performing Bradford assays, we confirmed the formation of protein corona around our model nano-sonosensitizer, Ver-PLGA@Lecithin, after the nanoparticle's 12-h incubation under protein-present conditions (*i.e.*, in *H. pylori* culture supernatant-supplemented water, or in *H. pylori* culture supernatant-supplemented SGF) (Figure 3a in the revised manuscript).

Secondly, we checked whether protein corona formation affects the ROS-generating efficacy of Ver-PLGA@Lecithin, using DCFH as the fluorescent probe for ROS. To assure protein formation before exposure to ultrasound (US) for triggering ROS-generation, we incubated Ver-PLGA@Lecithin for 12 h in a protein-present medium (phosphate buffered saline (PBS) supplemented with *H. pylori* culture supernatant at 10 mg/mL, or simulated gastric fluid (SGF) supplemented with *H. pylori* culture supernatant at 10 mg/mL) and then separated the nanoparticles from free proteins (*via* centrifuge) to re-collect nanoparticles for subsequent ROS-generation detection, with PBS included as a protein-absent medium for experiments under comparable conditions. Our results (Figure 3b in the revised manuscript) show that nanoparticles re-collected after 12-h co-culture with either protein-present medium (PBS supplemented with *H. pylori* culture supernatant, or SGF supplemented with *H. pylori* culture supernatant) exhibited similar US-triggered generation of ROS as those collected after co-culture with PBS (Supplementary Figure 19a-c in the revised manuscript), suggesting negligible effects of protein-adsorption onto Ver-PLGA@Lecithin nanoparticle surface on the particle's ability to generate ROS in response to US exposure. Moreover, in US-exposed and re-collected nanoparticle-present media, the emission curves of DCFH almost overlapped with each other no matter whether the specific medium was protein-present or absent (Figure 3b in the revised manuscript), with only slight reduction in fluorescence intensity at 500-580 nm for the protein-present media compared to the protein-absent one (*i.e.*, PBS here), indicating slight impact on Ver-PLGA@Lecithin

nanoparticle's efficacy in US-triggered ROS-generation due to protein adsorption onto its surface. Collectively, our results show that the protein corona formation over Ver-PLGA@Lecithin nanoparticle neither deletes the particle's ability to generate ROS in response to US exposure nor impacts its ROS-generation efficacy to appreciable level.

Thirdly, we further checked whether protein corona formation affects the *H. pylori* killing potency of Ver-PLGA@Lecithin. To assure protein formation before exposure to ultrasound (US) for triggering ROS-generation, we co-cultured Ver-PLGA@Lecithin with a protein-present medium (PBS supplemented with *H. pylori* culture supernatant at 10 mg/mL, or SGF supplemented with *H. pylori* culture supernatant at 10 mg/mL) for 12 h and then separated the nanoparticles from free proteins (*via* centrifuge) to re-collect nanoparticles for subsequent *H. pylori* killing assays, with PBS included as a protein-absent medium for experiments under comparable conditions. Our results (Figure 3c in the revised manuscript) show that nanoparticles re-collected after 12-h co-culture with either of the two protein-present media above exhibited similar US-triggered killing of *H. pylori* and were necessary for the observed *H. pylori*, just as those collected after co-culture with PBS (Supplementary Figure 19d-f in the revised manuscript), suggesting negligible effects of protein-adsorption onto Ver-PLGA@Lecithin nanoparticle surface on the particle's ability to kill *H. pylori* in response to US exposure. Moreover, in the US-exposed and re-collected nanoparticle-present media, the *H. pylori* killing potency of Ver-PLGA@Lecithin almost overlapped with each other no matter whether the specific medium was protein-present or absent (Figure 3c in the revised manuscript), indicating negligible impact of protein corona formation over Ver-PLGA@Lecithin nanoparticle on the particle's potency of US-triggered killing against *H. pylori*. Collectively, our results show that the protein corona formation over Ver-PLGA@Lecithin nanoparticle neither deletes the particle's ability to kill *H. pylori* in response to US exposure nor impacts its *H. pylori* killing potency to detectable level.

**Figure 3. Effects of protein corona on Ver-PLGA@Lecithin's performances *in vitro*.** **a.** Contents of total adsorbed proteins on Ver-PLGA@Lecithin (100  $\mu\text{g}/\text{mL}$ , in Ver dose) re-collected *via* centrifuge after 12-h incubation in HCS-supplemented water or in HCS-supplemented SGF (both at 10  $\text{mg}/\text{mL}$ ), with those after 12-h incubation in water included for comparison. Bar heights are reported as average  $\pm$  standard deviation ( $n = 3$  in each independent trial).  $P$  values are calculated using two-sided Student's t-test. **b.** Fluorescence emission spectra of DCFH (40  $\mu\text{M}$ ) upon US exposure (0.5  $\text{W}/\text{cm}^2$ , 10 min) in the presence of Ver-PLGA@Lecithin (100  $\mu\text{g}/\text{mL}$  in Ver dose) re-collected *via* centrifuge after 12-h incubation in HCS-supplemented PBS or in HCS-supplemented SGF (both at 10  $\text{mg}/\text{mL}$ ), with that after 12-h incubation in PBS included for comparison. **c.** Counts of viable *H. pylori* cells (in Log CFU/mL) upon US exposure (0.5  $\text{W}/\text{cm}^2$ ; 10 min) after treatment with Ver-PLGA@Lecithin (100  $\mu\text{g}/\text{mL}$  in Ver dose) re-collected *via* centrifuge after 12-h incubation in HCS-supplemented PBS or in HCS-supplemented SGF (both at 10  $\text{mg}/\text{mL}$ ), with that after 12-h incubation in PBS included for comparison. Controls are those treated similarly but without US exposure. Bar height represents the average of two independent trials ( $n = 3$  in each independent trial).  $P$  values are calculated using two-sided Student's t-test. **d.** Schematic illustration on the negligible effects of protein corona formation over Ver-PLGA@Lecithin on the particle's efficiency in ROS generation and consequent *H. pylori* killing upon US exposure.

**Supplementary Figure 19. Effects of protein adsorption on Ver-PLGA@Lecithin's performances *in vitro*.** **a-c.** Fluorescence emission spectra of DCFH (100 µL, 40 µM) in the presence of US exposure (0.5 W/cm2, 10 min) and Ver-PLGA@Lecithin (100 µL, 0 or 100 µg/mL) re-collected after 12-h pre-incubation in (a) PBS, (b) HCS-supplemented PBS (10 mg/mL), or (c) HCS-supplemented SGF (10 mg/mL). Fluorescence emission spectra of samples treated similarly but without DCFH (100 µL, 40 µM) were included, to exclude the possibility of self-fluorescence from the nanoparticle and/or protein-present media. **d-f.** Counts of viable *H. pylori* cells (in Log CFU/mL) after treatment with Ver-PLGA@Lecithin (100 µL, 0 or 100 µg/mL) re-collected after 12-h pre-incubation in (d) PBS, (e) HCS-supplemented PBS (10 mg/mL), or (f) HCS-supplemented SGF (10 mg/mL) under US exposure (0.5 W/cm2, 10 min). Controls are samples treated similarly but without US exposure. Bar height represents the average of two independent trials (n = 3 in each independent trial). *P* values are calculated using two-sided Student's t-test.

*8. Although the manuscript has many figures (1-8), a lot of schemes are presented in the figures.*

Author response: We gratefully thank the reviewer for the kind reminder. In revision, we have accordingly revised our manuscript. Now, many schemes unnecessary for the main-text have been removed or replaced into the Supplementary Information.

*9. The lecithin chemical structure in Fig 4a lacks an O atom near the fatty acid chains.*

Author response: We gratefully thank the reviewer for the kind reminder. In revision, we have accordingly removed the chemical structure for lecithin, because lecithin is actually a mixture of multiple phospholipids. In the previous manuscript, we placed the chemical structure of the main component to represent lecithin, which in revision was realized to be unreasonable.

*10. The English expression should be briefer and to the point. The numbers of experiments are not very many, but the presentation words have exceeded the SI.*

Author response: We gratefully thank the reviewer for the kind reminder. In revision, we have accordingly revised the manuscript to make it more concise.

*Reviewer #3 (Remarks to the Author):*

*The manuscript submitted by Liu et al entitled "A safe and gut microbiota-friendly therapy for gastric Helicobacter pylori infection" reports a novel strategy for Helicobacter pylori eradication in mice.*

*The so called sonodynamic therapy reflects the increasing number of primary antibiotic resistances against frequently used antibiotics as metronidazole, clarithromycin and levofloxacin and also the temporary disturbance of gut microbiota after eradication treatment.*

*Helicobacter pylori infection affects more than the half worlds population and is associated with more than 90% of gastric cancer cases. In 2015, H.pylori infection was defined as infectious*

*disease with the need to eradicate irrespective of symptoms and complications.*

*In daily routine, virulence factors of the pathogen associated to distinct phenotypes and risks for complication are not used to stratify eradication treatment.*

*Unfortunately, there are some points in the manuscript submitted have to be discussed here:*

*1. The references used are not up to date and some relevant publications are missed. E.g.*

*Systematic review: gastric microbiota in health and disease. Rajilic-Stojanovic M, Figueiredo C, Smet A, Hansen R, Kupcinskas J, Rokkas T, Andersen L, Machado JC, Ianiro G, Gasbarrini A, Leja M, Gisbert JP, Hold GL. Aliment Pharmacol Ther. 2020 Mar;51(6):582-602. doi: 10.1111/apt.15650. Epub 2020 Feb 13. PMID: 32056247.*

*The interplay between Helicobacter pylori and gastrointestinal microbiota. Chen CC, Liou JM, Lee YC, Hong TC, El-Omar EM, Wu MS. Gut Microbes. 2021 Jan-Dec;13(1):1-22. doi: 10.1080/19490976.2021.1909459. PMID: 33938378.*

*Long-term changes of gut microbiota, antibiotic resistance, and metabolic parameters after Helicobacter pylori eradication: a multicentre, open-label, randomised trial. Liou JM, Chen CC, Chang CM, Fang YJ, Bair MJ, Chen PY, Chang CY, Hsu YC, Chen MJ, Chen CC, Lee JY, Yang TH, Luo JC, Chen CY, Hsu WF, Chen YN, Wu JY, Lin JT, Lu TP, Chuang EY, El-Omar EM, Wu MS; Taiwan Gastrointestinal Disease and Helicobacter Consortium. Lancet Infect Dis. 2019 Oct;19(10):1109-1120. doi: 10.1016/S1473-3099(19)30272-5. PMID: 31559966.*

*Second-line levofloxacin-based quadruple therapy versus bismuth-based quadruple therapy for Helicobacter pylori eradication and long-term changes to the gut microbiota and antibiotic resistome: a multicentre, open-label, randomised controlled trial. Liou JM, Jiang XT, Chen CC, Luo JC, Bair MJ, Chen PY, Chou CK, Fang YJ, Chen MJ, Chen CC, Lee JY, Yang TH, Yu CC, Kuo CC, Chiu MC, Chen CY, Shun CT, Hu WH, Tsai MH, Hsu YC, Tseng CH, Chang CY, Lin JT, El-Omar EM, Wu MS; Taiwan Gastrointestinal Disease and Helicobacter Consortium. Lancet Gastroenterol Hepatol. 2023 Mar;8(3):228-241. doi: 10.1016/S2468-*

1253(22)00384-3. Epub 2022 Dec 19.

PMID: 36549320.

*..and many others in the field. As the authors discussed, there are ongoing discussions regarding the effect of antibiotic treatment of H.pylori infection and (temporary) disturbances of gut microbial pattern. But, as shown in the publications of Liou and his group, the effect is only temporary and initial changes of alpha and beta diversity are recovered after one year. Please discuss.*

Author response: We gratefully thank the reviewer for the kind reminder and for suggesting references. In revision, we have accordingly performed literature research on the effects of gastric *H. pylori* infection on host gut microbiota (as summarized in Supplementary Dataset 2) and that on the effects of antibiotic-based *H. pylori* treatments on host gut microbiota (as summarized in Supplementary Dataset 1). Our results show that, shortly after antibiotic-based *H. pylori* eradication treatment, patients readily exhibit gut microbiota dysbiosis (Supplementary Dataset 1). Notably, such gut microbiota dysbiosis could be long-term, as some patients still exhibit significant changes in gut microbiota diversity and in the relative abundance for some gut commensal bacteria even at  $\geq 12$  months after antibiotic-based *H. pylori* eradication treatment (Supplementary Dataset 1: effects of antibiotic-based *H. pylori* treatments on host gut microbiota).

*2. The structure of the manuscript needs to be revised from my point of view. Reporting a novel and innovative antibacterial strategy the main focus have to be linked to this concept. The data presented about the effect a triple therapy on microbial dysbiosis might be an argument for novel strategies, but, on the one hand, are overcome in human studies as shown above and, on the other hand, the argument is explained in the introduction. The novelty has to be rated low.*

Author response: We gratefully thank the reviewer for the critical comments. In revision, we have accordingly revised the structure of the manuscript and shifted the focus of this work to be the development of a new treatment strategy. Specifically, we have combined the Results and Discussion section on how triple therapy affects gut microbiota with that on whether our sonodynamic therapy affects gut microbiota, with the triple therapy being used as a reference for

comparison with our sonodynamic therapy. Besides, we have combined the Results and Discussion section on how triple therapy affects serum level of IL-1RA with that on whether our sonodynamic therapy affects serum level of IL-1RA and placed the as-combined content into the Results and Discussion section on the bio-safety of our sonodynamic therapy, with the triple therapy being used as a reference for comparison with our sonodynamic therapy.

*The increased risk for the development of NAFLD as shown in the experiment can be presented in another manuscript, but the long term effect of these changes has to be studied before this conclusion can be made.*

Author response: We gratefully thank the reviewer for the critical comments. We agree with the reviewer that the increase in IL-1RA serum level in *H. pylori*-infected mice after triple therapy is interesting. Note that the applicability of IL-1RA as an independent biomarker for non-alcoholic steatohepatitis (NASH) (*J. Hepatol.*, 2012, 56,663-670), a severe type of NAFLD, is reported only by just one research article, rather than a widely-used practice in research or in clinics. Therefore, in revision, we have accordingly adopted the kind suggestions by both reviewers 1&3 and removed our previous statement that the observed increase in IL-1RA serum level in *H. pylori*-infected mice after triple therapy indicates increased risk of NAFLD.

Still, it should be noted that endogenous IL-1RA is a natural anti-inflammatory protein and its elevated level in blood is associated with diverse animal models of disease and with human diseases such as rheumatic diseases and chronic renal failure (*Ann. Rev. Immunol.*, 1998, 16, 27-55). Therefore, our observation that IL-1RA serum levels of the triple therapy group, but not those of the control group (*i.e.*, *H. pylori*-infected but treated with PBS), were significantly higher than those of the healthy group (Figure 3c-f in the previous manuscript) does suggest certain adverse effects after triple therapy, though what specific adverse effects or disease(s) the observed increase in IL-1RA serum level correspond to are unknown at the current stage. And it is noteworthy that, though five mouse models were included in each of these three treatment groups, the five corresponding data points exhibited a narrow distribution without any odd one, and this was the case for all these three treatment groups.

In revision, we have combined the Results and Discussion content on the IL-1RA serum levels after triple therapy with that after our sonodynamic therapy and moved the as-combined section as a part for the evaluation on bio-safety of our sonodynamic therapy, with the triple therapy being used as a reference for comparison. By doing this, we hope to arouse the audiences' attention to potential adverse effects after triple therapy from an extra-gastrointestinal perspective.

In fact, this work is not the first one that correlate gut microbiota dysbiosis after triple therapy with adverse effects from an extra-gastrointestinal perspective. A prior study shows that, for *H. pylori*-positive patients, 1-week triple therapy renders their *Bacteroidetes*-to-*Firmicutes* (B:F) ratios in gut microbiota significantly higher even at 3 months after the therapy than those before the therapy and, meanwhile, makes their plasma levels of active ghrelin significantly lower even at 3 months after the therapy than those before the therapy and that significant correlation is observed in their plasma level of active ghrelin and their change in B:F ratio (*BMJ Open Gastro.*, 2017, e000182). Considering that ghrelin is a multifunctional hormone that facilitates fat storage and regulates body weight (*Nature*, 2000, 407, 908-913) and a high value for the ghrelin blood test (greater than 650 pg/mL in human subjects) may indicate fasting, cachexia, and anorexia, the authors of this prior study think the observed increase in plasma level of active ghrelin suggests a “lean” phenotype for patients after triple therapy (*BMJ Open Gastro.*, 2017, e000182).

*3. Please explain more detailed the infection challenge of the model used, especially I couldn't found the strain used in your experiments.*

Author response: We gratefully thank the reviewer for the kind reminder. In revision, we have accordingly checked to make sure that the information on the strain of *H. pylori* used in this work, which is ATCC no. 43504 purchased from American Type Culture Collection (ATCC) (Virginia, USA), is clearly shown in the **MATERIALS**. It reads: “The *H. pylori* strain (ATCC no. 43504) used in this work was purchased from the American Type Culture Collection (ATCC) (Virginia, USA).”

Besides, we in revision have checked to make sure that detailed information the infection challenge of the model used in this work is clearly shown in the **MATERIALS AND METHODS** section

**(S29. Biosafety of sonodynamic therapy in mouse models bearing gastric *H. pylori* infection.).**

It reads: “Briefly, thirty C57BL/6J female mice (8-week-old, Shanghai Slack Laboratory Animal Technology Co., Ltd.) were randomly divided into 6 groups (n = 5 per group), with one group to stay uninfected throughout the whole assay (*i.e.*, the healthy group) while the rest five groups to be infected *via* gavage administration of *H. pylori* containing FBS-supplemented TSB (0.3 mL of  $\sim 1.0 \times 10^8$  CFU/mL for each mouse) every 48 hours for 4 times (on day 1, 3, 5, and 7, respectively; and fasting for 12 hours prior to bacterial inoculation in mice and gavage NaHCO3 (0.15 mL 0.2 M) at the first 10 minutes of the day of bacterial inoculation to increase gastric pH to improve survival and colonization rates of *H. pylori* without decomposing urea and establishing resistance to strong acids) and then be left alone for 2 weeks to let *H. pylori* infection get established.”

The establishment of mouse models of *H. pylori* infection is difficult, mainly determined by the mouse species and the bacterial subtype of *H. pylori*, such as *H. pylori* SS1 strain is more likely to colonize in gastric of C57BL/6J mice. In addition, the bacterial burden and infection time of gavage to the stomach of mice are also influencing factors for the successful establishment of the model, too few bacteria for gavage or short infection time can hardly successfully establish the model, and the excessive numbers of bacteria for gavage will cause the death of mice. However, so far, the establishment method of mouse models of *H. pylori* infection has been relatively mature, and many studies have successfully established models of *H. pylori* infection. We determined the establishment protocol of animal models based on previous literature (*Nat. Commun.* 2021, 12, 2085; *Vaccine* 2018, 36, 371–380), fasting for 12 hours prior to bacterial inoculation in mice and gavage NaHCO3 (0.15 mL 0.2 M) at the first 10 minutes of the day of bacterial inoculation to increase gastric pH to improve survival and colonization rates of *H. pylori* without decomposing urea and establishing resistance to strong acids. Mouse models of *H. pylori* infection were successfully established by gavage 0.3 mL of approximately  $10^8$  CFU/mL of bacterial suspension every 48 hours for four times and after two weeks of continuous infection.

*To summarize, I recommend to focus on the novel and important antibacterial strategy and the methods used. The arguments for the need of novel strategies after WHO statement are explained in the introduction and do not have been proven again in the experiments (Fig 1 and 2), especially when using only short periods of follow-up.*

Author response: We gratefully thank the reviewer for the inspiring suggestion. In revision, we have accordingly revised the structure of the manuscript and shifted the focus of this work to be the development of a new antibacterial strategy and the methods used.

*\*\* See Nature Portfolio's author and referees' website at [www.nature.com/authors](http://www.nature.com/authors) for information about policies, services and author benefits.*

*This email has been sent through the Springer Nature Tracking System NY-610A-NPG&MTS*

*Confidentiality Statement:*

*This e-mail is confidential and subject to copyright. Any unauthorised use or disclosure of its contents is prohibited. If you have received this email in error please notify our Manuscript Tracking System Helpdesk team at <http://platformsupport.nature.com>.*

*Details of the confidentiality and pre-publicity policy may be found here <http://www.nature.com/authors/policies/confidentiality.html>*

## REVIEWER COMMENTS

### Reviewer #1 (Remarks to the Author):

With the revised version of the manuscript the authors have provided a detailed response to previous critiques and have significantly improved the manuscript. Only a few minor points should still be addressed.

1. In response to the previous critique, high magnification H&E images were included in Figure 5 to demonstrate that the sonodynamic therapy did not cause epithelial damage. However, the areas chosen for the high magnification images are not very informative, since they only show the bases of the glands, smooth muscle, and/or white space (VerPLGA@Lecithin w/o US). Images that show the entire glands or the luminal portions would be preferable – such as the ones shown in Fig. 22 with an increased magnification. The authors state “We found that the infiltration of inflammatory cells in gastric mucosa for the “Ver-PLGA@Lecithin + US” group was significantly lower than that for the “control” group but comparable to that for the “triple therapy” group” (lines 315-317). However, I was unable to detect any major inflammatory infiltrates in the images provided in either Fig. 5 or Supp. Fig. 22, and a statistical analysis of histopathological changes was not provided. Please explain.

2. The TUNEL assay (Supplemental Fig. 23) is a good alternative for analyzing cytotoxic treatment effects, but the results provided are difficult to interpret, since no TUNEL positive cells appear to be present at all. In the normal stomach, 2% of cells are expected to be apoptotic due to normal cell turnover (von Herbay and Rudi, *Microscopy Research and Technique*, 2000). As with the H&E stained samples, including images with a higher magnification may be useful. Therefore, the question whether increased cell death occurs has not yet been sufficiently addressed.

3. Figure 3 c,f, and g and Fig. 2l still do not show statistical analyses.

4. Fig. 3i (and h), Fig 5, and Fig. 7: As previously mentioned, a T test is not appropriate for comparing a range of different concentrations or multiple different treatment groups - please use ANOVA with a post hoc test or mixed modeling. A T test should only be used when comparing the means of two groups.

5. Fig. 5c,d,e: Include the P values for the comparisons between control and Ver-PLGA@Lecithin in this main figure rather than presenting the same figures again in the supplemental data with additional P values (Supplemental Fig. 21). Avoid showing the same data shown in main figures again

in slightly different form in the supplemental figures (e.g. left half of Fig. 5f is the same as left half of supplemental Fig. 22, many of the panels from Fig 7 are repeated in Suppl Fig 29, etc.). This will reduce the high number of supplemental figures for this study.

6. Lines 294-296: “Ver-PLGA@Lecithin alone suppressed the *H. pylori* infection-induced up-regulation of TNF- $\alpha$  ( $0.01 < P < 0.05$ ) (Fig. 5c-e and Supplementary Fig. 21), likely owing to its ability to remove VacA (Fig. 1f)”. Please temper this statement (and similar statements in the following sentence, the abstract, and elsewhere in the manuscript), since the impact of VacA absorption to the nanoparticle in vivo has not been investigated. I suggest replacing “likely” with “possibly”.

7. Please make sure that Figures including supplementary figures are numbered in the order of their appearance in the text.

Reviewer #2 (Remarks to the Author):

The authors addressed all my points raised before. I would recommend the manuscript to be accepted as it is.

## ***Responses to Reviewer Comments***

# **A safe and gut microbiota-friendly therapy for gastric *Helicobacter pylori* infection**

*Tao Liu*1,2,3, *Shuang Chai*1,2,3, *Mingyang Li*1,2,3, *Xu Chen*1,2,3, *Yutao Xie*1,2,3, *Zehui Zhao*1,2,3,  
*Jingjing Xie*1,2,3, *Yunpeng Yu*1,2,3, *Feng Gao*1, *Feng Zhu*4, *Lihua Yang*1,2,3\*

\*\*\*\*\*

### ***REVIEWER COMMENTS***

*Reviewer #1.*

*With the revised version of the manuscript the authors have provided a detailed response to previous critiques and have significantly improved the manuscript. Only a few minor points should still be addressed.*

*1. In response to the previous critique, high magnification H&E images were included in Figure 5 to demonstrate that the sonodynamic therapy did not cause epithelial damage. However, the areas chosen for the high magnification images are not very informative, since they only show the bases of the glands, smooth muscle, and/or white space (VerPLGA@Lecithin w/o US). Images that show the entire glands or the luminal portions would be preferable – such as the ones shown in Fig. 22 with an increased magnification.*

Author response: We gratefully thank the reviewer for the critical comments. In revision, to show the entire glands or the luminal portions as suggested by the reviewer, we have accordingly replaced the high magnification H&E images in Figure 5f of the previous manuscript with those in Supplementary Figure 22 of the previous manuscript, which led to Figure 6j in the revised manuscript. Meanwhile, to avoid repeated presentation of the same images, we have accordingly removed Supplementary Figure 22 of the previous manuscript. Now, these H&E images on gastric mucosa (Figure 6j in the revised manuscript) appear.

**Fig. 6 Efficacy of Ver-PLGA@Lecithin mediated sonodynamic therapy.** **a.** Schematic illustrations on the triple therapy regimen used for treating gastric *H. pylori* infection **b.** Schematic illustration on the schedule of different treatments including sonodynamic therapy (*i.e.*, “Ver-PLGA@Lecithin + US”) to mouse models bearing gastric *H. pylori* infection and follow-up blood and tissue collection. **c.** *H. pylori* burden per gram of stomach tissues (in Log CFU/g) from mice treated with “Ver-PLGA@Lecithin + US” group, with those from mice treated with PBS (*i.e.*, control), triple therapy, US exposure alone, and Ver-PLGA@Lecithin alone included for comparison. In the box plot, the upper and lower quartile as outlined by top and bottom boundary were divided by line showing the median value (n = 5 biologically independent mice in once trial). Statistical analysis was carried out with a one-way ANOVA with Tukey’s multiple-comparison test. **d-f.** Serum levels of (d) interleukin-6 (IL-6), (e) tumor necrosis factor- $\alpha$  (TNF- $\alpha$ ), and (f) interleukin-1 beta (IL-1 $\beta$ ) in gastric *H. pylori* infection-bearing mouse models after differing treatment. Bar heights are reported as average  $\pm$  standard deviation (n = 5 biologically independent mice in once trial). Statistical analysis was carried out with a one-way ANOVA with Tukey’s multiple-comparison test. **g-i.** Serum levels of (g) pepsinogen I (PG-1), (h) pepsinogen II (PG-2) and (i) gastrin-17 (G-17) in gastric *H. pylori* infection-bearing mouse models at 48-h after differing treatment. Bar heights are reported as average  $\pm$  standard deviation (n = 5 biologically independent mice in once trial). Red dash-line indicates the average serum level in healthy mice. Statistical analysis was carried out with a one-way ANOVA with Tukey’s multiple-comparison test. **j.** Microscopy images of hematoxylin and eosin (H&E) stained stomach tissues collected from gastric *H. pylori* infection-bearing mouse models at 48-h after differing treatment, with those of H&E stained stomach tissues collected from healthy counterparts included as references.

*The authors state “We found that the infiltration of inflammatory cells in gastric mucosa for the “Ver-PLGA@Lecithin + US” group was significantly lower than that for the “control” group but comparable to that for the “triple therapy” group” (lines 315-317). However, I was unable to detect any major inflammatory infiltrates in the images provided in either Fig. 5 or Supp. Fig. 22, and a statistical analysis of histopathological changes was not provided. Please explain.*

Author response: We gratefully thank the reviewer for the critical comments and for attention to details. In revision, we have accordingly rephrased the statement noted by the reviewer. It now reads:

“... Images on H&E stained tissues revealed that the stomach tissues from mice treated with our sonodynamic therapy (*i.e.*, the “Ver-PLGA@Lecithin + US” group) appeared similar as those from their healthy counterparts (*i.e.*, the healthy group) (Fig. 6j). ...”

Moreover, we have re-examined the same TUNEL stained stomach tissue samples but at higher magnification and over 50 different regions for samples from each treatment group, which offered us fluorescence images of clearer view especially for green fluorescence signals and also enabled us to perform statistical analysis on the percentage of green fluorescence intensity (Supplementary Figure 19). In our images of TUNEL stained tissues, green fluorescence indicates apoptotic cell death while blue fluorescence indicates DNA and consequently all cells. So, for an image of TUNEL stained tissue, we can estimate its relative ratio of apoptotic cells to all cells by calculating its percentage of green fluorescence intensity relative to the intensities of both green and blue fluorescence signals with the following function:

$$\text{percentage of green fluorescence intensity} = I_{\text{Green}} / (I_{\text{Green}} + I_{\text{Blue}})$$

where  $I_{\text{Green}}$  and  $I_{\text{Blue}}$  represent the mean intensity of green fluorescence and the mean intensity of blue fluorescence, respectively.

**Supplementary Figure 19. Analysis on TUNEL stained stomach tissues.** **a.** Representative fluorescence microscopy images of TUNEL (TdT-mediated dUTP Nick-End Labeling) stained stomach tissues collected from gastric *H. pylori* infection-bearing mouse models at 48-h after differing treatments (*i.e.*, control, triple therapy, US exposure alone, Ver-PLGA@Lecithin alone, and “Ver-PLGA@Lecithin + US”), with those from their healthy counterparts included as references. Blue fluorescence (by DAPI) indicates DNA, and green fluorescence (by FITC- dUTP) indicates apoptotic cell death. **b.** Percentages of green fluorescence intensity in (a) the images of TUNEL stained tissue samples, which were obtained by imaging 50 different regions for each

treatment group. Bar heights are reported as average  $\pm$  standard deviation. Statistical analysis was carried out with a one-way ANOVA with Tukey's multiple-comparison test.

Our results show that, for the healthy group, the percentages of green fluorescence intensity were within the range of 0.48-1.54%, which are slightly lower than the expected percentage of apoptotic cells (*i.e.*, 2%) in normal stomach due to normal cell turnover (von Herbay and Rudi, *Microscopy Research and Technique*, 2000) as mentioned by the reviewer latter. In the TUNEL staining images, the control group exhibited significantly higher percentages of green fluorescence intensity (0.79-2.14%) than the healthy group ( $P < 0.0001$ ), indicative of increased cell apoptosis due to gastric *H. pylori* infection. To our surprise, the triple therapy group exhibited further higher percentages of green fluorescence intensity (2.56-4.56%,  $P < 0.0001$ ), suggesting deterioration rather than mitigation of cell apoptosis after *H. pylori* eradication with triple therapy, possibly due to triple therapy itself and its neglect of VacA, a cytotoxin secreted by *H. pylori* into the extracellular environment and also a major virulence factor in the colonization and infection of *H. pylori* in stomach. In stark contrast, our sonodynamic therapy (*i.e.*, the "Ver-PLGA@Lecithin + US" group) reduced the percentages of green fluorescence intensity (0.58-2.95%) and rendered them totally comparable to those of the healthy group ( $P > 0.3$ ), indicative of re-normalized cell apoptosis in stomach tissues after our sonodynamic therapy. Similar as our sonodynamic therapy, Ver-PLGA@Lecithin alone (*i.e.*, the "Ver-PLGA@Lecithin" group) reduced the percentages of green fluorescence intensity (0.39-2.18%) and rendered them totally comparable to those of the healthy group ( $P > 0.9$ ), possibly owing to the particle's ability to remove VacA (Figure 2); nevertheless, ultrasound exposure alone (*i.e.*, the "US" group) failed to do so (0.86-3.20%,  $P < 0.0001$  compared to the healthy group), indicative of inability of US exposure alone to re-normalize the cell apoptosis in stomach tissues. Collectively, these results suggest that the observed ability of our sonodynamic therapy to re-normalize the cell apoptosis level in stomach tissue is attributable to both the therapy's ability to eradicate *H. pylori* cells and the ability of Ver-PLGA@Lecithin to remove VacA (Figure 2), which again underscores the importance of VacA removal in gastric *H. pylori* infection treatment.

*2. The TUNEL assay (Supplemental Fig. 23) is a good alternative for analyzing cytotoxic treatment effects, but the results provided are difficult to interpret, since no TUNEL positive cells appear to be present at all. In the normal stomach, 2% of cells are expected to be apoptotic due to normal cell turnover (von Herbay and Rudi, Microscopy Research and Technique, 2000). As with the H&E stained samples, including images with a higher magnification may be useful. Therefore, the question whether increased cell death occurs has not yet been sufficiently addressed.*

Author response: We gratefully thank the reviewer for the critical comments. In revision, we have accordingly re-examined the same TUNEL stained stomach tissue samples (Supplementary Figure 23 in the previous Supplementary Information file) but at higher magnification, which offered us fluorescence images of clearer view especially for green fluorescence signals (Supplementary Figure 19 in the revised Supplementary Information file). The resulting high magnification TUNEL images revealed the presence of green fluorescence (which indicates apoptotic cells) for all these samples no matter whether they were collected from the healthy mouse models or from their infected and differently treated counterparts. To provide clearer view of the fluorescence signals, we have accordingly replaced the high magnification TUNEL images in the previous

Supplementary Figure 23 with our newly obtained high magnification TUNEL images (as shown in Supplementary Figure 19 in the revised Supplementary Information file).

**Supplementary Figure 19. Analysis on TUNEL stained stomach tissues.** **a.** Representative fluorescence microscopy images of TUNEL (TdT-mediated dUTP Nick-End Labeling) stained stomach tissues collected from gastric *H. pylori* infection-bearing mouse models at 48-h after differing treatments (*i.e.*, control, triple therapy, US exposure alone, Ver-PLGA@Lecithin alone, and “Ver-PLGA@Lecithin + US”), with those from their healthy counterparts included as references. Blue fluorescence (by DAPI) indicates DNA, and green fluorescence (by FITC- dUTP) indicates apoptotic cell death. **b.** Percentages of green fluorescence intensity in (a) the images of TUNEL stained tissue samples, which were obtained by imaging 50 different regions for each

treatment group. Bar heights are reported as average  $\pm$  standard deviation. Statistical analysis was carried out with a one-way ANOVA with Tukey's multiple-comparison test.

While doing the re-examination, we imaged 50 different regions for samples from each treatment group, which enabled us to do statistical analysis on the percentage of green fluorescence intensity (Supplementary Figure 19). In our images of TUNEL stained tissues, green fluorescence indicates apoptotic cell death while blue fluorescence indicates DNA and consequently all cells. So, for an image of TUNEL stained tissue, we can estimate its relative ratio of apoptotic cells to all cells by calculating its percentage of green fluorescence intensity relative to the intensities of both green and blue fluorescence signals with the following function:

$$\text{percentage of green fluorescence intensity} = I_{\text{Green}} / (I_{\text{Green}} + I_{\text{Blue}})$$

where  $I_{\text{Green}}$  and  $I_{\text{Blue}}$  represent the mean intensity of green fluorescence and the mean intensity of blue fluorescence, respectively.

Our results show that, for the healthy group, the percentages of green fluorescence intensity were within the range of 0.48-1.54%, which are slightly lower than the expected percentage of apoptotic cells (*i.e.*, 2%) in normal stomach due to normal cell turnover (von Herbay and Rudi, *Microscopy Research and Technique*, 2000) as mentioned by the reviewer latter. In the TUNEL staining images, the control group exhibited significantly higher percentages of green fluorescence intensity (0.79-2.14%) than the healthy group ( $P < 0.0001$ ), indicative of increased cell apoptosis due to gastric *H. pylori* infection. To our surprise, the triple therapy group exhibited further higher percentages of green fluorescence intensity (2.56-4.56%,  $P < 0.0001$ ), suggesting deterioration rather than mitigation of cell apoptosis after *H. pylori* eradication with that triple therapy, possibly due to triple therapy itself and its neglect of VacA, a cytotoxin secreted by *H. pylori* into the extracellular environment and also a major virulence factor in the colonization and infection of *H. pylori* in stomach. In stark contrast, our sonodynamic therapy (*i.e.*, the "Ver-PLGA@Lecithin + US" group) reduced the percentages of green fluorescence intensity (0.58-2.95%) and rendered them totally comparable to those of the healthy group ( $P > 0.3$ ), indicative of re-normalized cell apoptosis in stomach tissues after our sonodynamic therapy. Similar as our sonodynamic therapy, Ver-PLGA@Lecithin alone (*i.e.*, the "Ver-PLGA@Lecithin" group) reduced the percentages of green fluorescence intensity (0.39-2.18%) and rendered them totally comparable to those of the healthy group ( $P > 0.9$ ) as well, possibly owing to the particle's ability to remove VacA (Figure 2); nevertheless, ultrasound exposure alone (*i.e.*, the "US" group) failed to do so (0.86-3.20%,  $P < 0.0001$  compared to the healthy group), indicative of inability of US exposure alone to re-normalize the cell apoptosis in stomach. Collectively, these results suggest that the observed ability of our sonodynamic therapy to re-normalize the cell apoptosis level in stomach is attributable to the ability of Ver-PLGA@Lecithin to remove VacA (Figure 2), which again underscores the importance of VacA removal in gastric *H. pylori* infection treatment.

3. Figure 3 c,f, and g and Fig. 2l still do not show statistical analyses.

Author response: We gratefully thank the reviewer for the kind suggestion. In revision, we have accordingly added statistical analysis to Fig. 2l of the previous manuscript (*i.e.*, Fig. 3l in the revised manuscript). In the previous manuscript, Fig. 3c has statistical analysis added and Fig. 3f mentioned by the reviewer does not exist. So, we think the reviewer may mean to say Fig. 1 c,f, but, because of typo, have put down Fig. 3 c,f. In revision, we have also added statistical analysis

to Fig. 1 c,f of the previous manuscript (*i.e.*, Fig. 2 c,f in the revised manuscript). Now, these figures appear.

**Fig. 2 Selecting the nanoparticle efficient in removing vacuolating cytotoxin A (VacA).** **a.** Schematic illustration on the pathways by which VacA secreted by *H. pylori* attacks gastric epithelial cells and their block due to VacA removal *via* protein adsorption onto externally added nanoparticles. **b.** Schematic illustration on our model nanoparticles, PLGA@Lecithin and PLGA@PEG. **c.** Contents of total adsorbed proteins on PLGA@PEG and PLGA@Lecithin (both at 10 mg/mL) re-collected at different time-points after the initiation of their incubation in water supplemented with *H. pylori* culture supernatant (HCS) (10 mg/mL). Statistical analysis was carried out with a two-way analysis of variance (ANOVA) with Tukey's multiple-comparison test. **d.** Cryo-EM images of **(top)** PLGA@PEG and **(bottom)** PLGA@Lecithin (both at 30 mg/mL) re-collected after 12-h pre-incubation **(left)** in water and **(right)** in HCS-supplemented water (10 mg/mL). Scale bar = 50 nm. **e.** Photographs of SDS-PAGE gels with absorbed proteins on PLGA@PEG and PLGA@Lecithin (80 mg/mL) with and without 12-h pre-incubation in HCS-supplemented water (10 mg/mL), with that of HCS-supplemented water (5 mg/mL) included as a reference. **f.** Contents of adsorbed VacA on PLGA@PEG and PLGA@Lecithin (8 mg/mL) re-collected after 24-h pre-incubation in HCS-supplemented water (20 mg/mL) and those of residual VacA in the as-treated solution. Bar heights are reported as averages of two independent trials (n = 3 in each independent trial). Statistical analysis was carried out with a one-way ANOVA with Tukey's multiple-comparison test. **g-i.** Viability ratios of human gastric adenocarcinoma cells (AGS cells) **(g)** after 24-h treatment with PLGA@PEG and PLGA@Lecithin at differing doses in PBS. **(h)** 24-h incubation in HCS-supplemented PBS at differing concentrations **(i)** after 24-h incubation in HCS-supplemented PBS (20 mg/mL) that was pre-treated (for 12 h) with a nanoparticle (PLGA@PEG or PLGA@Lecithin) at differing doses. Bar height represents the average of two independent trials (n = 3 in each independent trial). Statistical analysis was carried out with a (h) one-way and (i) two-way ANOVA with Tukey's multiple-comparison test.

**Fig. 3** *In vitro* performances of our nano-sonosensitizers. **a.** Schematic illustration on the detection of reactive oxygen species (ROS) generated by a USS-PLGA@Lecithin in response to ultrasound (US) exposure. **b.** Fluorescence intensity of DCFH solution with a USS-PLGA@Lecithin (100  $\mu\text{g/mL}$  in USS dose) or its empty counterpart PLGA@Lecithin upon US exposure (at 0.5  $\text{W/cm}^2$  for 10 min). DCFH solution without any nanoparticle was included as a blank. **c-e.** Fluorescence intensity of DCFH solution containing a USS-PLGA@Lecithin (c-d) at a fixed nanoparticle concentration (100  $\mu\text{g/mL}$  in USS dose) upon US exposure (c) at differing output power density (for 10 min) and (d) for differing exposure time (at 0.5  $\text{W/cm}^2$ ) and (e) at differing nanoparticle concentration (in USS dose) upon US exposure (at 0.5  $\text{W/cm}^2$  for 10 min). **f.** Fluorescence intensity of SOSG solution containing a USS-PLGA@Lecithin at differing

concentration (in USS dose) upon US exposure (at 0.5 W/cm2 for 10 min). **g.** Schematic illustration on the *in vitro* antibacterial assays against *H. pylori* using a USS-PLGA@Lecithin. **h.** Survival *H. pylori* cell counts (in Log CFU/mL, where CFU means colony forming unit) after treatment with a USS-PLGA@Lecithin at differing concentration (in USS dose) in PBS (right) with and (left) without US exposure (at 0.5 W/cm2 for 10 min). **i.** Survival *H. pylori* cell counts (in Log CFU/mL) after treatment with Ver-PLGA@Lecithin in PBS upon US exposure (left) at differing output power density (for 10 min) and (right) for differing exposure time (at 0.5 W/cm2). **j.** Scanning electron microscopy (SEM) images of *H. pylori* cells after different treatments. Ver-PLGA@Lecithin dose was fixed at 1 mg/mL in PBS, and US exposure was at 0.5 W/cm2 for 10 min. **k-l.** Survival *H. pylori* cell counts (in Log CFU/mL) after treatment with Ver-PLGA@Lecithin at differing concentration (in Ver dose) (k) in  $\beta$ -carotene supplemented (1.5 mg/mL) PBS and (l) in simulated gastric fluid (SGF) in the presence and absence of US exposure (at 0.5 W/cm2 for 10 min). Bar height represents the average of two independent trials (n = 3 in each independent trial). Statistical analysis was carried out with a two-way ANOVA with Tukey's multiple-comparison test.

4. Fig. 3i (and h), Fig 5, and Fig. 7: As previously mentioned, a T test is not appropriate for comparing a range of different concentrations or multiple different treatment groups - please use ANOVA with a post hoc test or mixed modeling. A T test should only be used when comparing the means of two groups.

Author response: We gratefully thank the reviewer for his/her patience and for pointing out this. In revision, we have accordingly looked more into the details of these differing tests including the differences between them. We agree with the reviewer that a T test is not appropriate for comparing a range of different concentrations or multiple different treatment group. So, in revision, we have accordingly double-checked all the statistical analysis-involved figures throughout the manuscript (including those in the Supplementary Information file) and, for all those where a T test is not appropriate, have re-done the statistical analysis there with an ANOVA with Tukey's multiple-comparison test and have added the specific tests used into the corresponding figure captions.

Figures that have been updated with *P* values got through statistical analysis there with an ANOVA with Tukey's multiple-comparison test include: Fig. 1 of the previous manuscript (*i.e.*, Fig. 2 in the revised manuscript); Fig. 2 of the previous manuscript (*i.e.*, Fig. 3 in the revised manuscript); Fig. 3 of the previous manuscript (*i.e.*, Fig. 4 in the revised manuscript); Fig. 5 of the previous manuscript (*i.e.*, Fig. 6 in the revised manuscript); Fig. 6 of the previous manuscript (*i.e.*, Fig. 7 in the revised manuscript); Fig. 7 of the previous manuscript (*i.e.*, Fig. 8 in the revised manuscript); Supplementary Fig. 3 of the previous manuscript (*i.e.*, Supplementary Fig. 3 in the revised manuscript); Supplementary Fig. 5 of the previous manuscript (*i.e.*, Supplementary Fig. 5 in the revised manuscript); Supplementary Fig. 6 of the previous manuscript (*i.e.*, Supplementary Fig. 6 in the revised manuscript); Supplementary Fig. 15 of the previous manuscript (*i.e.*, Supplementary Fig. 14 in the revised manuscript); Supplementary Fig. 17 of the previous manuscript (*i.e.*, Supplementary Fig. 17 in the revised manuscript); Supplementary Fig. 18 of the previous manuscript (*i.e.*, Supplementary Fig. 18 in the revised manuscript); Supplementary Fig. 19 of the previous manuscript (*i.e.*, Supplementary Fig. 15 in the revised manuscript); Supplementary Fig. 24 of the previous manuscript (*i.e.*, Supplementary Fig. 20 in the revised manuscript); Supplementary Fig. 29 of the previous manuscript (*i.e.*, Supplementary Fig. 22 in the revised

manuscript); Supplementary Fig. 31 of the previous manuscript (*i.e.*, Fig. 9 in the revised manuscript).

The above list of updated figures includes the previous Fig. 5 and Fig. 7 mentioned here by the reviewer. In the previous manuscript, Fig. 3 does not contain panels Fig. 3i and Fig. 3h mentioned here by the reviewer. We think the reviewer may mean to say Fig. 1i and Fig. 1h but, because of typo, have put down Fig. 3i (and h). So, in revision, we have re-done statistical analysis for Fig. 1i and Fig. 1h of the previous manuscript (*i.e.*, Fig. 2i and Fig. 2h in the revised manuscript) but with a (Fig. 2h) one-way and (Fig. 2i) two-way ANOVA with Tukey's multiple-comparison test, respectively.

Now, Fig. 1 of the previous manuscript (*i.e.*, Fig. 2 in the revised manuscript), Fig. 5 of the previous manuscript (*i.e.*, Fig. 6 in the revised manuscript), and Fig. 7 of the previous manuscript (*i.e.*, Fig. 8 in the revised manuscript) appear.

**Fig. 2** Selecting the nanoparticle efficient in removing vacuolating cytotoxin A (VacA). **a.** Schematic illustration on the pathways by which VacA secreted by *H. pylori* attacks gastric epithelial cells and their block due to VacA removal *via* protein adsorption onto externally added nanoparticles. **b.** Schematic illustration on our model nanoparticles, PLGA@Lecithin and

PLGA@PEG. **c.** Contents of total adsorbed proteins on PLGA@PEG and PLGA@Lecithin (both at 10 mg/mL) re-collected at different time-points after the initiation of their incubation in water supplemented with *H. pylori* culture supernatant (HCS) (10 mg/mL). Statistical analysis was carried out with a two-way analysis of variance (ANOVA) with Tukey's multiple-comparison test. **d.** Cryo-EM images of (**top**) PLGA@PEG and (**bottom**) PLGA@Lecithin (both at 30 mg/mL) re-collected after 12-h pre-incubation (**left**) in water and (**right**) in HCS-supplemented water (10 mg/mL). Scale bar = 50 nm. **e.** Photographs of SDS-PAGE gels with absorbed proteins on PLGA@PEG and PLGA@Lecithin (80 mg/mL) with and without 12-h pre-incubation in HCS-supplemented water (10 mg/mL), with that of HCS-supplemented water (5 mg/mL) included as a reference. **f.** Contents of adsorbed VacA on PLGA@PEG and PLGA@Lecithin (8 mg/mL) re-collected after 24-h pre-incubation in HCS-supplemented water (20 mg/mL) and those of residual VacA in the as-treated solution. Bar heights are reported as averages of two independent trials (n = 3 in each independent trial). Statistical analysis was carried out with a one-way ANOVA with Tukey's multiple-comparison test. **g-i.** Viability ratios of human gastric adenocarcinoma cells (AGS cells) (**g**) after 24-h treatment with PLGA@PEG and PLGA@Lecithin at differing doses in PBS. (**h**) 24-h incubation in HCS-supplemented PBS at differing concentrations (**i**) after 24-h incubation in HCS-supplemented PBS (20 mg/mL) that was pre-treated (for 12 h) with a nanoparticle (PLGA@PEG or PLGA@Lecithin) at differing doses. Bar height represents the average of two independent trials (n = 3 in each independent trial). Statistical analysis was carried out with a (h) one-way and (i) two-way ANOVA with Tukey's multiple-comparison test.

**Fig. 6 Efficacy of Ver-PLGA@Lecithin mediated sonodynamic therapy.** **a.** Schematic illustrations on the triple therapy regimen used for treating gastric *H. pylori* infection **b.** Schematic illustration on the schedule of different treatments including sonodynamic therapy (*i.e.*, “Ver-PLGA@Lecithin + US”) to mouse models bearing gastric *H. pylori* infection and follow-up blood and tissue collection. **c.** *H. pylori* burden per gram of stomach tissues (in Log CFU/g) from mice treated with “Ver-PLGA@Lecithin + US” group, with those from mice treated with PBS (*i.e.*, control), triple therapy, US exposure alone, and Ver-PLGA@Lecithin alone included for comparison. In the box plot, the upper and lower quartile as outlined by top and bottom boundary were divided by line showing the median value (n = 5 biologically independent mice in once trial). Statistical analysis was carried out with a one-way ANOVA with Tukey’s multiple-comparison test. **d-f.** Serum levels of (d) interleukin-6 (IL-6), (e) tumor necrosis factor- $\alpha$  (TNF- $\alpha$ ), and (f) interleukin-1 beta (IL-1 $\beta$ ) in gastric *H. pylori* infection-bearing mouse models after differing treatment. Bar heights are reported as average  $\pm$  standard deviation (n = 5 biologically independent mice in once trial). Statistical analysis was carried out with a one-way ANOVA with Tukey’s multiple-comparison test. **g-i.** Serum levels of (g) pepsinogen I (PG-1), (h) pepsinogen II (PG-2) and (i) gastrin-17 (G-17) in gastric *H. pylori* infection-bearing mouse models at 48-h after differing treatment. Bar heights are reported as average  $\pm$  standard deviation (n = 5 biologically independent mice in once trial). Red dash-line indicates the average serum level in healthy mice. Statistical analysis was carried out with a one-way ANOVA with Tukey’s multiple-comparison test. **j.** Microscopy images of hematoxylin and eosin (H&E) stained stomach tissues collected from gastric *H. pylori* infection-bearing mouse models at 48-h after differing treatment.

**Fig. 8 Biosafety of our sonodynamic therapy.** **a.** Schematic illustration on the schedule of differing treatment to mouse models bearing gastric *H. pylori* infection and follow-up blood collection for liver and renal function tests and complete blood count (CBC) test. **b-e.** Serum levels

of (b) alanine aminotransferase (ALT), (c) aspartate aminotransferase (AST), (d) creatinine (CREA), and (e) blood urea nitrogen (BUN). **f-m.** Blood levels of (f) red blood cell count (RBC), (g) hemoglobin, (h) white blood cell count (WBC), (i) eosinophil, (j) lymphocyte, (k) neutrophils, (l) monocyte, and (m) platelet. Red dash-line indicates the average serum level in healthy mice. Bar heights are reported as average  $\pm$  standard deviation ( $n = 5$  biologically independent mice in once trial). Statistical analysis was carried out with a one-way ANOVA with Tukey's multiple-comparison test.

*5. Fig. 5c,d,e: Include the P values for the comparisons between control and Ver-PLGA@Lecithin in this main figure rather than presenting the same figures again in the supplemental data with additional P values (Supplemental Fig. 21).*

Author response: We gratefully thank the reviewer for the kind reminder. In revision, we have accordingly added the *P* values for the comparisons between control and Ver-PLGA@Lecithin to Fig. 5 of the previous manuscript (*i.e.*, Fig. 6 in the revised manuscript) and, to avoid dual-presentation of these same data, have removed the previous Supplementary Fig. 21 from the Supplementary Information file. Now, Fig. 6 in the revised manuscript appears.

**Fig. 6 Efficacy of Ver-PLGA@Lecithin mediated sonodynamic therapy.** **a.** Schematic illustrations on the triple therapy regimen used for treating gastric *H. pylori* infection **b.** Schematic illustration on the schedule of different treatments including sonodynamic therapy (*i.e.*, “Ver-PLGA@Lecithin + US”) to mouse models bearing gastric *H. pylori* infection and follow-up blood and tissue collection. **c.** *H. pylori* burden per gram of stomach tissues (in Log CFU/g) from mice treated with “Ver-PLGA@Lecithin + US” group, with those from mice treated with PBS (*i.e.*, control), triple therapy, US exposure alone, and Ver-PLGA@Lecithin alone included for comparison. In the box plot, the upper and lower quartile as outlined by top and bottom boundary were divided by line showing the median value (n = 5 biologically independent mice in once trial). Statistical analysis was carried out with a one-way ANOVA with Tukey’s multiple-comparison test. **d-f.** Serum levels of (d) interleukin-6 (IL-6), (e) tumor necrosis factor- $\alpha$  (TNF- $\alpha$ ), and (f) interleukin-1 beta (IL-1 $\beta$ ) in gastric *H. pylori* infection-bearing mouse models after differing treatment. Bar heights are reported as average  $\pm$  standard deviation (n = 5 biologically independent mice in once trial). Statistical analysis was carried out with a one-way ANOVA with Tukey’s multiple-comparison test. **g-i.** Serum levels of (g) pepsinogen I (PG-1), (h) pepsinogen II (PG-2) and (i) gastrin-17 (G-17) in gastric *H. pylori* infection-bearing mouse models at 48-h after differing treatment. Bar heights are reported as average  $\pm$  standard deviation (n = 5 biologically independent mice in once trial). Red dash-line indicates the average serum level in healthy mice. Statistical analysis was carried out with a one-way ANOVA with Tukey’s multiple-comparison test. **j.** Microscopy images of hematoxylin and eosin (H&E) stained stomach tissues collected from gastric *H. pylori* infection-bearing mouse models at 48-h after differing treatment.

*Avoid showing the same data shown in main figures again in slightly different form in the supplemental figures (e.g. left half of Fig. 5f is the same as left half of supplemental Fig. 22, many of the panels from Fig 7 are repeated in Suppl Fig 29, etc.). This will reduce the high number of supplemental figures for this study.*

Author response: We gratefully thank the reviewer for kindly pointing out this. In revision, we have accordingly replaced the high magnification H&E images in Figure 5f of the previous manuscript with those in the previous Supplementary Figure 22 (which show the entire glands or the luminal portions), which led to Figure 6j in the revised manuscript. Meanwhile, to avoid repeated presentation of the same images, we have accordingly removed the previous Supplementary Figure 22 from the Supplementary Information file.

In revision, we have adopted the reviewer’s suggestion and accordingly removed Supplementary Fig. 29i-l of the previous manuscript, which is shown as change in Supplementary Fig. 22 of the revised manuscript.

Moreover, to avoid dual-presentation of same data/images in other figures, we have accordingly double-checked all the figures in the manuscript (including those in the Supplementary Information file) and performed revision wherever necessary. Specifically, the changes we made are as follows.

1) In revision, we have added the *P* values for the comparisons between control and Ver-PLGA@Lecithin to Fig. 5 of the previous manuscript (*i.e.*, Fig. 6 in the revised manuscript) and,

to avoid dual-presentation of these same data, have removed the previous Supplementary Fig. 21 from the Supplementary Information file.

2) In revision, we have removed the previous Supplementary Fig. 26b-f, because the previous Supplementary Fig. 26b shows the schematic illustration on a same experimental schedule as that in the previous Fig. 6a and the data shown in previous Supplementary Fig. 26c-f have already been presented in the previous Fig. 6b-e.

3) In revision, to better demonstrate the comparison and difference between our sonodynamic therapy and triple therapy, we have replaced the previous Fig. 6g and the previous Fig. 6i with the previous Supplementary Fig. 26g and the previous Supplementary Fig. 26i, respectively, which are labeled as Figure 7g,i in the revised manuscript. Meanwhile, we have combined the previous Fig. 6g,i (*i.e.*, Supplementary Fig. 21a,b in the revised manuscript) and the previous Supplementary Fig. 26h,j (*i.e.*, Supplementary Fig. 21c,d in the revised manuscript) into a new figure that was numbered as Supplementary Fig. 21 in the revised manuscript.

4) In revision, we have moved the previous Supplementary Fig. 26a into the previous Figure 5 and labeled this panel as Figure 6a in the revised manuscript.

5) In the previous manuscript, the schematic illustration and the data presented in Supplementary Fig. 27 have been presented in Supplementary Fig. 24. So, in revision, we have accordingly removed Supplementary Fig. 27 of the previous manuscript.

6) In the previous manuscript, Supplementary Fig. 31a and Supplemental Fig. 31f present schematic illustrations on the experimental schedule of a same assay, and the data for the healthy and control groups in Supplementary Fig. 31b-e are the same as those for the healthy and control groups in Supplementary Fig. 31g-j. So, in revision, we have combined Supplementary Fig. 31a and Supplemental Fig. 31f into a single schematic illustration (*i.e.*, Figure 9a in the revised manuscript), have combined Supplementary Fig. 31g with Supplementary Fig. 31b into a single panel Supplementary Fig. 31b (*i.e.*, Figure 9b in the revised manuscript), have combined Supplementary Fig. 31h with Supplementary Fig. 31c into a single panel Supplementary Fig. 31c (*i.e.*, Figure 9c in the revised manuscript), have combined Supplementary Fig. 31i with Supplementary Fig. 31d into a single panel Supplementary Fig. 31d (*i.e.*, Figure 9d in the revised manuscript), and have combined Supplementary Fig. 31j with Supplementary Fig. 31e into a single panel Supplementary Fig. 31e (*i.e.*, Figure 9e in the revised manuscript).

7) In the previous manuscript, Supplementary Fig. 25b shows a same photograph as the one in Supplementary Fig. 25a but at slightly higher magnification. So, in revision, we have replaced the photograph shown in the previous Supplementary Fig. 25b with a high magnification one that was got by zooming-out a selected area of the photograph shown in the previous Supplementary Fig. 25a, which led to Supplementary Fig. 23 in the revised manuscript.

Now, figures mentioned here appear.

**Fig. 6 Efficacy of Ver-PLGA@Lecithin mediated sonodynamic therapy.** **a.** Schematic illustrations on the triple therapy regimen used for treating gastric *H. pylori* infection **b.** Schematic illustration on the schedule of different treatments including sonodynamic therapy (*i.e.*, “Ver-PLGA@Lecithin + US”) to mouse models bearing gastric *H. pylori* infection and follow-up blood and tissue collection. **c.** *H. pylori* burden per gram of stomach tissues (in Log CFU/g) from mice treated with “Ver-PLGA@Lecithin + US” group, with those from mice treated with PBS (*i.e.*, control), triple therapy, US exposure alone, and Ver-PLGA@Lecithin alone included for comparison. In the box plot, the upper and lower quartile as outlined by top and bottom boundary were divided by line showing the median value ( $n = 5$  biologically independent mice in once trial). Statistical analysis was carried out with a one-way ANOVA with Tukey’s multiple-comparison test. **d-f.** Serum levels of (d) interleukin-6 (IL-6), (e) tumor necrosis factor- $\alpha$  (TNF- $\alpha$ ), and (f) interleukin-1 beta (IL-1 $\beta$ ) in gastric *H. pylori* infection-bearing mouse models after differing treatment. Bar heights are reported as average  $\pm$  standard deviation ( $n = 5$  biologically independent mice in once trial). Statistical analysis was carried out with a one-way ANOVA with Tukey’s multiple-comparison test. **g-i.** Serum levels of (g) pepsinogen I (PG-1), (h) pepsinogen II (PG-2) and (i) gastrin-17 (G-17) in gastric *H. pylori* infection-bearing mouse models at 48-h after differing treatment. Bar heights are reported as average  $\pm$  standard deviation ( $n = 5$  biologically independent mice in once trial). Red dash-line indicates the average serum level in healthy mice. Statistical analysis was carried out with a one-way ANOVA with Tukey’s multiple-comparison test. **j.** Microscopy images of hematoxylin and eosin (H&E) stained stomach tissues collected from gastric *H. pylori* infection-bearing mouse models at 48-h after differing treatment.

**Supplementary Figure 23. Pictures of mice receiving US exposure in sonodynamic therapy.** The ultrasound probe was placed snugly on the skin over stomach of a C57BL/6J mouse. Prior to the US exposure, the mouse was subjected to a depilatory treatment on the skin over stomach.

**Supplementary Figure 22. Biosafety of our sonodynamic therapy.** The extended data for Fig. 7, which show serum levels of (a) albumin and (b) uric acid (UA) in the liver and renal function tests and blood levels of (c) hematocrit, (d) mean corpuscular hemoglobin (MCH), (e) mean corpuscular volume (MCV), (f) mean corpuscular hemoglobin concentration (MCHC), (g) red blood cell distribution width-standard deviation (RDW-SD) and (h) red blood cell volume distribution width-coefficient of variation (RDW-CV), (i) mean platelet volume (MPV), (j) plateletcrit (PCT) %, (k) platelet-larger cell ratio (P-LCR). and (l) platelet distribution width (PDW) in the CBC test. All blood samples collected on day 9 from mice treated with different therapies. Bar heights are reported as average  $\pm$  standard deviation (n = 5 biologically independent mice in

once trial). Statistical analysis was carried out with a one-way ANOVA with Tukey’s multiple-comparison test.

**Supplementary Figure 21. Gut microbiota after triple therapy and sonodynamic therapy.** Statistical analysis on how (a, b) sonodynamic therapy and (c, d) triple therapy respectively affect the composition of gut commensal bacteria, using (a, c) the healthy group and (b, d) the control group as the references. The change in relative abundance of gut bacterial species after sonodynamic therapy is indicated both as (a) the absolute value of change compared to the healthy group (*i.e.*, “Relative Abundance (Sonodynamic therapy-Healthy)”) and (c) that compared to the control group (*i.e.*, “Relative Abundance (Sonodynamic therapy-Control)”). Similarly, the change in relative abundance of gut bacterial species after triple therapy is indicated both as (c) the absolute value of change compared to the healthy group (*i.e.*, “Relative Abundance (Triple therapy-Healthy)”) and (d) that compared to the control group (*i.e.*, “Relative Abundance (Triple therapy-Control)”). For bacterium to be marked as a significantly perturbed one, its absolute value of change in bacterial relative abundance needs to be  $\geq 0.1\%$  or  $\leq -0.1\%$  and, meanwhile, its  $P$  value needs to be  $< 0.05$ . Bacterial species whose relative abundances got significantly up- and down-regulated after sonodynamic therapy are marked in red and in blue, respectively.

**Fig. 7 Effects of sonodynamic therapy on gut microbiota.** **a.** Schematic illustration on the schedule of differing treatment to mouse models bearing gastric *H. pylori* infection and follow-up feces collection for (b-e) gut microbiota analysis. **b-c.** The (b)  $\alpha$  diversity and (c)  $\beta$  diversity of gut microbiota by principal component analysis (PCA) in mouse models after differing treatment, with those of the healthy group included for comparison. In the box plot, the upper and lower quartile as outlined by top and bottom boundary were divided by line showing the median value. Statistical analysis was carried out with a one-way ANOVA with Tukey's multiple-comparison test. **d-e.** The relative abundances of gut bacterial species at the (d) phylum and (e) genus levels in mouse models after differing treatment, with those of the healthy group included for comparison. The data were clustered according to the number of samples per group on the abscissa. **f-i.** Statistical analysis on how (f, h) sonodynamic therapy and (g, h) triple therapy respectively affect the composition of gut commensal bacteria, using (f, g) the healthy group and (h, i) the control group as the references. The change in relative abundance of gut bacterial species after sonodynamic therapy is indicated both as (f) the fold of change compared to the healthy group (*i.e.*, " $\log_2(\text{Sonodynamic therapy/Healthy})$ ") and (h) that compared to the control group (*i.e.*, " $\log_2(\text{Sonodynamic therapy/Control})$ "). Similarly, the change in relative abundance of gut bacterial species after triple therapy is indicated both as (g) the fold of change compared to the healthy group (*i.e.*, " $\log_2(\text{Triple therapy/Healthy})$ ") and (i) that compared to the control group (*i.e.*, " $\log_2(\text{Triple therapy/Control})$ "). For bacterium to be marked as a significantly perturbed one, its fold change in bacterial relative abundance needs to be  $\geq 2$  or  $\leq 1/2$  and, meanwhile, its *P* value needs to be  $< 0.05$ . Bacterial species whose relative abundances got significantly up- and down-regulated after sonodynamic therapy are marked in red and in blue, respectively.

**Fig. 9 Effects of different treatments on serum level of IL-1RA.** **a.** Schematic illustration on the schedule of different treatments including triple therapy and sonodynamic therapy (i.e., “Ver-PLGA@Lecithin + US”) to mouse models bearing gastric *H. pylori* infection and on that of follow-up blood sample collection for monitoring (b-e) serum levels of IL-1RA. **b-e.** Serum levels of IL-1RA in mouse models at different time-points after treatment completion. Bar heights are reported as average  $\pm$  standard deviation (n = 5 biologically independent mice in one trial). Statistical analysis was carried out with a one-way ANOVA with Tukey’s multiple-comparison test.

6. Lines 294-296: “Ver-PLGA@Lecithin alone suppressed the *H. pylori* infection-induced up-regulation of TNF- $\alpha$  ( $0.01 < P < 0.05$ ) (Fig. 5c-e and Supplementary Fig. 21), likely owing to its ability to remove VacA (Fig. 1f)”. Please temper this statement (and similar statements in the following sentence, the abstract, and elsewhere in the manuscript), since the impact of VacA absorption to the nanoparticle in vivo has not been investigated. I suggest replacing “likely” with “possibly”.

Author response: We gratefully thank the reviewer for the kind suggestion. In revision, we have accordingly replaced “likely” in the sentence mentioned here with “possibly”. Now, it reads:

“... Of note, Ver-PLGA@Lecithin alone significantly suppressed the *H. pylori* infection-induced up-regulation of IL-6, TNF- $\alpha$ , and IL-1 $\beta$  ( $P < 0.05$ ) (Fig. 6d-f), possibly owing to its ability to remove VacA (Fig. 2f), ...”

*7. Please make sure that Figures including supplementary figures are numbered in the order of their appearance in the text.*

Author response: We gratefully thank the reviewer for the kind reminder. In revision, we have accordingly checked the numbering of figures in the manuscript (including those supplementary figures) and made sure that they are numbered in the order of their appearance in the text. For example, in the Supplementary Information file, we have moved the previous Supplementary Fig. 19 to the position marked as Supplementary Fig. 15. We have also re-numbered panels f-i in Fig. 5 (specifically, f *versus* g-i) of the previous manuscript, to make their order of appearance in the figure (the results on PG-1, PG-2 and G-17 level analysis before the H&E staining images) mirrors that in the text.

*Reviewer #2.*

*The authors addressed all my points raised before. I would recommend the manuscript to be accepted as it is.*

Author response: We gratefully thank the reviewer for the highly positive remarks.

## REVIEWERS' COMMENTS

Reviewer #1 (Remarks to the Author):

The authors have addressed all points raised in my previous critique. I would recommend publication of the manuscript, but have small remaining point that I would like the authors to clarify:

Previous point #2 and supplementary figure 19: It is still not entirely clear how the image analysis for TUNEL staining was performed and how the “percentage of green fluorescence intensity” was determined and what this readout means. Please include additional information about the image analysis that was performed, e.g., specific analysis modules or ImageJ plugins that were used. Were images thresholded? Individual cells identified? Was only green fluorescence that colocalized with blue fluorescence measured? The bright green cluster in Suppl. Fig. 19a/triple therapy (which had the highest intensity score) looks morphologically different from the TUNEL-positive nuclei in the other treatment groups.

## *Responses to Reviewer Comments*

### **A biocompatible nanosensitizer for eliminating VacA and *Helicobacter pylori* without disrupting gut microbiota**

*Tao Liu*1,2,3, *Shuang Chai*1,2,3, *Mingyang Li*1,2,3, *Xu Chen*1,2,3, *Yutao Xie*1,2,3, *Zehui Zhao*1,2,3,  
*Jingjing Xie*1,2,3, *Yunpeng Yu*1,2,3, *Feng Gao*1, *Feng Zhu*4, *Lihua Yang*1,2,3\*

\*\*\*\*\*

#### **REVIEWER COMMENTS**

*Reviewer #1 (Remarks to the Author):*

*The authors have addressed all points raised in my previous critique. I would recommend publication of the manuscript, but have small remaining point that I would like the authors to clarify:*

*Previous point #2 and supplementary figure 19: It is still not entirely clear how the image analysis for TUNEL staining was performed and how the “percentage of green fluorescence intensity” was determined and what this readout means.*

Author response: We gratefully thank the reviewer for the critical comments. In the previous round of revision, we re-imaged the TUNEL-stained tissue slices under a fluorescence microscope and, for tissue samples from each treatment group, imaged 50 different fields of view at 32× magnification. The resultant fluorescence microscopy images were then subjected to statistical analysis with ImageJ (ImageJ2) software. Specifically, for each fluorescence image, the average intensity of green fluorescence and that of blue fluorescence over the whole image were obtained as specific values from ImageJ, and the percentage of the average green fluorescence intensity was calculated by dividing the average green fluorescence intensity by the sum of the average green fluorescence intensity and the average blue fluorescence intensity.

In this way, we calculated the percentage of the average green fluorescence intensity for each of the 300 fluorescence images (6 treatment groups, with 50 images for each treatment group). Since there are 50 images for each treatment group, we further calculated the mean and standard deviation for the percentages of the average green fluorescence intensity over the 50 images for each treatment group (Supplementary Figure 19).

**Supplementary Figure 19. Analysis on TUNEL stained stomach tissues.** **a.** Representative fluorescence microscopy images of TUNEL (TdT-mediated dUTP Nick-End Labeling) stained stomach tissues collected from gastric *H. pylori* infection-bearing mouse models at 48-h after differing treatments (*i.e.*, control, triple therapy, US exposure alone, Ver-PLGA@Lecithin alone, and “Ver-PLGA@Lecithin + US”), with those from their healthy counterparts included as references. Blue fluorescence (by DAPI) indicates DNA, and green fluorescence (by FITC-dUTP) indicates apoptotic cell death. **b.** Percentages of green fluorescence intensity in (a) the images of TUNEL stained tissue samples, which were obtained by imaging 50 different regions for each treatment group. Bar heights are reported as average  $\pm$  standard deviation. Statistical analysis was carried out with a one-way ANOVA with Tukey’s multiple-comparison test.

The green fluorescence indicates TUNEL positive cells and the blue fluorescence indicates TUNEL negative cells. Therefore, the as-calculated percentage of the average green fluorescence intensity for one image indicates the percentage of TUNEL positive cells relative to all cells in that

image, and the mean percentage of the average green fluorescence intensity calculated over the 50 images of one treatment group indicates the average percentage of TUNEL-positive cells relative to all imaged cells for that treatment group.

*Please include additional information about the image analysis that was performed, e.g., specific analysis modules or ImageJ plugins that were used. Were images thresholded? Individual cells identified?*

Author response: We gratefully thank the reviewer for attention to details. The analysis using ImageJ was performed through the following steps:

Step 1, launch ImageJ and open the image file to be analyzed.

Step 2, click “Image” on the toolbar and, in its drop-down list, select “Color” and then “Split Channels”.

Step 3, click “Image” on the toolbar and, in its drop-down list, select “Adjust” and then “Threshold”.

Step 4, in the auto-popup window titled “Threshold”, select “Default”, check “Dark background”, click “Auto”, and then close this “Threshold” window; the reason why we selected “Default” here was to eliminate possible errors caused by manually selecting the threshold for different images and, after this step was done, individual cells could be identified.

Step 5, click “Analyze” on the toolbar and, in its drop-down list, select “Measure”, which makes the software to run automatically;

Step 6, ImageJ automatically returns the results in a window titled “Results”, in which values of “Area”, “Mean”, “Min”, “Max”, “Min Thr” and “Max Thr” for the splitted color channel are listed.

*Was only green fluorescence that colocalized with blue fluorescence measured?*

Author response: We gratefully thank the reviewer for the critical comments. In fact, for each fluorescence image, we analyzed all the green and blue fluorescence signals in that image and did not pay extra attention to whether the green fluorescence colocalized with the blue fluorescence.

*The bright green cluster in Suppl. Fig. 19a/triple therapy (which had the highest intensity score) looks morphologically different from the TUNEL-positive nuclei in the other treatment group*

Author response: We gratefully thank the reviewer for attention to details. In fact, bright green clusters in the fluorescence images of TUNEL stained tissue slices were observed not only in the triple therapy group but also in other treatment groups, as shown in Figure below.

**Figure. Analysis on TUNEL stained stomach tissues.** Representative fluorescence microscopy images of TUNEL (TdT-mediated dUTP Nick-End Labeling) stained stomach tissues collected from gastric *H. pylori* infection-bearing mouse models at 48-h after different treatments (*i.e.*, control, triple therapy, US exposure alone, Ver-PLGA@Lecithin alone, and sonodynamic therapy (*i.e.*, Ver-PLGA@Lecithin + US)), with those from their healthy counterparts included as references. Blue fluorescence (by DAPI) indicates DNA, and green fluorescence (by FITC- dUTP) indicates apoptotic cell death.